# A Comprehensive Framework for Analyzing the Convergence of Adam: Bridging the Gap with SGD

**Ruinan Jin** [1]  **Xiao Li** [2]  **Yaoliang Yu** [3 4]  **Baoxiang Wang** [2 4]

## Abstract

Adaptive moment estimation (Adam) is a cornerstone optimization algorithm in deep learning, widely recognized for its flexibility with adaptive learning rates and efficiency in handling large-scale data. However, despite its practical success, the theoretical understanding of Adam's convergence has been constrained by stringent assumptions, such as almost surely bounded stochastic gradients or uniformly bounded gradients, which are more restrictive than those typically required for analyzing stochastic gradient descent (SGD).

In this paper, we introduce a novel and comprehensive framework for analyzing the convergence properties of Adam. This framework offers a versatile approach to establishing Adam's convergence. Specifically, we prove that Adam achieves asymptotic (last iterate sense) convergence in both the almost sure sense and the $L_1$ sense under the relaxed assumptions typically used for SGD, namely $L$-smoothness and the ABC inequality. Meanwhile, under the same assumptions, we show that Adam attains non-asymptotic sample complexity bounds similar to those of SGD.

## 1. Introduction

Adaptive Moment Estimation (Adam) is one of the most widely used optimization algorithms in deep learning due to its adaptive learning rate properties and efficiency in handling large-scale data (Kingma & Ba, 2014). Despite its

widespread use, the theoretical understanding of Adam's convergence is not as advanced as its practical success. Previous studies have often imposed stringent assumptions on the loss function and stochastic gradients, such as uniformly bounded loss functions and almost surely bounded gradients (Reddi et al., 2018; Zou & Shen, 2019), which are more restrictive than those required for analyzing classical stochastic gradient descent (SGD).

In this paper, we introduce a novel and comprehensive framework for analyzing the convergence properties of Adam. Our framework unifies various aspects of convergence analysis, including non-asymptotic (average iterate sense) sample complexity, asymptotic (last iterate sense) almost sure convergence, and asymptotic $L_1$ convergence. Crucially, we demonstrate that under this framework, Adam can achieve convergence under the same assumptions typically used for SGD—namely, the $L$-smooth condition and the ABC inequality ($L_2$ sense) (Khaled & Richtárik, 2023; Bottou, 2010; Ghadimi & Lan, 2013).

Several recent works have attempted to relax the stringent conditions required for Adam's convergence, each focusing on different aspects of the stochastic gradient assumptions and convergence guarantees. However, limitations still exist in terms of assumptions and the types of convergence results obtained. Table 1 provides the references and a summary of the works and compares the assumptions on stochastic gradients, the resulting complexities, and the convergence properties achieved.

Our approach builds upon these prior works and seeks to offer a more comprehensive and general framework for analysis. In contrast to these previous works, we study Adam under the ABC inequality, which is more general and less restrictive compared to the assumptions made in the previous studies. Our analysis successfully establishes non-asymptotic sample complexity and achieves asymptotic almost sure convergence and $L_1$ convergence under conditions that align with those required for SGD. This makes our framework theoretically sound and versatile for analyzing multiple convergence properties of Adam. Our framework might also be of independent interest in analyzing different variants of Adam. In summary, our work presents a novel and general theoretical framework for Adam, unifying vari-

---

[1]Centre for Artificial Intelligence and Robotics, Hong Kong Institute of Science and Innovation, Chinese Academy of Sciences, New Territories, Hong Kong, China. Most of the work of Ruinan Jin was completed during he was with the Vector Institute, Toronto, Canada, and The Chinese University of Hong Kong, Shenzhen, China. [2]The Chinese University of Hong Kong, Shenzhen, China [3]University of Waterloo, Waterloo, Canada [4]Vector Institute, Toronto, Canada. Correspondence to: Xiao Li <lixiao@cuhk.edu.cn>.

*Proceedings of the $42^{nd}$ International Conference on Machine Learning*, Vancouver, Canada. PMLR 267, 2025. Copyright 2025 by the author(s).

ous convergence properties. This framework demonstrates that Adam's convergence guarantees can be aligned with those of SGD, which justifies the applicability of Adam across a wide range of machine learning problems.

### 1.1. Related Works

In recent years, the convergence properties of Adam have been extensively studied, with various works focusing on different assumptions about stochastic gradients and the types of convergence guarantees provided. In the following discussion, we categorize and review key contributions based on the different types of stochastic gradient assumptions they employ, as summarized in Table 1.

**Bounded Variance and Coordinate Affine Noise Variance:** Wang et al. (2024a) considered Adam's convergence under the assumption of bounded variance or coordinate affine noise variance. The coordinate affine noise variance condition (Eq. (10)) is particularly stringent as it requires that each component of the stochastic gradient satisfies an affine noise variance inequality, which is stronger than the traditional affine noise variance condition (Eq. (9)) applied to the entire gradient. Under these assumptions, Wang et al. successfully achieved a complexity free of $\mathcal{O}(1/\mu)$. However, their work did not focus on analyzing almost sure convergence or $L_1$ convergence, as the primary emphasis was on the sample complexity of the algorithm's behavior.

**Exponential-Tailed Affine Variance Noise Condition** Hong & Lin (2024) explored the assumption of affine variance noise with an exponential tail distribution (Eq. (11)), which closely approximates the almost-sure form of affine variance noise. The exponential-tailed affine variance noise condition is stronger than the traditional affine variance noise assumption, which is based on the second moment of the stochastic gradient. Under these assumptions, they successfully derived a complexity that eliminates $\mathcal{O}(1/\mu)$. However, their work did not focus on analyzing almost sure convergence or $L_1$ convergence, as their primary emphasis was on the sample complexity of the algorithm's performance.

**Almost Surely Bounded Stochastic Gradients:** Several works, including He et al. (2023) and Xiao et al. (2024), have explored Adam's convergence under the assumption that the stochastic gradients are almost surely bounded. This is a particularly strong assumption, as it implies several other commonly made assumptions about stochastic gradients, such as bounded variance, affine noise variance, coordinate affine noise variance, and sub-Gaussian properties. The assumption is often impractical in non-convex settings where gradients can become unbounded. Moreover, studies in

Wang et al. (2023) have highlighted that this assumption is unrealistic in many common machine learning frameworks, failing to hold even for simple quadratic functions, let alone for deep neural networks. While these works achieved almost sure convergence and, in some cases, $L_1$ convergence, the complexity result they obtained includes $\mathcal{O}(1/\mu)$.

$L_2$ **Bounded Stochastic Gradients:** Zou et al. (2019) analyzed Adam under the assumption of $L_2$ bounded stochastic gradients. Although this condition is milder than the almost surely bounded gradients assumption, it is still stronger than the traditional affine noise variance condition and the ABC inequality. In the standard analytical framework, this assumption can at best be weakened to the coordinate affine noise variance condition, which remains more restrictive than the assumptions typically considered for SGD. At the same time, this work focused on complexity analysis without addressing asymptotic convergence.

**Randomly Reshuffled Stochastic Gradients:** In other works, such as those by Zhang et al. (2022) and Wang et al. (2024b), the authors considered the case where the stochastic gradients are randomly reshuffled. Randomly reshuffled stochastic gradients represent a special case where the gradients are typically assumed to satisfy certain inequalities almost surely. This reliance on almost sure properties forms a much stronger and more restrictive analytical framework compared to those based on traditional affine noise variance conditions or the ABC inequality. Meanwhile, they did not focus on analyzing the asymptotic convergence property.

## 2. Preliminaries

In this section, we introduce the necessary preliminaries and establish the foundational framework for our convergence analysis of Adam. We begin by recalling the Adam optimization algorithm. We then state the assumptions that will be used throughout our analysis. These assumptions are standard in stochastic optimization and are crucial for deriving our main results. By laying out these assumptions explicitly, we also facilitate a clear comparison with the conditions used in previous works, highlighting the less restrictive nature of our approach.

### 2.1. Adam

Adam is an extension of SGD that computes adaptive learning rates for each parameter by utilizing estimates of the first and second moments of the gradients.

It combines the advantages of two other extensions of SGD: AdaGrad, which works well with sparse gradients, and RMSProp, which works well in online and non-stationary settings.

*Table 1.* Comparison of Assumptions and Convergence Results. ($\diamondsuit$) The smoothing term $\mu$ is often set to small values like $10^{-8}$ in practice. It is difficult and relevant to avoid the $\mathcal{O}(\text{poly}(\frac{1}{\mu}))$ dependence (Wang et al., 2024a), which our analysis achieves. ($\spadesuit$) The work focuses on learning rates and hyperparameters dependent on the total number of epochs $T$, leading to results without a $\mathcal{O}(\ln T)$ term. As our asymptotic analysis uses $T$-independent parameters, terms regarding $\mathcal{O}(\ln T)$ inevitably appear, though our method can be easily extended to $T$-dependent settings. ($\diamondsuit\diamondsuit$) These works have weakened the classical $L$-smooth condition, which is different from the focus of this paper.

| Reference | Assumptions on Stochastic Gradient | Sample Complexity | A.S. Convergence | $L_1$ Convergence |
|---|---|---|---|---|
| (Wang et al., 2024a)$\spadesuit$ | Bounded Variance (or Coordinate Affine Noise Variance) | $\mathcal{O}\left(\frac{1}{\sqrt{T}}\right)$ | No | No |
| (Hong & Lin, 2024) | Exponential-tailed Affine Variance Noise | $\mathcal{O}\left(\frac{\ln T}{\sqrt{T}}\right)$ | No | No |
| (He et al., 2023)$\diamondsuit$ | Almost Surely Bounded Stochastic Gradient | $\mathcal{O}\left(\text{poly}\left(\frac{1}{\mu}\right) \cdot \frac{\ln T}{\sqrt{T}}\right)$ | Yes | Yes |
| (Zou et al., 2019) | $L_2$ Bounded Stochastic Gradient | $\mathcal{O}\left(\frac{\ln T}{\sqrt{T}}\right)$ | No | No |
| (Zhang et al., 2022) | Randomly Reshuffled Stochastic Gradient | $\mathcal{O}\left(\frac{\ln T}{\sqrt{T}}\right)$ | No | No |
| (Li et al., 2024)$\diamondsuit$ $\diamondsuit\diamondsuit$ | Almost Surely Bounded Stochastic Gradient or Sub-Gaussian Variance | $\mathcal{O}\left(\text{poly}\left(\frac{1}{\mu}\right) \cdot \frac{\ln T}{\sqrt{T}}\right)$ | No | No |
| (Wang et al., 2024b)$\diamondsuit\diamondsuit$ | Randomly Reshuffled Stochastic Gradient | $\mathcal{O}\left(\frac{\ln T}{\sqrt{T}}\right)$ | No | No |
| (Xiao et al., 2024)$\diamondsuit\diamondsuit$ | Almost Surely Bounded Stochastic Gradient | No Result | Yes | No |
| **Our Work** | ABC Inequality | $\mathcal{O}\left(\frac{\ln T}{\sqrt{T}}\right)$ | Yes | Yes |

---

**Algorithm 1** Adam

**Input:** Stochastic oracle $\mathcal{O}$, initial learning rate $\eta_1 \geq 0$, initial iterate $w_1 \in \mathbb{R}^d$, initial exponential moving averages $m_0 = 0$, $v_0 = v \cdot \mathbf{1}^\top$ with $v > 0$, hyperparameters $\beta_1 \in [0,1)$, $\beta_{2,1} \in (0,1]$, smoothing term $\mu > 0$, number of epochs $T$

**Output:** Final iterate $w_T$

    $t = 1$ **to** $T$ Generate learning rate $\eta_t$; Generate conditioner parameter $\beta_{2,t}$; Sample a random data point $z_t$ and compute the stochastic gradient $g_t = \mathcal{O}_f(w_t, z_t)$; Update the estimate: $v_t = \beta_{2,t} v_{t-1} + (1 - \beta_{2,t}) g_t^{\circ 2}$; Update the estimate: $m_t = \beta_1 m_{t-1} + (1 - \beta_1) g_t$; Compute the adaptive learning rate: $\eta_{v_t} = \eta_t \cdot \frac{1}{\sqrt{v_t} + \mu}$; Update the iterate: $w_{t+1} = w_t - \eta_{v_t} \circ m_t$;

---

In Adam, the random variables $\{z_t\}_{t \geq 1}$ are mutually independent. The stochastic gradient at epoch $t$ is denoted by $g_t$. The quantities $m_t$ and $v_t$ represent the exponential moving averages of the first and second moments of the gradients, respectively. The hyperparameters $\beta_1$ and $\beta_{2,t}$ control the exponential decay rates for the moment estimates. A small smoothing term $\mu$ is introduced to prevent division by zero, and $\eta_{v_t}$ represents the adaptive learning rate for each parameter.

**Notations:** The Hadamard product (element-wise multiplication) is represented by $\beta \circ \gamma$, and the element-wise square root of a vector $\gamma \in \mathbb{R}^d$ is written as $\sqrt{\gamma}$. Operations such as $\beta + v_0$, $\frac{1}{\beta}$, and $\beta^{\circ 2}$ are performed element-wise. Additionally, for a vector with subscripts, such as $\beta_t$, we use $\beta_{t,i}$ to denote its $i$-th coordinate. However, for a scalar with subscripts, such as $\Phi_t$, the double subscript $\Phi_{t,i}$ carries a specific meaning, which will be explicitly defined when it

appears.

When analyzing Adam, $\nabla f(w_t)$ refers to the true gradient of the loss function at epoch $t$. We define $\mathscr{F}_t = \sigma(g_1, \ldots, g_t)$ as the $\sigma$-algebra generated by the stochastic gradients up to epoch $t$, with $\mathscr{F}_0 = \{\Omega, \emptyset\}$ and $\mathscr{F}_\infty = \sigma\left(\bigcup_{t \geq 1} \mathscr{F}_t\right)$. Throughout this paper, unless explicitly stated otherwise, the norm $\|\cdot\|$ denotes the Euclidean norm.

## 2.2. Assumptions

To establish our convergence results, we make the following standard assumptions. The assumption regarding smoothness is the classical $L$-smooth assumption. The assumption about stochastic gradient are less restrictive than those imposed in some prior works, as highlighted in Table 1.

**Assumption 2.1.** *(Bounded from Below Loss Function) Let $f : \mathbb{R}^d \to \mathbb{R}$ be a loss function defined on $\mathbb{R}^d$. We assume that there exists a constant $f^* \in \mathbb{R}$ such that for all $w \in \mathbb{R}^d$, the following inequality holds: $f(w) \geq f^*$.*

This assumption ensures that the loss function $f$ is bounded from below, preventing it from decreasing indefinitely during the optimization process.

**Assumption 2.2.** *(L-Smoothness) Let $f : \mathbb{R}^d \to \mathbb{R}$ be a differentiable loss function. We assume that the gradient $\nabla f$ is Lipschitz continuous. That is, there exists a constant $L_f \geq 0$ such that for all $w, w' \in \mathbb{R}^d$, the following inequality holds: $\|\nabla f(w) - \nabla f(w')\| \leq L_f \|w - w'\|$. The constant $L_f$ is known as the Lipschitz constant of the gradient.*

**Assumption 2.3.** *(ABC Inequality) We assume that the stochastic gradient $g_t$ is an unbiased estimate of the true gradient, i.e., $\mathbb{E}[g_t \mid \mathscr{F}_{t-1}] = \nabla f(w_t)$, and there exist constants $A, B, C \geq 0$ such that for all epochs $t$, we have:*

$$\mathbb{E}[\|g_t\|^2 \mid \mathscr{F}_{t-1}] \leq A(f(w_t) - f^*) + B\|\nabla f(w_t)\|^2 + C.$$

The ABC inequality provides a bound on the second moment of the stochastic gradients, which is crucial for analyzing the convergence of stochastic optimization algorithms. Notice identity $\mathbb{E}\left[\|g_t - \nabla f(w_t)\|^2 \mid \mathscr{F}_{t-1}\right] = \mathbb{E}\left[\|g_t\|^2 \mid \mathscr{F}_{t-1}\right] - \|\nabla f(w_t)\|^2$. We can conclude that the above ABC inequality has the following equivalent form based on the variance of the stochastic gradients, i.e., there exist constants $A \geq 0$, $B \geq 0$, and $C \geq 0$ such that: $\mathbb{E}\left[\|g_t - \nabla f(w_t)\|^2 \mid \mathscr{F}_{t-1}\right] \leq A(f(w_t) - f^*) + B\|\nabla f(w_t)\|^2 + C$.

### 2.3. Comparison with Prior Works on Stochastic Gradient Assumptions

**Due to space limitations in the main text, these comparisons have been moved to Appendix A.**

Next, we introduce a property. We know that when the loss function is $L$-smooth, the true gradient of the loss function can be controlled by the loss function value $f(w_t) - f^*$ (as shown in Lemma C.2). Therefore, we can simplify the ABC inequality as follows.

**Property 1.** *Under Assumptions 2.2 and 2.3, for all epochs $t$, we have:* $\mathbb{E}[\|g_t\|^2 \mid \mathscr{F}_{t-1}] \leq (A + 2L_f B)(f(w_t) - f^*) + C$.

This property demonstrates that the variance of the stochastic gradients can be bounded by the function value difference, which is a key component in our convergence analysis.

### 2.4. Hyperparameter Settings

In this paper, to keep the proofs concise, we focus on a specific set of representative parameter configurations, defined as follows:

$$\beta_{2,t} := \begin{cases} 1 - \alpha_0, & \text{if } t = 1, \\ 1 - \frac{1}{t^\gamma}, & \text{if } t \geq 2, \end{cases} \quad \beta_1 \in [0, 1), \quad \eta_t = \frac{1}{t^{\frac{1}{2}+\delta}},$$

where $\alpha_0 \in [0, 1), \gamma \in [1, 2\delta + 1]$, and $\delta \in \left[0, \frac{1}{2}\right)$.

It is essential to impose certain constraints on Adam's parameters, particularly on $\beta_{2,t}$, to ensure the algorithm converges. Early studies (Reddi et al., 2018) have shown that without proper restrictions on $\beta_{2,t}$, counterexamples exist where the algorithm fails to converge. Furthermore, for the gradient norm to converge to zero, it is necessary that $\beta_{2,t}$ approaches 1, as noted in earlier works (Zou et al., 2019; He et al., 2023).

Some studies on complexity allow $\beta_{2,t}$ to be constant. However, these studies typically focus on the algorithm's complexity over a finite number of epochs $T$. In such cases, the constant value of $1 - \beta_{2,t}$ is inversely related to $T$, effectively causing $\beta_{2,t}$ to approach 1 as $T$ increases. This is another

means of ensuring that $\beta_{2,t}$ asymptotically approaches 1, which is crucial for convergence.

The hyperparameter settings adopted in this paper are representative and have been considered in previous studies (Zou et al., 2019; He et al., 2023). Our configuration includes settings that can achieve near-optimal complexity of $\mathcal{O}(\ln T/\sqrt{T})$. The logarithmic factor $\ln T$ arises because $\beta_{2,t}$ is chosen independent of the total number of epochs $T$, which is an unavoidable consequence with this class of parameters.

Our choice of hyperparameters simplifies the analysis while capturing the essential behavior of the Adam. Although the proof techniques can be extended to a broader range of parameter settings, this paper focuses primarily on the assumptions related to the convergence of the algorithm rather than an exhaustive exploration of hyperparameter configurations.

## 3. Theoretical Results

In this section, we establish both non-asymptotic and asymptotic convergence guarantees for Adam within our smooth non-convex framework, as defined by Assumptions 2.1–2.3. For the non-asymptotic analysis, we derive a sample complexity bound that is independent of $\mathcal{O}(1/\mu)$, providing an explicit bound on the number of epochs required to achieve a specified accuracy. In the asymptotic analysis, we consider two forms of convergence: almost sure convergence and convergence in the $L_1$ norm. The almost sure convergence result demonstrates that, the gradient norm of almost every trajectory converges to zero. Meanwhile, the $L_1$ convergence result reveals that the convergence across different trajectories is uniform with respect to the $L_1$ norm of the gradient, where the $L_1$ norm is taken in the sense of the underlying random variable, meaning the expectation of the gradient norm.

### 3.1. Non-Asymptotic Sample Complexity

**Theorem 3.1** (Non-Asymptotic Sample Complexity). *Consider the Adam algorithm as specified in Algorithm 2.1, and assume that Assumptions 2.1–2.3 are satisfied. Then, for any initial point, any $T \geq 1$, and any $s \in (0, 1)$, the following bound holds with probability at least $1 - s$:*

$$\frac{1}{T}\sum_{t=1}^{T}\|\nabla f(w_t)\|^2 \leq \begin{cases} \mathcal{O}\left(\frac{1}{s^2}\frac{1}{T^{\frac{1}{2}-\delta}}\right), & \text{if } \delta \in (0, 1/2) \\ \mathcal{O}\left(\frac{1}{s^2}\frac{\ln T}{\sqrt{T}}\right), & \text{if } \gamma > 1, \ \delta = 0 \\ \mathcal{O}\left(\frac{1}{s^2}\frac{\ln^2 T}{\sqrt{T}}\right), & \text{if } \gamma = 1, \ \delta = 0. \end{cases}$$

*The constant implied by the $\mathcal{O}$ notation depends on the initial point, the constants in Assumptions 2.1–2.3 (excluding $1/\mu$), and the parameters $\delta$ and $\alpha_0$.*

This theorem provides a non-asymptotic rate of convergence for the square of the gradient norm, highlighting how the choice of hyperparameters affects the convergence rate.

## 3.2. Asymptotic Convergence

We now present our main asymptotic convergence results, demonstrating that the gradients of the Adam converge to zero both almost surely and in the $L_1$ sense under appropriate conditions.

**Theorem 3.2 (Asymptotic Almost Sure Convergence).** *Under Assumptions 2.1–2.3, consider the Adam with hyperparameters specified in Subsection 2.4 with $\gamma > 1$ and $\delta > 0$. Then, the gradients of the Adam converge to zero almost surely, i.e., $\lim_{t\to\infty} \|\nabla f(w_t)\| = 0$   a.s.*

This theorem shows that the gradients evaluated at the iterates converge to zero almost surely, indicating that the algorithm approaches a critical point of the loss function along almost every trajectory.

**Remark 1.** *(Almost sure vs $L_1$ convergence) As stated in the introduction, it is important to note that the almost sure convergence does not imply $L_1$ convergence. To illustrate this concept, let us consider a sequence of random variables $\{\zeta_n\}_{n\geq 1}$, where $\mathbb{P}(\zeta_n = 0) = 1 - 1/n^2$ and $\mathbb{P}(\zeta_n = n^2) = 1/n^2$. According to the Borel-Cantelli lemma, it follows that $\lim_{n\to+\infty} \zeta_n = 0$ almost surely. However, it can be shown that $\mathbb{E}[|\zeta_n|] = 1$ for all $n > 0$ by simple calculations.*

**Theorem 3.3 (Asymptotic $L_1$-Convergence).** *Under Assumptions 2.1–2.3, consider the Adam with hyperparameters specified in Subsection 2.4 with $\gamma > 1$ and $\delta > 0$. Then, the gradients of the Adam converge to zero in the $L_1$ sense, i.e., $\lim_{t\to\infty} \mathbb{E}[\|\nabla f(w_t)\|] = 0$.*

This result establishes convergence in the mean sense, showing that the expected gradient norm approaches zero as the number of epochs increases. It indicates that the convergence of gradient norms across different trajectories is uniform in the $L_1$ norm of the random variables.

In previous works (He et al., 2023; Xiao et al., 2024), the assumption that the stochastic gradients are uniformly bounded, i.e., $\|g_t\| \leq M$  a.s. $(\forall\, t \geq 1)$, or that the gradients themselves are uniformly bounded, i.e., $\|\nabla f(w_t)\| \leq M$ $(\forall\, t \geq 1)$, allows almost sure convergence to directly imply $L_1$ convergence via the *Lebesgue's Dominated Convergence* theorem. However, in our framework, which deals with potentially unbounded stochastic gradients or gradients, proving $L_1$ convergence is much more challenging. We will elaborate on this in the next section.

# 4. Framework for Analyzing Adam

In this section, we present the analytical framework that underpins our convergence analysis for the Adam. Our approach is built upon the insights provided by existing methods, while introducing new techniques to address the limitations of previous analyses and provide a more comprehensive understanding of Adam's behavior under weaker assumptions. Our core innovations are detailed in Section 4.3.1, Section 4.4, and Section 4.5.

## 4.1. Key Properties of Adaptive Learning Rates

We begin by characterizing the fundamental properties of the adaptive learning rate sequence $\eta_{v_t}$ in Section 2.4. These properties are critical as they directly influence the behavior of the algorithm and are foundational to our subsequent analysis. By understanding how these properties interact with the algorithm's dynamics, we obtain more insights on the conditions under which Adam converges.

**Property 2.** *Each element $\eta_{v_t,i}$ of the sequence $\{\eta_{v_t}\}_{t\geq 1} = \{[\eta_{v_t,1}, \eta_{v_t,2}, \ldots, \eta_{v_t,d}]^\top\}_{t\geq 1}$ is monotonically decreasing with respect to $t$.*

This property ensures that the learning rate becomes progressively smaller as the algorithm progresses, which is a crucial factor in the stability and convergence of Adam.

**Property 3.** *Each element $\eta_{v_t,i}$ of the sequence $\{\eta_{v_t}\}_{t\geq 1} = \{[\eta_{v_t,1}, \eta_{v_t,2}, \ldots, \eta_{v_t,d}]^\top\}_{t\geq 1}$ satisfies the inequality $t^\gamma v_{t,i} \geq \alpha_1 S_{t,i}$, where we define $\alpha_1 := \min\{1 - \alpha_0, \alpha_0\}$, $S_{t,i} := v + \sum_{k=1}^t g_{k,i}^2$ for all $t \geq 1$, and $S_{0,i} := v$.*

This property highlights the relationship between the accumulated gradient information $S_{t,i}$ and the adaptive learning rate, ensuring that the latter appropriately scales with the former as epochs proceed.

**Remark 4.1.** *For the purpose of simplifying the proofs of subsequent theorems, we define two auxiliary parameters: $\Sigma_{v_t} := \sum_{i=1}^d v_{t,i}$ and $S_t := \sum_{i=1}^d S_{t,i}$. Additionally, for convenience in the subsequent proofs, we define a new initial parameter based on $S_{0,i}$ as $\eta_{v_0,i} = S_{0,i}/\alpha_1 = v/\alpha_1$.*

These definitions of auxiliary parameters help streamline the analysis, making the mathematical expressions more manageable and the proofs more concise.

With the key properties of the adaptive learning rates established, we now turn our attention to analyzing the momentum term, which plays a crucial role in the Adam.

## 4.2. Handling the Momentum Term

To effectively analyze the momentum term in the Adam, we adopt a classical method introduced by Liu et al. (2020). The momentum term introduces additional complexity in the analysis due to its recursive nature, which can complicate the convergence proofs. To address this, we construct an auxiliary variable $u_t$ that simplifies the analysis by decoupling the momentum term from the update process. This

auxiliary variable is defined as follows:

$$u_t := \frac{w_t - \beta_1 w_{t-1}}{1 - \beta_1} = w_t + \frac{\beta_1}{1 - \beta_1}(w_t - w_{t-1})$$

$$= w_t - \frac{\beta_1}{1 - \beta_1}\eta_{v_{t-1}} \circ m_{t-1}. \qquad (1)$$

The introduction of $u_t$ allows us to handle the momentum term more effectively by transforming the recursive nature of the updates into a more tractable form. Specifically, we can express the relationship between successive epochs of $u_t$ as follows:

$$u_{t+1} - u_t = -\eta_{v_t} \circ g_t$$
$$+ \frac{\beta_1}{1 - \beta_1}\underbrace{(\eta_{v_{t-1}} - \eta_{v_t})}_{\Delta_t} \circ m_{t-1}. \qquad (2)$$

This recursive relation is instrumental in breaking down the complex dependencies introduced by the momentum term, which will facilitate the convergence analysis.

Then we establish two key properties that connect the original variable $w_t$ and the auxiliary variable $u_t$. These properties are crucial for bounding the changes in the momentum term and connecting the function values at different points in the epoch process.

**Property 4.** *For any epoch t, the following inequality holds:*

$$m_{t,i}^2 - m_{t-1,i}^2 \le -(1 - \beta_1)m_{t-1,i}^2 + (1 - \beta_1)g_{t,i}^2.$$

This property establishes a bound on the change in the momentum term, which is critical for ensuring that the momentum does not increase indefinitely during the optimization process. Controlling the momentum in this manner is an important step in proving the convergence.

**Property 5.** *For any epoch t, the following inequality holds:*

$$f(w_t) \le (L_f + 1)f(u_t) + \frac{(L_f + 1)\beta_1^2}{2(1 - \beta_1)^2}\left\|\eta_{v_{t-1}} \circ m_{t-1}\right\|^2.$$

This property connects the function values at $w_t$ and $u_t$, which provides a foundation for analyzing the convergence of $f(w_t)$. By establishing this relationship, we can relate the behavior of the original variable $w_t$ to the more manageable auxiliary variable $u_t$, thereby simplifying the overall convergence analysis.

### 4.3. Establishing the Approximate Descent Inequality

In the convergence analysis of stochastic gradient descent (SGD), a fundamental tool is the *approximate descent inequality*, which quantifies the expected decrease in the objective function at each epoch. Specifically, for SGD, the approximate descent inequality is given by:

$$f(w_{t+1}) - f(w_t) \le \underbrace{-\eta_t\|\nabla f(w_t)\|^2}_{Descent\ Term} + \underbrace{\frac{\eta_t^2 L}{2}\|g_t\|^2}_{Quadratic\ Error}$$

$$+ \underbrace{\eta_t\nabla f(w_t)^\top(\nabla f(w_t) - g_t)}_{Martingale\ Difference\ Term}, \quad (3)$$

where $\eta_t$ is the learning rate, $L$ is the Lipschitz constant, and $g_t$ is the stochastic gradient.

Motivated by the success of this approach in analyzing SGD, we aim to establish a similar approximate descent inequality for the Adam. The goal is to develop a descent inequality that captures the adaptive nature of Adam's learning rates while maintaining the essential structure seen in the analysis of SGD.

To this end, we present the following key result, which forms the cornerstone of our convergence analysis for Adam.

**Lemma 4.1 (Approximate Descent Inequality).** *Consider the sequences $\{w_t\}_{t\ge 1}$, $\{v_t\}_{t\ge 1}$, and $\{u_t\}_{t\ge 1}$ generated by Algorithm 2.1 and Eq. (2). Under Assumptions 2.1–2.3, the following sufficient decrease inequality holds:*

$$\Pi_{\Delta,t}\hat{f}(u_{t+1}) - \Pi_{\Delta,t-1}\hat{f}(u_t)$$

$$\le -\frac{1}{2}\Pi_{\Delta,t}\sum_{i=1}^d \zeta_i(t) + C_2\|\eta_{v_{t-1}} \circ m_{t-1}\|^2$$

$$+ \sum_{i=1}^d \Delta_{t,i}|\nabla_i f(u_t)m_{t-1,i}|$$

$$+ (L_f + 1)\sum_{i=1}^d \eta_{v_t,i}^2 g_{t,i}^2 + \Pi_{\Delta,t}M_t. \qquad (4)$$

*Here,*

$$\hat{f}(u_t) := f(u_t) - f^* + C\sum_{i=1}^d \eta_{v_{t-1},i},$$

$$\zeta_i(t) := \eta_{v_{t-1},i}(\nabla_i f(w_t))^2,$$

$$\Pi_{\Delta,t} := \prod_{k=1}^t \left(1 + \left(\frac{D_1}{1 - \sqrt{\beta_1}} + 1\right)\overline{\Delta}_{\sqrt{\beta_1},k}\right)^{-1} \quad (t \ge 1),$$

$$\Pi_{\Delta,0} := 1,$$

$$\overline{\Delta}_{\sqrt{\beta_1},k} := \sum_{i=1}^d \mathbb{E}\left[\sum_{t=k}^{+\infty}(\sqrt{\beta_1})^{t-k}\Delta_{t,i}\Big|\mathscr{F}_{k-1}\right],$$

*where, $\Delta_{t,i}$ denotes the $i$-th component of $\Delta_t$ (defined in Eq. 2). $M_t := M_{t,1} + M_{t,2} + M_{t,3}$.* $\qquad (5)$

*Constants $C_2, D_1$ is defined in Eq. (26) and Lemma E.2; $M_{t,1}$ is defined in Eq. (20); $M_{t,2}$ and $M_{t,3}$ are defined in Eq. (21).*

This lemma introduces $\Pi_{\Delta,t-1}\hat{f}(u_t)$ as a new Lyapunov function for Adam, which plays a crucial role in our analysis. In Eq. (4), the term $-\frac{1}{2}\Pi_{\Delta,t}\sum_{i=1}^d \zeta_i(t)$ can be interpreted as the descent term, representing the expected decrease in the Lyapunov function. We collectively refer to the 2nd,

3rd, and 4th terms on the right side of the inequality as the quadratic error terms. According to subsequent results (Lemma D.2), we can show that the expectation of the summation from 1 to $T$ over $t$ of these terms is of the same order as $\mathcal{O}\left(\sum_{t=1}^{T}\sum_{i=1}^{d}\mathbb{E}\left[\eta_{v_t,i}^2 g_{t,i}^2\right]\right)$. The 5th term, $\Pi_{\Delta,t}M_t$, is a martingale difference sequence with respect to the filtration $\{\mathcal{F}_t\}_{t\geq 1}$, which, due to its zero expectation, can be considered to have no overall impact on the algorithm's epoch process.

This structure closely resembles the approximate descent inequality commonly used in the analysis of SGD. For comparison, the approximate descent inequality for SGD is given by Eq. (3).

We now proceed to provide the main idea of proving Lemma 4.1 and highlight the key steps and challenges involved in establishing this result for Adam.

To begin with, we calculate the difference in the loss function values between two consecutive auxiliary variables $\{u_t\}_{t\geq 1}$ that we introduced. We obtain the following expression (informal):

$$f(u_{t+1}) - f(u_t)$$

$$\leq -\underbrace{\sum_{i=1}^{d}\mathbb{E}\left[\eta_{v_t,i}\nabla_i f(w_t)g_{t,i}|\mathcal{F}_{t-1}\right]}_{Term_{t,1}} + \underbrace{\mathcal{O}\left(\sum_{i=1}^{d}\eta_{v_t,i}^2 g_{t,i}^2\right)}_{Term_{t,2}}$$

$$+ \underbrace{\sum_{i=1}^{d}\mathbb{E}\left[\eta_{v_t,i}\nabla_i f(w_t)g_{t,i}|\mathcal{F}_{t-1}\right] - \sum_{i=1}^{d}\eta_{v_t,i}\nabla_i f(w_t)g_{t,i}}_{Term_{t,3}}$$

$$+ R_t. \tag{6}$$

It can be observed that the above equation is simply a second-order Taylor expansion of $f(u_{t+1}) - f(u_t)$ (since an L-smooth function is almost everywhere twice differentiable). $Term_{t,1}$ represents the first-order term, which in general serves as the descent term. $Term_{t,2}$ is the quadratic error, and $Term_{t,3}$ is a martingale difference sequence. The remaining term $R_t$ is negligible and can be ignored. In the informal explanation provided in the sketch, these were collectively referred to as remainder terms. For the exact formulation, refer to the detailed proof in Appendix D.3.2.

While handling the quadratic error term $Term_{t,2}$ is relatively straightforward using standard scaling techniques, addressing the first-order term $Term_{t,1}$ is more challenging due to the adaptive nature of Adam's learning rates. Specifically, $\eta_{v_t,i}$ and $g_{t,i}$ are both $\mathcal{F}_t$-measurable, which necessitates the introduction of an auxiliary random variable $\tilde{\eta}_{v_t,i}$ which is $\mathcal{F}_{t-1}$-measurable to facilitate the extraction of the learning rate from the conditional expectation. In this paper, we choose the auxiliary random variable $\eta_{v_{t-1},i}$ to approximate $\eta_{v_t,i}$. There are also other forms of this approximation, as

discussed by (Wang et al., 2023; 2024a). This allows us to rewrite the first-order term as:

$$-Term_{t,1} = -\sum_{i=1}^{d}\mathbb{E}\left[\eta_{v_t,i}\nabla_i f(w_t)g_{t,i}|\mathcal{F}_{t-1}\right]$$

$$= -\underbrace{\sum_{i=1}^{d}\mathbb{E}\left[\eta_{v_{t-1},i}\nabla_i f(w_t)g_{t,i}|\mathcal{F}_{t-1}\right]}_{Descent\text{-}Term_t}$$

$$+ \underbrace{\sum_{i=1}^{d}\mathbb{E}\left[(\eta_{v_{t-1},i} - \eta_{v_t,i})\nabla_i f(w_t)g_{t,i}|\mathcal{F}_{t-1}\right]}_{Term_{t,4}}.$$

The presence of $Term_{t,4}$ introduces an additional layer of complexity in the analysis, as it reflects the difference between successive adaptive learning rates. Addressing this extra error term is crucial for establishing robust convergence guarantees under the ABC inequality or affine noise variance conditions. Existing approaches to handling such terms, which often rely on the cancellation of errors through preceding descent terms, fall short in this context. This necessitates a more innovative strategy, which we present in the following section.

### 4.3.1. ADDRESSING THE EXTRA ERROR TERM: OUR INNOVATIVE APPROACH

The term $Term_{t,4}$, introduced by the difference between $\eta_{v_{t-1},i}$ and $\eta_{v_t,i}$, presents a significant challenge in the convergence analysis of Adam under the ABC inequality or affine noise variance conditions. In existing methods, it is common to attempt to cancel out such error terms by leveraging the preceding descent term $Descent\text{-}Term_t$. However, this approach might not work within the ABC framework. Recent works such as Wang et al. (2023; 2024a) have shown that, under existing techniques, the best one can achieve is a weakened form of the stochastic gradient assumption, namely the coordinate affine noise variance condition. To overcome these limitations, we introduce a novel approach to handle $Term_{t,4}$. We scale it as follows:

$$Term_{t,4} \leq \frac{1}{2}\sum_{i=1}^{d}\mathbb{E}\left[\eta_{v_{t-1},i}\nabla_i f(w_t)g_{t,i}\mid\mathcal{F}_{t-1}\right]$$

$$+ C_1 f(u_t)\cdot\sum_{i=1}^{d}\mathbb{E}[\Delta_{t,i}\mid\mathcal{F}_{t-1}] + \tilde{R}_t$$

$$+ C\sum_{i=1}^{d}\Delta_{t,i} + C\underbrace{\sum_{i=1}^{d}\left(\mathbb{E}[\Delta_{t,i}\mid\mathcal{F}_{t-1}] - \Delta_{t,i}\right)}_{Term_5},$$

where $\Delta_{t,i} := \eta_{v_{t-1},i} - \eta_{v_t,i}$, $C_1 := \frac{A+2L_f B}{2}(L_f+1)$, and $\tilde{R}_t$ represents a negligible remainder term, primarily stemming from the difference between $f(w_t)$ and $f(u_t)$. The

critical term in this inequality is $C_1 f(u_t) \sum_{i=1}^{d} \mathbb{E}[\Delta_{t,i} \mid \mathscr{F}_{t-1}]$, which cannot be effectively canceled out using existing methods.

To handle this issue, we assign $\overline{\Delta}_t := \sum_{i=1}^{d} \mathbb{E}[\Delta_{t,i} \mid \mathscr{F}_{t-1}]$, and move the term $C_1 \overline{\Delta}_t f(u_t)$ to the left-hand side of inequality 6 and combine it with the existing $f(u_t)$ term. This leads to a new epoch inequality of the form:

$$
\begin{aligned}
& f(u_{t+1}) - (1 + C_1 \overline{\Delta}_t) f(u_t) \\
& \leq -\frac{1}{2} \text{Descent-Term}_t + \text{M-Term}_t + \text{Term}_{t,2} \\
& + \text{R-Term}_t.
\end{aligned} \tag{7}
$$

In the inequality $\text{M-Term}_t = \text{Term}_{t,3} + \text{Term}_{t,5}$ is a martingale difference sequence and $\text{R-Term}_t$ is the (neglectable) remainder term by combining all other terms from the inequalities. To express this inequality in a form resembling a Lyapunov function, we introduce an auxiliary product variable: $\Pi_{\Delta,t} := \prod_{k=1}^{t} (1 + C_1 \overline{\Delta}_k)^{-1} \quad (\forall t \geq 2), \quad \Pi_{\Delta,1} := 1$ (Informal). Note that $\Pi_{\Delta,t}$ here is merely a simplified version of the actual $\Pi_{\Delta,t}$ used in the formal lemma; it is not the version we employ in practice. Multiplying both sides of the inequality by $\Pi_{\Delta,t}$, we obtain the following reformulated inequality:

$$
\begin{aligned}
& \Pi_{\Delta,t} f(u_{t+1}) - \Pi_{\Delta,t-1} f(u_t) \\
& \leq -\frac{1}{2} \Pi_{\Delta,t} \cdot \text{Descent-Term}_t + \Pi_{\Delta,t} \cdot \text{M-Term}_t \\
& + \Pi_{\Delta,t} \cdot \text{Term}_{t,2} + \Pi_{\Delta,t+1} \cdot \text{R-Term}_t.
\end{aligned} \tag{8}
$$

This reformulation introduces $\Pi_{\Delta,t}$ as a scaling factor, which, along with the original Lyapunov function, captures the impact of $\text{Term}_{t,4}$. The resulting inequality closely parallels the approximate descent inequality for SGD, with additional terms accounting for Adam's adaptive nature.

The handling of $\text{Term}_{t,4}$ in our analysis framework is a significant advancement over existing methods. It allows us to establish stronger convergence guarantees under more general conditions.

### 4.4. Deriving Sample Complexity and Almost Sure Convergence

After establishing the *Approximate Descent Inequality*, the next step is to derive the sample complexity and almost sure convergence results for Adam. The methodology for obtaining these results largely mirrors the approaches traditionally used in the analysis of SGD. Specifically, the inequality provides a foundation for bounding the expected decrease in the loss function, which can then be used to establish both sample complexity and almost sure convergence.

However, a key difference in our analysis lies in the introduction of the term $\Pi_{\Delta,t}$ within the *Approximate Descent*

*Inequality*. This term introduces a new layer of complexity not present in the standard SGD analysis. In particular, we are required to bound the $p$-th moment of the reciprocal of this term, i.e., $\mathbb{E}[\Pi_{\Delta,t}^{-p}]$, $(p \geq 1)$. Due to the unique structure of $\Pi_{\Delta,t+1}$, determining a bound for this $p$-th moment is a non-trivial task.

To address this challenge, we leverage tools from discrete martingale theory, particularly the *Burkholder's* inequality. It allows us to establish a recursive relationship between the $p$-th moment $\mathbb{E}[\Pi_{\Delta,t}^{-p}]$ and the $p/2$-th moment $\mathbb{E}[\Pi_{\Delta,t}^{-p/2}]$. This recursive structure is crucial as it enables us to iteratively bound the higher moments of $\Pi_{\Delta,t}^{-1}$.

Once the recursive relationship is established, we apply fundamental theorems from measure theory, such as the *Lebesgue's Monotone Convergence* theorem or the *Lebesgue's Dominated Convergence* theorem, to obtain the final bound on the $p$-th moment. The detailed process for bounding $\mathbb{E}[\Pi_{\Delta,t}^{-p}]$ can be found in Lemma C.3, Lemma C.5 and Lemma D.1.

### 4.5. Establishing Asymptotic $L_1$ Convergence

Since we have already proved almost sure convergence in Theorem 3.2, it is natural to attempt to prove $L_1$ convergence via the *Lebesgue's Dominated Convergence* theorem. To achieve this, we need to find a function $h$ that is $\mathscr{F}_\infty$-measurable and satisfies $\mathbb{E}|h| < +\infty$, and such that for all $t \geq 1$, we have $\|\nabla f(w_t)\| \leq |h|$. Since for all $t$ we naturally have $\|\nabla f(w_t)\| \leq \sup_{k \geq 1} \|\nabla f(w_k)\|$, we only need to prove that $\mathbb{E}[\sup_{k \geq 1} \|\nabla f(w_k)\|] < +\infty$.

This task presents a significant challenge because, within our analytical framework, we cannot assume that the gradients are uniformly bounded, which means we cannot directly apply the *Lebesgue's Dominated Convergence* theorem. Instead, we need to utilize advanced techniques from discrete martingale theory, specifically the first hitting time decomposition method, to obtain a bound on this maximal expectation. The detailed process can be found in Appendix D.3.13.

## 5. Conclusion

We have introduced a novel and comprehensive framework for analyzing the convergence properties of Adam. Our frame starts with weak assumptions such as the ABC inequality. By identifying the key properties of the learning rate, handling the momentum term, and establishing the approximate descent inequality, the frame concludes the sample complexity, almost surely convergence, and asymptotic $L_1$ convergence results of Adam. Our techniques overcome existing limitations, aligning Adam's convergence guarantees with those of SGD, thereby justifying Adam's broad applicability in machine learning.

## Acknowledgements

We thank the anonymous reviewers for their helpful feedback and suggestions. Xiao Li is supported in part by the National Natural Science Foundation of China (NSFC) under grant 12201534 and in part by the Shenzhen Science and Technology Program under grant RCYX20221008093033010. Baoxiang Wang is partially supported by the National Natural Science Foundation of China (72394361, 62106213) and an extended support project from the Shenzhen Science and Technology Program.

## Impact Statement

This paper presents work whose goal is to advance the field of Machine Learning. There are many potential societal consequences of our work, none of which we feel must be specifically highlighted here.

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

# Contents

## A. Comparison with Prior Works on Stochastic Gradient Assumptions

Our assumption on the stochastic gradient (Assumption 2.3) is relatively mild compared to those in prior works. Here, we focus on comparing with the traditional affine noise variance condition, coordinate affine noise variance assumption, and the almost surely bounded stochastic gradient assumption.

**Traditional Affine Noise Variance Condition**    The traditional affine noise variance condition (Affine noise variance) (e.g., (Bottou et al., 2018; Nguyen et al., 2018)) assumes that there exist constants $B \geq 0$ and $C \geq 0$ such that:

$$\mathbb{E}[\|g_t\|^2 \mid \mathscr{F}_{t-1}] \leq B\|\nabla f(w_t)\|^2 + C. \tag{9}$$

This condition bounds the expected squared norm of the stochastic gradient by a linear function of the squared norm of the true gradient plus a constant. It is stronger than our ABC inequality because it does not include the term involving the function value difference $f(w_t) - f^*$.

Some works adopt the following form of affine variance noise:

$$\mathbb{E}\left[\|g_t - \nabla f(w_t)\|^2 \mid \mathscr{F}_{t-1}\right] \leq B\|\nabla f(w_t)\|^2 + C.$$

It is important to note that, due to the identity

$$\mathbb{E}\left[\|g_t - \nabla f(w_t)\|^2 \mid \mathscr{F}_{t-1}\right] = \mathbb{E}\left[\|g_t\|^2 \mid \mathscr{F}_{t-1}\right] - \|\nabla f(w_t)\|^2,$$

it is straightforward to see that these two forms are equivalent.

Even under this condition, current methods for analyzing Adam encounter significant difficulties. We will explain these challenges in the proof sketch of Lemma 4.1. Besides, both (Huang et al., 2021) and (Guo et al., 2021) provided convergence bounds under traditional affine noise variance condition. However, they relied on the assumption for step-size where $C_l \leq \left\|\frac{1}{\sqrt{v_t}+\mu}\right\|_\infty \leq C_u \ \forall \ t \in [T]$.

**Coordinate Affine Noise Variance Assumption**    Wang et al. (2024a) introduce the *coordinate affine noise variance* assumption, which requires that each component of the stochastic gradient satisfies an affine noise variance inequality. Specifically, for each coordinate $i$, there exist constants $B, C \geq 0$ such that:

$$\mathbb{E}[g_{t,i}^2 \mid \mathscr{F}_{t-1}] \leq B\|\nabla_i f(w_t)\|^2 + C, \tag{10}$$

where $g_{t,i}$ and $\nabla_i f(w_t)$ are the $i$-th components of $g_t$ and $\nabla f(w_t)$, respectively.

This assumption is stronger than the traditional affine noise variance condition because it imposes the inequality on each coordinate individually, rather than on the overall gradient.

**Exponential-tailed Affine Variance Noise Condition**    Certain works, such as (Hong & Lin, 2024), examine the assumption of affine variance noise with an exponential tail distribution, i.e.,

$$\mathbb{E}\left[\exp\left\{\frac{\|g_t - \nabla f(w_t)\|^2}{B\|\nabla f(w_t)\|^2 + C}\right\} \,\middle|\, \mathscr{F}_{t-1}\right] \leq e. \tag{11}$$

This assumption is close to the almost sure form of affine variance noise, specifically $\|g_t\|^2 \leq B\|\nabla f(w_t)\|^2 + C$ a.s. It is important to emphasize that this assumption (exponential-tailed affine variance noise condition) is stronger than the traditional affine variance noise assumption based on the second moment of the stochastic gradient. Furthermore, the methods developed under this stronger assumption are not applicable to affine variance noise models that rely on the second moment.

**Almost Surely Bounded Stochastic Gradient Assumption**    Some prior works, such as (He et al., 2023; Xiao et al., 2024), assume that the stochastic gradients are almost surely bounded. That is, there exists a constant $M \geq 0$ such that for all epochs $t$: $\|g_t\| \leq M$ almost surely. This is a strong assumption, as it requires that the stochastic gradient norm is uniformly bounded almost surely at all epochs. In practice, especially in non-convex optimization problems, this assumption

is often violated (see Wang et al. 2023). For instance, when optimizing deep neural networks, gradient norms can become unbounded due to the complexity and non-linearity of the models. Moreover, this assumption implies that the true gradient is also bounded by $M$, because $\|\nabla f(w_t)\|^2 \leq \mathbb{E}[\|g_t\|^2 \mid \mathscr{F}_{t-1}] \leq M^2$. Our assumption is clearly weaker than the almost surely bounded stochastic gradient assumption, as we only require a bound on the expected squared norm of the stochastic gradient, which can depend on the current function value and gradient norm, rather than a uniform almost sure bound.

Moreover, assuming almost surely bounded stochastic gradients is hard to satisfy in practice and may not reflect realistic scenarios. As discussed in (Wang et al., 2023; Khaled & Richtárik, 2023), such assumptions can be unrealistic and limit the applicability of theoretical results.

## B. Proofs of the Properties of Adaptive Learning Rates and Momentum Term

### B.1. Proof of Property 2

*Proof.* Due to Algorithm 2.1, we observe that

$$v_{t+1} = \beta_{2,t+1}v_t + (1 - \beta_{2,t+1})g_{t+1}^{\circ 2} = \left(1 - \frac{1}{(t+1)^\gamma}\right)v_t + \frac{1}{(t+1)^\gamma}g_{t+1}^{\circ 2}, \ (\forall\, t \geq 1).$$

which means

$$(t+1)^\gamma v_{t+1,i} = \left((t+1)^\gamma - 1\right)v_{t,i} + g_{t+1,i}^2 \geq t^\gamma v_{t,i}. \tag{12}$$

This implies that $t^\gamma v_{t,i}$ is monotonically non-decreasing. Subsequently, we have

$$\eta_{v_t,i} = \frac{\eta_t}{\sqrt{v_{t,i}} + \mu} = \frac{\sqrt{t^\gamma}\eta_t}{\sqrt{t^\gamma v_{t,i}} + \sqrt{t^\gamma}\mu} = \frac{\frac{1}{t^{\delta - \frac{\gamma-1}{2}}}}{\sqrt{t^\gamma v_{t,i}} + \sqrt{t^\gamma}\mu}.$$

Because the numerator is monotonically decreasing and greater than 0, while the denominator is monotonically non-increasing and greater than 0, we deduce the monotonic non-increasing property of $\eta_{v_t}$. □

### B.2. Proof of Property 3

*Proof.* For $v_{1,i}$, we derive the following estimate

$$v_{1,i} = \beta_{2,1}v_{0,i} + (1 - \beta_{2,1})g_{1,i}^2 = (1 - \alpha_0)v + \alpha_0 g_{1,i}^2.$$

It is immediate to find that $\alpha_1 S_{1,i} \leq v_{1,i} \leq S_{1,i}$. For $\forall\, k \geq 2$, by Eq. (12), we have $k^\gamma v_{k,i} \geq (k-1)^\gamma v_{k-1,i} + g_{k,i}^2$. Then, by summing up the above iterative equations, we obtain $\forall\, t \geq 2$,

$$t^\gamma v_{t,i} \geq v_{1,i} + \sum_{k=2}^{t} g_{k,i}^2.$$

Combining the estimate for $v_{1,i}$, we have $\forall\, t \geq 2$:

$$t^\gamma v_{t,i} \geq (1 - \alpha_0)v + \alpha_0 g_{1,i}^2 + \sum_{k=2}^{t} g_{k,i}^2.$$

Then $t^\gamma v_{t,i} \geq \alpha_1 S_{t,i}$, which completes the proof. □

### B.3. Proof of Property 4

*Proof.* According to Algorithm 2.1, we have the following iterative equations

$$m_{t,i} = \beta_1 m_{t-1,i} + (1 - \beta_1)g_{t,i}.$$

We take the square of the 2-norm on both sides, which yields

$$m_{t,i}^2 = (\beta_1 m_{t-1,i} + (1 - \beta_1)g_{t,i})^2$$

$$= \beta_1^2 m_{t-1,i}^2 + 2\beta_1(1-\beta_1)m_{t-1,i}g_{t,i} + (1-\beta_1)^2 g_{t,i}^2$$
$$\overset{(a)}{\leq} \beta_1 m_{t-1,i}^2 + (1-\beta_1)g_{t,i}^2.$$

In step (a), we used the *AM-GM* inequality, i.e.,

$$2\beta_1(1-\beta_1)m_{t-1,i}g_{t,i} \leq \beta_1(1-\beta_1)m_{t-1,i}^2 + \beta_1(1-\beta_1)g_{t,i}^2,$$

that is,

$$m_{t,i}^2 - m_{t-1,i}^2 \leq -(1-\beta_1)m_{t-1,i}^2 + (1-\beta_1)g_{t,i}^2,$$

which completes the proof. $\qquad\square$

### B.4. Proof of Property 5

*Proof.* Due to

$$|f(w_t) - f(u_t)| = \left| \nabla f(u_t)^\top (w_t - u_t) + \frac{L_f}{2}\|w_t - u_t\|^2 \right| \leq \|\nabla f(u_t)\|\|w_t - u_t\| + \frac{L_f}{2}\|w_t - u_t\|^2$$
$$\leq \frac{1}{2}\|\nabla f(u_t)\|^2 + \frac{L_f + 1}{2}\|w_t - u_t\|^2$$
$$= Lf(u_t) + \frac{(L_f + 1)\beta_1^2}{2(1-\beta_1)^2}\|\eta_{v_{t-1}} \circ m_{t-1}\|^2,$$

we have

$$f(w_t) \leq f(u_t) + |f(w_t) - f(u_t)| \leq (L_f + 1)f(u_t) + \frac{(L_f + 1)\beta_1^2}{2(1-\beta_1)^2}\|\eta_{v_{t-1}} \circ m_{t-1}\|^2. \qquad\square$$

## C. Lemmas in Probability Theory and Real Analysis

**Lemma C.1.** *If $0 < \mu < 1$ and $0 < \sigma < 1$ ($\sigma < \mu$) are two constants, then for any positive sequence $\{\psi_n\}$, there is*

$$\sum_{i=1}^n \mu^{n-i}\psi_i < \sum_{k=1}^n \mu^{n-k}\sum_{i=1}^k \sigma^{k-i}\psi_i \leq 1/(1-\omega_0)\sum_{i=1}^n \mu^{n-i}\psi_i,$$

*where $\omega_0 := \sigma/\mu$.*

**Lemma C.2.** *Suppose that $f(x)$ is differentiable and lower bounded, i.e. $f^* = \inf_{x \in \mathbb{R}^d} f(x) > -\infty$, and $\nabla f(x)$ is Lipschitz continuous with parameter $\mathcal{L} > 0$, then $\forall x \in \mathbb{R}^d$, we have*

$$\left\|\nabla f(x)\right\|^2 \leq 2\mathcal{L}(f(x) - f^*).$$

**Lemma C.3.** *Let $\{(X_n, \mathscr{F}_n)\}_{n\geq 1}$ be a non-negative adapted process such that $\sum_{n=1}^{+\infty} X_n = M < +\infty$ almost surely, where $M$ is a finite constant. Define the partial sum of conditional expectations as $\Lambda_T := \sum_{n=1}^T \mathbb{E}[X_n \mid \mathscr{F}_{n-1}]$. Then the following properties hold.*

(i) *The sequence $\{\Lambda_T\}_{T\geq 1}$ converges almost surely, i.e., $\Lambda_T \xrightarrow{a.s.} \Lambda$, where $\Lambda := \sum_{n=1}^{+\infty} \mathbb{E}[X_n \mid \mathscr{F}_{n-1}]$.*

(ii) *For any $p \geq 1$, the sequence $\{\Lambda_T\}_{T\geq 1}$ converges in $L_p$, i.e., $\lim_{T\to\infty} \mathbb{E}\left[|\Lambda_T - \Lambda|^p\right] = 0$. Meanwhile, the $p$-th moment of the limit $\Lambda$ is bounded by a constant $C_\Lambda(p) > 0$, where $C_\Lambda(p) = o((2M)^p p^{\sqrt{p}})$.*

**Lemma C.4.** *Let $l \in (0, 1)$. Then, for sufficiently large $n \in N_+$, we have*

$$\sum_{k=0}^\infty l^k k^{\sqrt{n}} \sim \frac{\Gamma\left(\sqrt{n} + 1\right)}{\left(\ln\frac{1}{l}\right)^{\sqrt{n}+1}}, \quad n \to \infty.$$

**Lemma C.5.** *Let $\{(X_n, \mathscr{F}_n)\}_{n \geq 1}$ be a non-negative adapted process such that $\sum_{n=1}^{+\infty} X_n = M < +\infty$ almost surely, where $M$ is a finite constant. For any $k > 1$, define the partial sum of conditional expectations as $\Lambda_{k,T} := \sum_{n=k}^{T} \mathbb{E}[X_n \mid \mathscr{F}_{n-k}]$. Then the following properties hold.*

(i) *The sequence $\{\Lambda_{k,T}\}_{T \geq 1}$ converges almost surely, i.e., $\Lambda_{k,T} \xrightarrow{a.s.} \Lambda^{(k)}$, where $\Lambda := \sum_{n=k}^{+\infty} \mathbb{E}[X_n \mid \mathscr{F}_{n-k}]$.*

(ii) *For any $p \geq 1$, the sequence $\{\Lambda_{k,T}\}_{T \geq 1}$ converges in $L_p$, i.e., $\lim_{T \to \infty} \mathbb{E}\left[|\Lambda_{k,T} - \Lambda^{(k)}|^p\right] = 0$. Meanwhile, the $p$-th moment of the limit $\Lambda^{(k)}$ is bounded by a constant $C_{\Lambda^{(k)}}(p) > 0$, where $C_{\Lambda}(p) = o((2M)^p(kp)^{\sqrt{p}})$.*

(iii) *For any $0 < l < 1$, the arbitrary $p$-th moment of the random variable $e^{\Lambda(l)}$ exists, where*

$$\Lambda(l) = \sum_{k=1}^{+\infty} \mathbb{E}\left[\left(\sum_{t=k}^{+\infty} l^{t-k} X_t\right)\Bigg| \mathscr{F}_{k-1}\right].$$

*The upper bound of this $p$-th moment depends only on $p$, $l$, and $M$. We denote this upper bound by $C_{e^{\Lambda(l)}}(p, M)$.*

## C.1. Proofs of These Lemmas

### C.1.1. PROOF OF LEMMA C.1

*Proof.* The proof of this lemma is through identities. We assume $\mu > \sigma$ (the case $\mu < \sigma$ is the similar), and let $\omega_0 = \log_\mu \sigma > 1$. Then we derive

$$\sum_{k=1}^{n} \mu^{n-k} \sum_{i=1}^{k} \sigma^{k-i}\psi_i = \sum_{k=1}^{n}\sum_{i=1}^{k} \mu^{n-k}\sigma^{k-i}\psi_i = \sum_{i=1}^{n}\sum_{k=i}^{n} \mu^{n-k}\sigma^{k-i}\psi_i = \sum_{i=1}^{n}\left(\sum_{k=i}^{n}\left(\frac{\sigma}{\mu}\right)^{k-i}\right)\mu^{n-i}\psi_i,$$

where $\omega_0 = \sigma/\mu$. Then combining $1 < \sum_{k=i}^{n}\left(\frac{\sigma}{\mu}\right)^{k-i} < \frac{1}{1-\omega_0}$ we get the result. $\qquad\square$

### C.1.2. PROOF OF LEMMA C.2

*Proof.* For $\forall x \in \mathbb{R}^d$, define the function

$$g(t) = f\left(x + t\frac{x' - x}{\|x' - x\|}\right),$$

where $x'$ is a constant point such that $x' - x$ is parallel to $\nabla f(x)$. By taking the derivative, we obtain

$$g'(t) = \nabla_{x+t\frac{x'-x}{\|x'-x\|}} f\left(x + t\frac{x' - x}{\|x' - x\|}\right)^\top \frac{x' - x}{\|x' - x\|}. \tag{13}$$

Through the Lipschitz condition of $\nabla f(x)$, we get $\forall t_1, t_2$

$$\left|g'(t_1) - g'(t_2)\right| = \left|\left(\nabla_{x+t\frac{x'-x}{\|x'-x\|}} f\left(x + t_1\frac{x' - x}{\|x' - x\|}\right) - \nabla_{x+t\frac{x'-x}{\|x'-x\|}} f\left(x + t_2\frac{x' - x}{\|x' - x\|}\right)\right)^\top \frac{x' - x}{\|x' - x\|}\right|$$

$$\leq \left\|\nabla_{x+t\frac{x'-x}{\|x'-x\|}} f\left(x + t_1\frac{x' - x}{\|x' - x\|}\right) - \nabla_{x+t\frac{x'-x}{\|x'-x\|}} f\left(x + t_2\frac{x' - x}{\|x' - x\|}\right)\right\|\left\|\frac{x' - x}{\|x' - x\|}\right\| \leq \mathcal{L}|t_1 - t_2|.$$

This indicates that $g'(t)$ satisfies the Lipschitz condition as well. Then $\inf_{t \in \mathbb{R}} g(t) \geq \inf_{x \in \mathbb{R}^d} f(x) > -\infty$. Let $g^* = \inf_{x \in \mathbb{R}} g(x)$. Subsequently, $\forall t_0 \in \mathbb{R}$,

$$g(0) - g^* \geq g(0) - g(t_0). \tag{14}$$

By using the *Newton-Leibniz's* formula,

$$g(0) - g(t_0) = \int_{t_0}^{0} g'(\alpha)d\alpha = \int_{t_0}^{0} \left(g'(\alpha) - g'(0)\right)d\alpha + \int_{t_0}^{0} g'(0)d\alpha.$$

Through the Lipschitz condition of $g'$, we get that

$$g(0) - g(t_0) \geq \int_{t_0}^{0} -\mathcal{L}|\alpha - 0|d\alpha + \int_{t_0}^{0} g'(0)d\alpha = \frac{1}{2\mathcal{L}}\left(g'(0)\right)^2.$$

Then we take a special value of $t_0$. Let $t_0 = -g'(0)/\mathcal{L}$. We obtain

$$
\begin{aligned}
g(0) - g(t_0) &\geq -\int_{t_0}^{0} \mathcal{L}|\alpha|d\alpha + \int_{t_0}^{0} g(0)dt = -\frac{\mathcal{L}}{2}(0 - t_0)^2 + g'(0)(-t_0) \\
&= -\frac{1}{2\mathcal{L}}\left(g'(0)\right)^2 + \frac{1}{\mathcal{L}}\left(g'(0)\right)^2 = \frac{1}{2\mathcal{L}}\left(g'(0)\right)^2.
\end{aligned}
\tag{15}
$$

Substituting Eq. (15) into Eq. (14), we have

$$g(0) - g^* \geq \frac{1}{2\mathcal{L}}\left(g'(0)\right)^2.$$

Due to $g^* \geq f^*$ and $\left(g'(0)\right)^2 = \|\nabla f(x)\|^2$, it follows that

$$\left\|\nabla f(x)\right\|^2 \leq 2\mathcal{L}\left(f(x) - f^*\right). \qquad \square$$

### C.1.3. PROOF OF LEMMA C.3

*Proof.* (i) Consider the non-negative adapted process $\{X_n, \mathscr{F}_n\}_{n \geq 1}$ and define the partial sum of conditional expectations as $\Lambda_T := \sum_{n=1}^{T} \mathbb{E}[X_n \mid \mathscr{F}_{n-1}]$.

First, we compute the expectation of $\Lambda_T$

$$\mathbb{E}[\Lambda_T] = \mathbb{E}\left[\sum_{n=1}^{T} \mathbb{E}[X_n \mid \mathscr{F}_{n-1}]\right] = \sum_{n=1}^{T} \mathbb{E}[X_n] \leq M.$$

Since $X_n$ are non-negative, we know that $\Lambda_T$ is a non-decreasing sequence. Because $\mathbb{E}(\Lambda_T)$ ($\forall T \geq 1$) is also bounded by $M$, we apply the *Lebesgue's Monotone Convergence* theorem. Thus, $\Lambda_T$ converges almost surely to a limit $\Lambda$:

$$\Lambda := \lim_{T \to \infty} \Lambda_T = \sum_{n=1}^{\infty} \mathbb{E}[X_n \mid \mathscr{F}_{n-1}] \quad \text{a.s.}$$

This concludes that the sequence of conditional expectation sums converges almost surely.

(ii) We begin by normalizing $X_n$ by considering the expression $Y_n = \frac{X_n}{2M}$. According to the *Lebesgue's Monotone Convergence* theorem, we only need to prove that

$$\forall \, p \geq 1, \quad \mathbb{E}\left[\sum_{n=1}^{\infty} \mathbb{E}[Y_n|\mathscr{F}_{n-1}]\right]^p := M(p) < +\infty.$$

Next, we proceed with the calculation, and we obtain that $\forall \, p \geq 2$, there is (Strictly speaking, we should first consider a finite $N$ and compute $\sum_{n=1}^{N}$, and only then take the limit as $N \to +\infty$, applying the *Lebesgue's monotone convergence theorem* to obtain the result for $\sum_{n=1}^{\infty}$. However, for the sake of simplicity in the proof, we have directly computed $\sum_{n=1}^{\infty}$.):

$$
\begin{aligned}
M(p) &= \mathbb{E}\left[\sum_{n=1}^{\infty} \mathbb{E}[Y_n|\mathscr{F}_{n-1}]\right]^p = \mathbb{E}\left[\sum_{n=1}^{\infty} Y_n + \sum_{n=1}^{\infty}(\mathbb{E}[Y_n|\mathscr{F}_{n-1}] - Y_n)\right]^p \\
&\overset{(a)}{\leq} \mathbb{E}\left[\frac{1}{2} + \sum_{n=1}^{\infty}(\mathbb{E}[Y_n|\mathscr{F}_{n-1}] - Y_n)\right]^p \overset{(b)}{\leq} 2^{p-1}\left(\frac{1}{2^p} + \mathbb{E}\left[\sum_{n=1}^{\infty}(\mathbb{E}[Y_n|\mathscr{F}_{n-1}] - Y_n)\right]^p\right) \\
&\overset{(c)}{\leq} \frac{1}{2} + 2^{p-1}C_p\,\mathbb{E}\left[\sum_{n=1}^{\infty}|\mathbb{E}[Y_n|\mathscr{F}_{n-1}] - Y_n|^2\right]^{p/2} \overset{(d)}{\leq} \frac{1}{2} + 2^{p-1}C_p\,\mathbb{E}\left[\sum_{n=1}^{\infty}|\mathbb{E}[Y_n|\mathscr{F}_{n-1}] - Y_n|\right]^{p/2}
\end{aligned}
$$

$$\overset{(f)}{\leq} \frac{1}{2} + 2^{p-2}C_p + 2^{\frac{3}{2}p-2}C_p \, \mathbb{E}\left[\sum_{n=1}^{\infty} \mathbb{E}[Y_n|\mathscr{F}_{n-1}]\right]^{p/2}$$

$$= \frac{1}{2} + 2^{p-2}C_p + 2^{\frac{3}{2}p-2}C_p M(p/2). \tag{16}$$

In the above derivation, Inequality $(a)$ requires noting that $\sum_{n=1}^{+\infty} Y_n = \frac{1}{2}$. Inequality $(b)$ uses the *AM-GM* inequality, specifically,

$$\left(\frac{a+b}{2}\right)^p \leq \frac{a^p + b^p}{2}.$$

Inequality $(c)$ involves using *Burkholder's* inequality,[1] where $C_p$ is a constant depending only on $p$, and its order with respect to $p$ is $\mathcal{O}(p)$ (see Theorem 5.27 in KHOSHNEVISAN (2006)). Inequality $(d)$ requires noting that

$$|\mathbb{E}[Y_n|\mathscr{F}_{n-1}] - Y_n|^2 \leq |\mathbb{E}[Y_n|\mathscr{F}_{n-1}] - Y_n|.$$

By repeatedly using Equation (16) and using the fact that $C_p = \mathcal{O}(p)$, we obtain the following estimate

$$M(p) = o(p^{\sqrt{p}}),$$

that is,

$$\mathbb{E}[\Lambda^p] = o((2M)^p \cdot p^{\sqrt{p}}). \qquad \square$$

### C.1.4. PROOF OF LEMMA C.4

*Proof.* Consider the function

$$f(x) = l^x x^{\sqrt{n}},$$

with its derivative

$$f'(x) = l^x x^{\sqrt{n}-1}(\ln l \cdot x + \sqrt{n}).$$

We observe that $f$ is decreasing for $x > \frac{\sqrt{n}}{\ln \frac{1}{l}}$. Therefore, we have the following estimate

$$0 \leq \sum_{0 \leq k \leq \frac{\sqrt{n}}{\ln \frac{1}{l}}+1} l^k k^{\sqrt{n}} \leq \left(\frac{\sqrt{n}}{\ln \frac{1}{l}}+1\right)^{\sqrt{n}} \sum_{k=0}^{\infty} l^k = \frac{1}{1-l}\left(\frac{\sqrt{n}}{\ln \frac{1}{l}}+1\right)^{\sqrt{n}} = O\left(\frac{\sqrt{n}}{\ln \frac{1}{l}}\right)^{\sqrt{n}}.$$

On the other hand, we can bound the remainder as follows:

$$\sum_{k > \frac{\sqrt{n}}{\ln \frac{1}{l}}+1} l^k k^{\sqrt{n}} \geq \sum_{k > \frac{\sqrt{n}}{\ln \frac{1}{l}}+1} \int_k^{k+1} l^x x^{\sqrt{n}} \, dx$$

$$= \sum_{k=0}^{\infty} \int_k^{k+1} l^x x^{\sqrt{n}} \, dx - \sum_{0 \leq k \leq \frac{\sqrt{n}}{\ln \frac{1}{l}}+1} \int_k^{k+1} l^x x^{\sqrt{n}} \, dx$$

$$\geq \int_0^{\infty} l^x x^{\sqrt{n}} \, dx - \left(\frac{\sqrt{n}}{\ln \frac{1}{l}}+2\right)^{\sqrt{n}} \sum_{k=0}^{\infty} \int_k^{k+1} l^x \, dx$$

$$= \frac{\Gamma\left(\sqrt{n}+1\right)}{\left(\ln \frac{1}{l}\right)^{\sqrt{n}+1}} + O\left(\frac{\sqrt{n}}{\ln \frac{1}{l}}\right)^{\sqrt{n}}.$$

---

[1] ***Burkholder's*** **inequality**: For any martingale $\{(M_n, \mathscr{F}_n)\}_{n \geq 1}$ with $M_0 = 0$ almost surely, and for any $1 \leq p < \infty$, there exist constants $c_p > 0$ and $C_p > 0$ depending only on $p$ such that:

$$c_p \, \mathbb{E}[(S(M))^p] \leq \mathbb{E}[(M^*)^p] \leq C_p \, \mathbb{E}[(S(M))^p],$$

where $M^* = \sup_{n \geq 0} |M_n|$ and $S(M) = \left(\sum_{i \geq 1} (M_i - M_{i-1})^2\right)^{1/2}$.

Similarly, we have the upper bound

$$
\sum_{k > \frac{\sqrt{n}}{\ln \frac{1}{l}} + 1} l^k k^{\sqrt{n}} \leq \sum_{k > \frac{\sqrt{n}}{\ln \frac{1}{l}} + 1} \int_{k-1}^{k} l^x x^{\sqrt{n}} \, dx
$$

$$
= \sum_{k=1}^{\infty} \int_{k-1}^{k} l^x x^{\sqrt{n}} \, dx - \sum_{1 \leq k \leq \frac{\sqrt{n}}{\ln \frac{1}{l}} + 1} \int_{k-1}^{k} l^x x^{\sqrt{n}} \, dx
$$

$$
\leq \int_{0}^{\infty} l^x x^{\sqrt{n}} \, dx = \frac{\Gamma\left(\sqrt{n} + 1\right)}{\left(\ln \frac{1}{l}\right)^{\sqrt{n}+1}}.
$$

Combining the estimates above, we have

$$
\sum_{k=0}^{\infty} l^k k^{\sqrt{n}} \sim \frac{\Gamma\left(\sqrt{n} + 1\right)}{\left(\ln \frac{1}{l}\right)^{\sqrt{n}+1}}, \quad n \to \infty. \qquad \square
$$

### C.1.5. PROOF OF LEMMA C.5

*Proof.* (i) Consider the non-negative adapted process $\{X_n, \mathscr{F}_n\}_{n \geq 1}$ and define the partial sum of conditional expectations as $\Lambda_{k,T} := \sum_{n=k}^{T} \mathbb{E}[X_n \mid \mathscr{F}_{n-k}]$.

First, we compute the expectation of $\Lambda_{k,T}$

$$
\mathbb{E}[\Lambda_{k,T}] = \mathbb{E}\left[\sum_{n=k}^{T} \mathbb{E}[X_n \mid \mathscr{F}_{n-k}]\right] = \sum_{n=k}^{T} \mathbb{E}[X_n] < \sum_{n=1}^{T} \mathbb{E}[X_n] \leq M.
$$

Since $X_n$ are non-negative, we know that $\Lambda_{k,T}$ is a non-decreasing sequence, and considering that $\mathbb{E}(\Lambda_{k,T}) (\forall T \geq 1)$ is also bounded by $M$, we apply the *Lebesgue's Monotone Convergence* theorem. Thus, $\Lambda_{k,T}$ converges almost surely to a limit $\Lambda^{(k)}$

$$
\Lambda^{(k)} := \lim_{T \to \infty} \Lambda_{k,T} = \sum_{n=k}^{\infty} \mathbb{E}[X_n \mid \mathscr{F}_{n-k}] \quad \text{a.s.}
$$

This concludes that the sequence of conditional expectation sums converges almost surely.

(ii) We begin by normalizing $X_n$ by considering the expression $Y_n = \frac{X_n}{2M}$. According to the *Lebesgue's Monotone Convergence* theorem, we only need to prove that

$$
\forall\, p \geq 1, \quad \mathbb{E}\left[\sum_{n=k}^{\infty} \mathbb{E}[Y_n \mid \mathscr{F}_{n-k}]\right]^{p} := M_k(p) < +\infty.
$$

Next, we proceed with the calculation, and we obtain $\forall\, p \geq 2$, there is:

$$
M(p) = \mathbb{E}\left[\sum_{i=0}^{k-1} \sum_{n=k, n \bmod k = i}^{\infty} \mathbb{E}[Y_n \mid \mathscr{F}_{n-k}]\right]^{p} = \mathbb{E}\left[\sum_{n=k}^{\infty} Y_n + \sum_{i=0}^{k-1} \sum_{n=k, n \bmod k = i}^{\infty} (\mathbb{E}[Y_n \mid \mathscr{F}_{n-k}] - Y_n)\right]^{p}
$$

$$
\overset{(a)}{\leq} \mathbb{E}\left[\frac{1}{2} + \sum_{i=0}^{k-1} \sum_{n=k, n \bmod k = i}^{\infty} (\mathbb{E}[Y_n \mid \mathscr{F}_{n-k}] - Y_n)\right]^{p} \overset{(b)}{\leq} 2^{p-1}\left(\frac{1}{2^p} + \mathbb{E}\left[\sum_{n=1}^{\infty} (\mathbb{E}[Y_n \mid \mathscr{F}_{n-1}] - Y_n)\right]^{p}\right)
$$

$$
\overset{(c)}{\leq} \frac{1}{2} + 2^{p-1} k^{p-1} C_p \sum_{i=0}^{k-1} \mathbb{E}\left[\sum_{n=k, n \bmod k = i}^{\infty} (\mathbb{E}[Y_n \mid \mathscr{F}_{n-k}] - Y_n)\right]^{p}
$$

$$
\overset{(d)}{\leq} \frac{1}{2} + 2^{p-1} k^{p-1} C_p \sum_{i=0}^{k-1} \mathbb{E}\left[\sum_{n=k, n \bmod k = i}^{\infty} (\mathbb{E}[Y_n \mid \mathscr{F}_{n-k}] - Y_n)^2\right]^{p/2}
$$

$$\overset{(e)}{\leq} \frac{1}{2} + 2^{p-1}k^{p-1}C_p \sum_{i=0}^{k-1} \mathbb{E}\left[\sum_{n=k,n \bmod k=i}^{\infty} |\mathbb{E}[Y_n|\mathscr{F}_{n-k}] - Y_n|\right]^{p/2}$$

$$\overset{(f)}{\leq} \frac{1}{2} + 2^{p-1}k^{p-1}C_p \mathbb{E}\left[\sum_{n=k}^{\infty} |\mathbb{E}[Y_n|\mathscr{F}_{n-k}] - Y_n|\right]^{p/2}$$

$$\leq \frac{1}{2} + 2^{p-2}k^{p-1}C_p + 2^{\frac{3}{2}p-2}k^{p-1}C_p \mathbb{E}\left[\sum_{n=1}^{\infty} \mathbb{E}[Y_n|\mathscr{F}_{n-1}]\right]^{p/2}$$

$$= \frac{1}{2} + 2^{p-2}k^{p-1}C_p + 2^{\frac{3}{2}p-2}k^{p-1}C_pM(p/2). \tag{17}$$

In the above derivation, Inequality $(a)$ is by noting that $\sum_{n=1}^{+\infty} Y_n = \frac{1}{2}$. Inequality $(b)$ uses the *AM-GM* inequality, specifically,

$$\left(\frac{a+b}{2}\right)^p \leq \frac{a^p + b^p}{2}.$$

Inequality $(c)$ involves using the *AM-GM* inequality for $k$ variables, specifically,

$$\left(\frac{a_1 + a_2 + ... + a_k}{k}\right)^p \leq \left(\frac{a_1^p + a_2^p + ... + a_k^p}{k}\right).$$

Inequality $(e)$ involves using *Burkholder's* inequality, where $C_p$ is a constant depending only on $p$, and its order with respect to $p$ is $\mathcal{O}(p)$. Inequality $(d)$ is by noting that

$$|\mathbb{E}[Y_n|\mathscr{F}_{n-k}] - Y_n|^2 \leq |\mathbb{E}[Y_n|\mathscr{F}_{n-k}] - Y_n|.$$

By repeatedly using Equation (17) and using the fact that $C_p = \mathcal{O}(p)$, we obtain the following estimate

$$M(p) = o((kp)^{\sqrt{p}}),$$

that is,

$$\mathbb{E}[(\Lambda^{(k)})^p] = o((2M)^p \cdot (kp)^{\sqrt{p}}).$$

(iii) For any $0 < l < 1$, we obtain:

$$\Lambda(l) = \sum_{k=1}^{+\infty} \mathbb{E}\left[\left(\sum_{t=k}^{+\infty} l^{t-k}X_t\right)\Big|\mathscr{F}_{k-1}\right] = \sum_{k=1}^{+\infty} \mathbb{E}\left[\left(\sum_{t=0}^{+\infty} l^t X_{k+t}\right)\Big|\mathscr{F}_{k-1}\right]$$

$$= \sum_{t=0}^{+\infty}\sum_{k=1}^{+\infty} \mathbb{E}\left[l^t X_{k+t}\Big|\mathscr{F}_{k-1}\right] = \sum_{t=0}^{+\infty} l^t \Lambda^{(t)}.$$

Next, we apply Hölder's inequality, we obtain $\forall n \geq 2$

$$\Lambda(l)^n = \left(\sum_{t=0}^{+\infty} l^t \Lambda^{(t)}\right)^n \leq \left(\frac{1}{1-l}\right)^{n-1} \sum_{t=0}^{+\infty} l^t (\Lambda^{(t)})^n.$$

Then we have:

$$\mathbb{E}[e^{p\Lambda(l)}] = \sum_{n=0}^{+\infty} \frac{p^n \mathbb{E}[\Lambda(l)^n]}{n!} \leq \sum_{n=0}^{+\infty} \left(\frac{p}{1-l}\right)^n \frac{\sum_{t=0}^{+\infty} l^t (\Lambda^{(t)})^n}{n!} = \sum_{n=0}^{+\infty}\sum_{t=0}^{+\infty} l^t (\Lambda^{(t)})^n \left(\frac{p}{1-l}\right)^n \frac{1}{n!}$$

$$\overset{(iii)}{=} \mathcal{O}\left(\sum_{n=0}^{+\infty}\sum_{t=0}^{+\infty} l^t (2M)^n \cdot (tp)^{\sqrt{n}} \left(\frac{p}{1-l}\right)^n \frac{1}{n!}\right)$$

$$= \mathcal{O}\left(\sum_{n=0}^{+\infty} \left(\sum_{t=0}^{+\infty} l^t t^{\sqrt{n}}\right) \left(\frac{p}{1-l}\right)^n \frac{(2M)^n \cdot (p)^{\sqrt{n}}}{n!}\right)$$

$$\overset{\text{Lemma C.4}}{=} \mathcal{O}\left(\sum_{n=0}^{+\infty} \Gamma(\sqrt{n}+1) \frac{1}{\left(\ln \frac{1}{l}\right)^{\sqrt{n}+1}} \left(\frac{p}{1-l}\right)^n \frac{(2M)^n \cdot (p)^{\sqrt{n}}}{n!}\right).$$

By substituting the factorial in the denominator with Stirling's approximation, it is evident that the series inside the $\mathcal{O}$ notation converges and depends only on $p$, $l$ and $M$. The lemma follows. $\qquad\square$

## D. Supporting Lemmas

This section introduces key lemmas that are essential for the proofs. We start with a diagram illustrating their relationships with the theorems. Rigorous proofs for all lemmas and theorems follow in the subsequent subsections. Due to its isolated, lengthy proof, Lemma D.2 is addressed separately at the end of the paper (see Section E for details).

### D.1. Dependency Graph of Lemmas and Theorems

Due to the large number of lemmas, we have combined these lemmas with those in the main text and theorems to create a lemma-theorem dependency graph. We refer the audience to this graph for a whole picture of our proofs, while the reader may also find the lemmas needed for a specific statement.

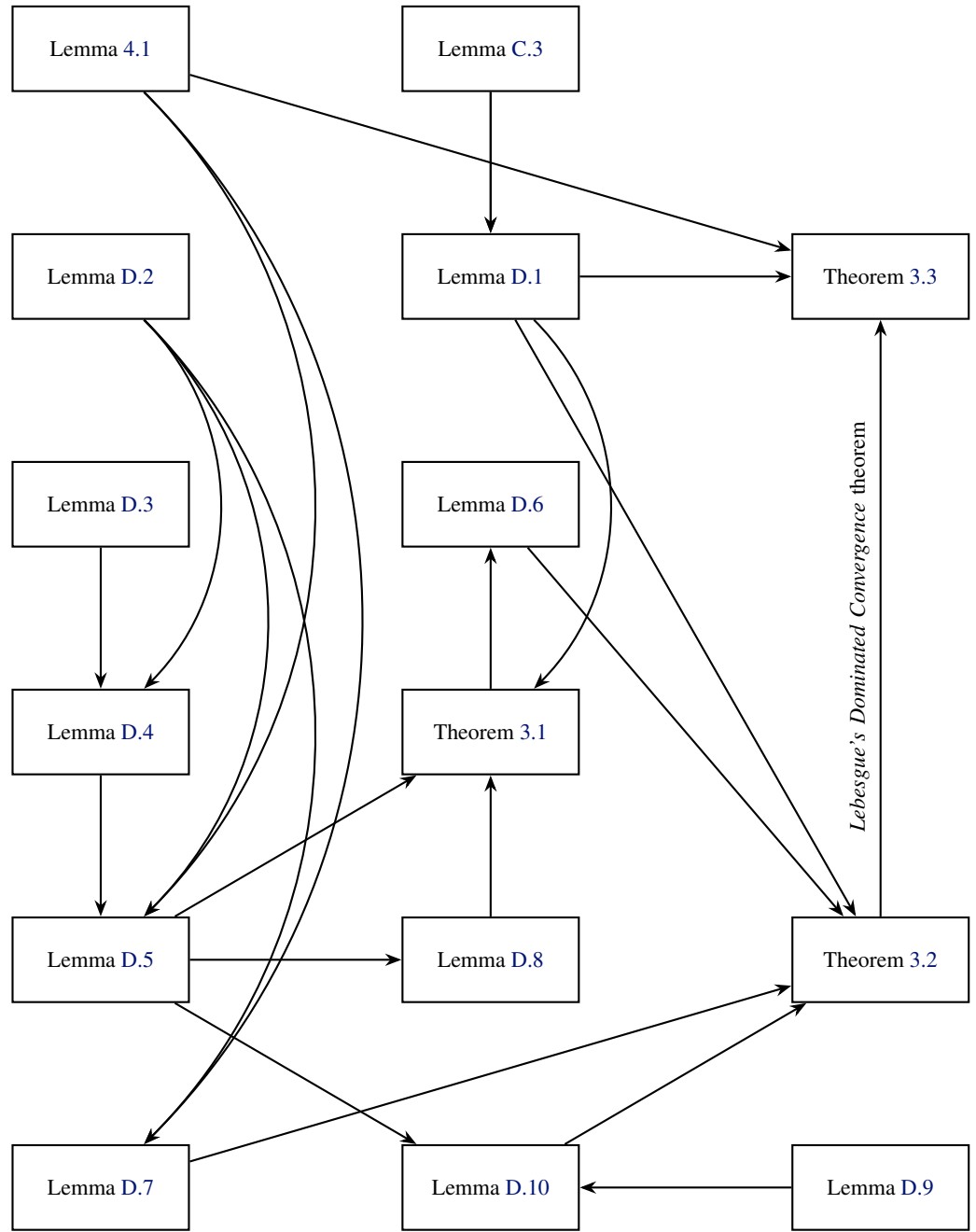

## D.2. Statements of the Lemmas

**Lemma D.1.** *For $\Pi_{\Delta,T}$ defined in Equation (5), for any $T \geq 0$ and any $p \geq 1$, the $p$-th moment of its reciprocal is bounded, i.e.,*

$$\mathbb{E}\left[\Pi_{\Delta,T}^{-p}\right] < C_{v,d,p} < +\infty,$$

*where $C_{v,d,p}$ is a constant that depends only on $v$, $d$, and $p$.*

*Moreover, we have that $\Pi_{\Delta,\infty}^{-1} := \lim_{t \to +\infty} \Pi_{\Delta,t}^{-1} < +\infty$ a.s., and for any $p \geq 1$, the $p$-th moment of $\Pi_{\Delta,\infty}^{-1}$ exists, with*

$$\mathbb{E}\left[\Pi_{\Delta,\infty}^{-p}\right] \leq C_{v,d,p} < +\infty.$$

**Lemma D.2.** *Consider the Adam algorithm in Algorithm 2.1 and suppose that Assumption 2.1 - 2.3 hold. Then for any initial point, and $T \geq 1$, the following results hold*

$$\mathbb{E}[\Pi_{\Delta,T}(f(w_T) - f^*)] = \mathcal{O}\left(\sum_{t=1}^{T}\mathbb{E}\left\|\eta_{v_t} \circ g_t\right\|^2\right) + \mathcal{O}(1),$$

$$\sum_{t=1}^{T}\mathbb{E}\left[\Pi_{\Delta,t}\sum_{i=1}^{d}\zeta_i(t)\right] = \mathcal{O}\left(\sum_{t=1}^{T}\mathbb{E}\left\|\eta_{v_t} \circ g_t\right\|^2\right) + \mathcal{O}(1),$$

$$\sum_{t=1}^{T}\sum_{i=1}^{d}\mathbb{E}\left[\Pi_{\Delta,t}\Delta_{t,i}|\nabla_i f(u_t)m_{t-1,i}|\right] = \mathcal{O}\left(\sum_{t=1}^{T}\mathbb{E}\left\|\eta_{v_t} \circ g_t\right\|^2\right) + \mathcal{O}(1).$$

*The specific form of the constants hidden behind the $\mathcal{O}()$ notation can be found in Equation (55) and Equation (56). All constants depend on the initial point and the constants in our required assumptions (excluding $1/\mu$).*

**Lemma D.3.** *Consider the Adam algorithm defined in Algorithm 2.1 and suppose that Assumption $2.1-2.3$ hold. Then for any initial point and $\forall \phi > 0$, we have for any $T \geq 1$, the following inequality holds*

$$\frac{\Pi_{\Delta,T}\sqrt{S_T}}{(T+1)^{\phi}} \leq \sqrt{dv} + \sum_{t=1}^{T}\Pi_{\Delta,t}\Lambda_{\phi,t}, \tag{18}$$

*where*

$$\Lambda_{\phi,t} := \frac{\|g_t\|^2}{(t+1)^{\phi}\sqrt{S_{t-1}}},$$

*and $S_T$ is defined in Remark 4.1.*

**Lemma D.4.** *Consider the Adam algorithm defined in Algorithm 2.1 and suppose that Assumptions 2.1 - 2.3 hold. Then for any initial point and for all $T \geq 1$, there exists a random variable $\zeta$ such that the following results hold*

(a) *$0 \leq \zeta < +\infty$ almost surely, and $\mathbb{E}(\zeta)$ is uniformly bounded above by a constant $C_{\zeta}$, which depends on the initial point and the constants in the required assumptions (excluding $1/\mu$). The explicit form of this upper bound is provided in Equation (29).*

(b) *$\sqrt{S_T} \leq (T+1)^4\Pi_{\Delta,\infty}^{-1}\zeta$, and $\ln\left(\frac{S_T}{v}\right) \leq \ln(T+1)\zeta'$, where $\zeta' \leq 4\left(1 + \frac{1}{2}\ln\left(\max\left\{e, \Pi_{\Delta,\infty}^{-1}\zeta\right\}\right)\right).$*

**Lemma D.5.** *Consider the Adam algorithm defined in Algorithm 2.1 and suppose that Assumption 2.1 - 2.3 hold. Then for any initial point and $T \geq 1$, the following results hold*

$$\sum_{t=1}^{T}\mathbb{E}\left[\Pi_{\Delta,t}\sum_{i=1}^{d}\zeta_i(t)\right] \leq \begin{cases} C_{4,\delta}, & \text{if } \delta \in (0,1/2), \\ C_5 + C_6\,\mathbb{E}\left[\ln(S_T)\right], & \text{if } \delta = 0, \end{cases}$$

*where $C_5$ and $C_6$ are constants that depend on the initial point and the constants in our required assumptions (excluding $1/\mu$), and $C_{4,\delta}$ is a constant that depends on the initial point, $\delta$, and the constants in our required assumptions (excluding $1/\mu$).*

**Lemma D.6** (**Subsequence Convergence**). *Under Assumptions 2.1 - 2.3, consider the Adam algorithm (Algorithm 2.1) with hyperparameters as specified in Subsection 2.4, where $\delta > 0$. Then, there exists a subsequence $\{w_{c_t}\}_{t\geq 1}$ such that its gradients converge to zero almost surely, i.e., $\lim_{t\to\infty}\|\nabla f(w_{c_t})\| = 0$ a.s.*

**Lemma D.7.** *Consider the Adam algorithm defined in Algorithm 2.1 and assume that Assumptions 2.1 - 2.3 hold. Then, for any initial point and for all $T \geq 1$, the following results hold*

*- When $\delta = 0$, we have*

$$\sup_{t\geq 1}\frac{\Pi_{\Delta,t}(f(w_t) - f^*)}{\ln^2(t+1)} < +\infty \ \ a.s., \ \ \sup_{T\geq 1}\mathbb{E}\left[\frac{\Pi_{\Delta,t}(f(w_t) - f^*)}{\ln^2(t+1)}\right] < M_0 < +\infty.$$

- *When $\delta > 0$, we have*

$$\sup_{t \geq 1} \Pi_{\Delta,t}(f(w_t) - f^*) < +\infty \ \ a.s., \ \ \sup_{T \geq 1} \mathbb{E}\left[\Pi_{\Delta,t}(f(w_t) - f^*)\right] < M_\delta < +\infty.$$

*In the above equations, $M_0$ and $M_\delta$ are two constants that depend on the initial point and the constants in our assumptions (excluding $1/\mu$).*

**Lemma D.8.** *Consider the Adam algorithm defined in Algorithm 2.1 and suppose that Assumption 2.1 - 2.3 hold. Then for any initial point, for all $T \geq 1$, $i \in [1, d]$, there is*

$$\mathbb{E}(S_T^{3/4}) = \begin{cases} \mathcal{O}(T^{3/4}), & \text{if } \delta \in (0, 1/2), \\ \mathcal{O}(T^{3/4} \ln^{3/2} T), & \text{if } \delta = 0, \end{cases}$$

*where constants hidden in $\mathcal{O}()$ depend on the initial point and the constants in our required assumptions (excluding $1/\mu$).*

**Lemma D.9.** *Under Assumptions 2.1 - 2.3, consider the Adam algorithm (Algorithm 2.1) with the hyperparameters specified in Subsection 2.4. Then, for any $t \geq 1$, the following inequality holds*

$$\sup_{t \geq 1} \mathbb{E}[\Pi_{\Delta,t} \Sigma_{v_t}] < \begin{cases} \mathbb{E}[\Pi_{\Delta,2} \Sigma_{v_1}] + (A + 2L_f B) M_\delta + C, & \text{if } \delta \in (0, 1/2), \\ \mathbb{E}[\Pi_{\Delta,2} \Sigma_{v_1}] + (A + 2L_f B) M_0 \ln^2 t + C, & \text{if } \delta = 0. \end{cases}$$

*Furthermore, if $\lambda > 1$, then*

$$\sup_{t \geq 1} \mathbb{E}[\Pi_{\Delta,t} \Sigma_{v_t}] < \begin{cases} ((A + 2L_f B) M_\delta + C) \sum_{t=1}^{+\infty} \frac{1}{(t+1)^\lambda}, & \text{if } \delta \in (0, 1/2) \\ ((A + 2L_f B) M_0 + C) \sum_{t=1}^{+\infty} \frac{\ln^2 t}{(t+1)^\lambda}, & \text{if } \delta = 0 \end{cases} < +\infty.$$

*Additionally,*

$$\sup_{t \geq 1} \Sigma_{v_t} < +\infty \ \ a.s.$$

**Lemma D.10.** *Under Assumption 2.1 - 2.3, consider the Adam algorithm (Algorithm 2.1) with hyperparameters in Subsection 2.4 with $\gamma > 1$, $\delta > 0$. Then for any initial point, the following results hold*

$$\sum_{t=1}^{+\infty} \eta_t \|\nabla f(w_t)\|^2 < +\infty \ \ a.s., \ \ \sum_{t=1}^{+\infty} \eta_t \|\nabla f(u_t)\|^2 < +\infty \ \ a.s., \ \ and \ \ \sum_{t=1}^{+\infty} \|\eta_{v_t} \circ m_t\|^2 < +\infty \ \ a.s.$$

## D.3. Proofs of the Lemmas and the Theorems

### D.3.1. PROOF OF LEMMA D.1

*Proof.* Consider the case where $T$ is finite. For any $T \in (0, +\infty)$, the exponential-logarithmic transformation can be applied to $\Pi_{\Delta,T}^{-p}$, resulting in

$$\Pi_{\Delta,T}^{-p} = \exp\left\{ p \sum_{k=1}^{T} \ln\left( 1 + \left( \frac{D_1}{1 - \sqrt{\beta_1}} + 1 \right) \overline{\Delta}_{\sqrt{\beta_1},k} \right) \right\}$$

$$\overset{\ln(1+x) < x \ \forall \ x > -1}{\leq} \exp\left\{ p \left( \frac{D_1}{1 - \sqrt{\beta_1}} + 1 \right) \sum_{k=1}^{T} \overline{\Delta}_{\sqrt{\beta_1},k} \right\}$$

$$= \exp\left\{ p \left( \frac{D_1}{1 - \sqrt{\beta_1}} + 1 \right) \sum_{k=1}^{T} \sum_{i=1}^{d} \mathbb{E}\left[ \sum_{t=k}^{+\infty} (\sqrt{\beta_1})^{t-k} \Delta_{t,i} \middle| \mathscr{F}_{k-1} \right] \right\}$$

$$= \exp\left\{ p \left( \frac{D_1}{1 - \sqrt{\beta_1}} + 1 \right) \sum_{k=1}^{T} \mathbb{E}\left[ \sum_{t=k}^{+\infty} (\sqrt{\beta_1})^{t-k} \left( \sum_{i=1}^{d} \Delta_{t,i} \right) \middle| \mathscr{F}_{k-1} \right] \right\}$$

$$\leq \exp\left\{ p \left( \frac{D_1}{1 - \sqrt{\beta_1}} + 1 \right) \sum_{k=1}^{+\infty} \mathbb{E}\left[ \sum_{t=k}^{+\infty} (\sqrt{\beta_1})^{t-k} \left( \sum_{i=1}^{d} \Delta_{t,i} \right) \middle| \mathscr{F}_{k-1} \right] \right\}$$

It can be readily verified that $\sum_{i=1}^{d} \Delta_{t,i}$ in the inequality above satisfies all the properties of $X_t$ outlined in Lemma C.5. Consequently, by Lemma C.5, we deduce that for any $0 < T < +\infty$ and any $p \geq 1$, the $p$-th moment of its reciprocal is bounded, i.e.,

$$\mathbb{E}\left[\Pi_{\Delta,T}^{-p}\right] < C_{v,d,p} < +\infty,$$

where $C_{v,d,p}$ is a constant depending only on $v$, $d$, and $p$. Letting $T \to +\infty$ and applying the *Lebesgue's Monotone Convergence Theorem*, we obtain

$$\mathbb{E}\left[\Pi_{\Delta,\infty}^{-p}\right] \leq C_{v,d,p} < +\infty.$$

$\square$

### D.3.2. PROOFS OF LEMMA 4.1

*Proof.* By the $L$-smoothness in Assumption 2.2, we have:

$$f(u_{t+1}) - f(u_t) \leq \nabla f(u_t)^\top (u_{t+1} - u_t) + \frac{L_f}{2}\|u_{t+1} - u_t\|^2.$$

Then, by substituting the iterative formula for $u_t$ from Equation (2) into the above inequality, we obtain

$$
\begin{aligned}
f(u_{t+1}) - f(u_t) \leq & -\sum_{i=1}^{d} \eta_{v_t,i}\nabla_i f(u_t)g_{t,i} + \frac{\beta_1}{1-\beta_1}\sum_{i=1}^{d}\Delta_{t,i}\nabla_i f(u_t)m_{t-1,i} + L_f\sum_{i=1}^{d}\eta_{v_t,i}^2 g_{t,i}^2 \\
& + L_f\left(\frac{\beta_1}{1-\beta_1}\right)^2\sum_{i=1}^{d}\Delta_{t,i}^2 m_{t-1,i}^2 \\
\overset{(a)}{=} & \underbrace{-\sum_{i=1}^{d}\eta_{v_t,i}\nabla_i f(w_t)g_{t,i}}_{\Theta_{t,1}} + \underbrace{\sum_{i=1}^{d}(\eta_{v_t,i}(\nabla_i f(w_t) - \nabla_i f(u_t))g_{t,i})}_{\Theta_{t,2}} \\
& + \frac{\beta_1}{1-\beta_1}\underbrace{\sum_{i=1}^{d}\Delta_{t,i}\nabla_i f(u_t)m_{t-1,i}}_{\Theta_{t,3}} + L_f\sum_{i=1}^{d}\eta_{v_t,i}^2 g_{t,i}^2 \\
& + L_f\left(\frac{\beta_1}{1-\beta_1}\right)^2\underbrace{\sum_{i=1}^{d}\Delta_{t,i}^2 m_{t-1,i}^2}_{\Theta_{t,4}}.
\end{aligned}
\tag{19}
$$

Step $(a)$ employs the identity $\nabla_i f(u_t) = \nabla_i f(w_t) + \nabla_i f(u_t) - \nabla_i f(w_t)$. Next, we handle $\Theta_{t,1}$, $\Theta_{t,2}$, $\Theta_{t,3}$ and $\Theta_{t,4}$ separately. First, for $\Theta_{t,1}$, we use the following identity.

$$
\begin{aligned}
\Theta_{t,1} = -\sum_{i=1}^{d}\eta_{v_t,i}\nabla_i f(w_t)g_{t,i} = & -\sum_{i=1}^{d}\eta_{v_{t-1},i}\nabla_i f(w_t)g_{t,i} + \sum_{i=1}^{d}\Delta_{t,i}\nabla_i f(w_t)g_{t,i} \\
= & -\sum_{i=1}^{d}\underbrace{\eta_{v_{t-1},i}(\nabla_i f(w_t))^2}_{\zeta_i(t)} + \underbrace{\sum_{i=1}^{d}\Delta_{t,i}\nabla_i f(w_t)g_{t,i}}_{\Theta_{t,1,1}} + \underbrace{\sum_{i=1}^{d}\eta_{v_{t-1},i}\nabla_i f(w_t)(\nabla_i f(w_t) - g_{t,i})}_{M_{t,1}},
\end{aligned}
\tag{20}
$$

where $\Delta_{t,i}$ in the above equality represents the $i$-th component of the vector $\Delta_t$, which is defined in Equation (2). In this way, we decompose $\Theta_1$ into a descent term $-\sum_{i=1}^{d}\zeta_i(t)$, an error term $\Theta_{t,1,1}$, and a martingale difference term $M_{t,1}$. We will further scale and control the error term $\Theta_{t,1,1}$. Specifically, we have

$$\Theta_{t,1,1} = \sum_{i=1}^{d}\mathbb{E}\left[\Delta_{t,i}\nabla_i f(w_t)g_{t,i} \mid \mathscr{F}_{t-1}\right]$$

$$+ \sum_{i=1}^{d} \left( \Delta_{t,i} \nabla_i f(w_t) g_{t,i} - \mathbb{E} \left[ \Delta_{t,i} \nabla_i f(w_t) g_{t,i} \mid \mathscr{F}_{t-1} \right] \right)$$
$$\underbrace{\phantom{\sum_{i=1}^{d} \left( \Delta_{t,i} \nabla_i f(w_t) g_{t,i} - \mathbb{E} \left[ \Delta_{t,i} \nabla_i f(w_t) g_{t,i} \mid \mathscr{F}_{t-1} \right] \right)}}_{M_{t,2}}$$

$$\overset{(a)}{<} \sum_{i=1}^{d} \sqrt{\eta_{v_{t-1},i}} \nabla_i f(w_t) \, \mathbb{E} \left[ \sqrt{\Delta_{t,i}} g_{t,i} \mid \mathscr{F}_{t-1} \right] + M_{t,2}$$

$$\overset{(b)}{\leq} \frac{1}{2} \sum_{i=1}^{d} \eta_{v_{t-1},i} (\nabla_i f(w_t))^2 + \frac{1}{2} \sum_{i=1}^{d} \mathbb{E}^2 \left[ \sqrt{\Delta_{t,i}} g_{t,i} \mid \mathscr{F}_{t-1} \right] + M_{t,2}$$

$$\overset{(c)}{\leq} \frac{1}{2} \sum_{i=1}^{d} \zeta_i(t) + \frac{1}{2} \sum_{i=1}^{d} \mathbb{E}[g_{t,i}^2 \mid \mathscr{F}_{t-1}] \cdot \mathbb{E}[\Delta_{t,i} \mid \mathscr{F}_{t-1}] + M_{t,2}$$

$$\leq \frac{1}{2} \sum_{i=1}^{d} \zeta_i(t) + \frac{1}{2} \sum_{i=1}^{d} \mathbb{E}[g_{t,i}^2 \mid \mathscr{F}_{t-1}] \cdot \mathbb{E}[\Delta_{t,i} \mid \mathscr{F}_{t-1}] + M_{t,2}$$

$$\leq \frac{1}{2} \sum_{i=1}^{d} \zeta_i(t) + \frac{1}{2} \left( \sum_{i=1}^{d} \mathbb{E}[g_{t,i}^2 \mid \mathscr{F}_{t-1}] \right) \cdot \left( \sum_{i=1}^{d} \mathbb{E}[\Delta_{t,i} \mid \mathscr{F}_{t-1}] \right) + M_{t,2}$$

$$\overset{(d)}{\leq} \frac{1}{2} \sum_{i=1}^{d} \zeta_i(t) + \frac{1}{2} \left( (A + 2L_f B) f(w_t) + C \right) \cdot \left( \sum_{i=1}^{d} \mathbb{E}[\Delta_{t,i} \mid \mathscr{F}_{t-1}] \right) + M_{t,2}$$

$$= \frac{1}{2} \sum_{i=1}^{d} \zeta_i(t) + \frac{1}{2} (A + 2L_f B) f(w_t) \cdot \left( \sum_{i=1}^{d} \mathbb{E}[\Delta_{t,i} \mid \mathscr{F}_{t-1}] \right)$$
$$+ C \left( \sum_{i=1}^{d} \mathbb{E}[\Delta_{t,i} \mid \mathscr{F}_{t-1}] \right) + M_{t,2}$$

$$= \frac{1}{2} \sum_{i=1}^{d} \zeta_i(t) + \frac{1}{2} (A + 2L_f B) f(w_t) \cdot \left( \underbrace{\sum_{i=1}^{d} \mathbb{E}[\Delta_{t,i} \mid \mathscr{F}_{t-1}]}_{\overline{\Delta}_t} \right) + C \sum_{i=1}^{d} \Delta_{t,i}$$

$$+ \underbrace{C \left( \sum_{i=1}^{d} \left( \mathbb{E}[\Delta_{t,i} \mid \mathscr{F}_{t-1}] - \Delta_{t,i} \right) \right)}_{M_{t,3}} + M_{t,2}. \tag{21}$$

In the above derivation, in step $(a)$, we utilized the property of conditional expectation, which states that for random variables $X \in \mathscr{F}_{n-1}$ and $Y \in \mathscr{F}_n$, we have $\mathbb{E}[XY|\mathscr{F}_{n-1}] = X \, \mathbb{E}[Y|\mathscr{F}_{n-1}]$. Additionally, note that $\Delta_{t,i} = \sqrt{\Delta_{t,i}} \sqrt{\Delta_{t,i}} < \sqrt{\eta_{v_{t-1}}} \sqrt{\Delta_{t,i}}$ (due to Property 2, we know $\Delta_{t,i} \geq 0$, so taking the square root is well-defined). In step $(b)$, we employed the *AM-GM* inequality, which states $ab \leq \frac{a^2+b^2}{2}$. In step $(c)$, we used the *Cauchy-Schwarz* inequality for conditional expectations, namely $\mathbb{E}[XY|\mathscr{F}_{n-1}] \leq \sqrt{\mathbb{E}[X^2|\mathscr{F}_{n-1}] \mathbb{E}[Y^2|\mathscr{F}_{n-1}]}$. For step $(d)$, we used Property 1. Specifically, we have

$$\sum_{i=1}^{d} \mathbb{E}[g_{t,i}^2|\mathscr{F}_{t-1}] = \mathbb{E}[\|g_t\|^2|\mathscr{F}_{t-1}] \leq (A + 2L_f B) f(w_t) + C.$$

Substituting the estimate of $\Theta_{t,1,1}$ into Equation (21), we obtain

$$\Theta_{t,1} = -\frac{1}{2} \sum_{i=1}^{d} \zeta_i(t) + \frac{A + 2L_f B}{2} \overline{\Delta}_t \cdot f(w_t) + C \sum_{i=1}^{d} \Delta_{t,i} + \underbrace{M_{t,1} + M_{t,2} + M_{t,3}}_{M_t}.$$

Then, we use Property 5 to replace $f(w_t)$ with $f(u_t)$ to obtain

$$
\begin{aligned}
\Theta_{t,1} = &-\frac{1}{2}\sum_{i=1}^{d}\zeta_i(t) + \frac{(A + 2L_f B)(L_f + 1)}{2}\overline{\Delta}_t \cdot f(u_t) + C\sum_{i=1}^{d}\Delta_{t,i} \\
&+ \frac{(L_f + 1)\beta_1^2}{2(1 - \beta_1)^2}\|\eta_{v_{t-1}} \circ m_{t-1}\|^2 + M_t.
\end{aligned}
\tag{22}
$$

Next, we handle the term $\Theta_{t,2}$ through the following derivation.

$$
\begin{aligned}
\Theta_{t,2} &= \frac{1}{2}\sum_{i=1}^{d}(\eta_{v_t,i}(\nabla_i f(w_t) - \nabla_i f(u_t))g_{t,i}) \\
&\leq \sum_{i=1}^{d}\eta_{v_t,i}^2 g_{t,i}^2 + \frac{1}{2}\sum_{i=1}^{d}(\nabla_i f(w_t) - \nabla_i f(u_t))^2 \\
&= \sum_{i=1}^{d}\eta_{v_t,i}^2 g_{t,i}^2 + \frac{1}{2}\|\nabla f(w_t) - \nabla f(u_t)\|^2 \\
&\leq \sum_{i=1}^{d}\eta_{v_t,i}^2 g_{t,i}^2 + \frac{L_f^2}{2}\|w_t - u_t\|^2 \\
&= \sum_{i=1}^{d}\eta_{v_t,i}^2 g_{t,i}^2 + \frac{\beta_1^2 L_f^2}{2(1 - \beta_1)^2}\|\eta_{v_{t-1}} \circ m_{t-1}\|^2.
\end{aligned}
\tag{23}
$$

Next, we handle the term $\Theta_{t,3}$. We have

$$
\Theta_{t,3} = \sum_{i=1}^{d}\Delta_{t,i}\nabla_i f(u_t)m_{t-1,i} \leq \sum_{i=1}^{d}\Delta_{t,i}|\nabla_i f(u_t)m_{t-1,i}|.
\tag{24}
$$

For $\Theta_{t,4}$, because $\Delta_{t,i} \leq \eta_{v_{t-1},i}$, we obtain that

$$
\Theta_{t,4} = \sum_{i=1}^{d}\Delta_{t,i}^2 m_{t-1,i}^2 < \sum_{i=1}^{d}\eta_{v_{t-1},i}^2 m_{t-1,i}^2 = \|\eta_{v_{t-1}} \circ m_{t-1}\|^2.
\tag{25}
$$

Finally, substituting the estimates of $\Theta_{t,1}$ from Equation (22), $\Theta_{t,2}$ from Equation (23), $\Theta_{t,3}$ from Equation (24), and $\Theta_{t,4}$ from Equation (25) back into Equation (19), we obtain

$$
\underbrace{\left(f(u_{t+1}) - f^* + C\sum_{i=1}^{d}\eta_{v_t,i}\right)}_{\hat{f}(u_{t+1})} - \underbrace{\left(f(u_t) - f^* + C\sum_{i=1}^{d}\eta_{v_{t-1},i}\right)}_{\hat{f}(u_t)} \leq -\frac{1}{2}\sum_{i=1}^{d}\zeta_i(t) + C_1\overline{\Delta}_t \cdot f(u_t)
$$

$$
+ C_2\|\eta_{v_{t-1}} \circ m_{t-1}\|^2 + \sum_{i=1}^{d}\Delta_{t,i}|\nabla_i f(u_t)m_{t-1,i}| + (L_f + 1)\sum_{i=1}^{d}\eta_{v_t,i}^2 g_{t,i}^2 + M_t,
$$

where

$$
C_1 := \frac{(A + 2L_f B)(L_f + 1)}{2}, \quad C_2 := \frac{\beta_1^2 L_f^2}{2(1 - \beta_1)^2} + L_f\left(\frac{\beta_1}{1 - \beta_1}\right)^2.
\tag{26}
$$

To the second term on the right side of the above inequality, we apply the inequality $[f(u_t) < f(u_t) - f^* + C\sum_{i=1}^{d}\eta_{v_{t-1},i} = \hat{f}(u_t)$ and then move the expanded term to the left side of the inequality. This obtains

$$
\hat{f}(u_{t+1}) - (1 + C_1\overline{\Delta}_t)\hat{f}(u_t) \leq -\frac{1}{2}\sum_{i=1}^{d}\zeta_i(t) + C_1\overline{\Delta}_t \cdot f(u_t)
$$

$$+ C_2 \|\eta_{v_{t-1}} \circ m_{t-1}\|^2 + \sum_{i=1}^{d} \Delta_{t,i} |\nabla_i f(u_t) m_{t-1,i}|$$

$$+ (L_f + 1) \sum_{i=1}^{d} \eta_{v_t,i}^2 g_{t,i}^2 + M_t.$$

Next, we define

$$\overline{\Delta}_{\sqrt{\beta_1},k} := \sum_{i=1}^{d} \mathbb{E}\left[\sum_{t=k}^{+\infty} (\sqrt{\beta_1})^{t-k} \Delta_{t,i} \,\middle|\, \mathscr{F}_{k-1}\right].$$

Observe that

$$1 + C_1 \overline{\Delta}_t \leq 1 + \left(\frac{D_1}{1 - \sqrt{\beta_1}} + 1\right) \overline{\Delta}_{\sqrt{\beta_1},t},$$

where $D_1$ is defined in Lemma E.2. Thus, we have

$$\hat{f}(u_{t+1}) - \left(1 + \left(\frac{D_1}{1 - \sqrt{\beta_1}} + 1\right) \overline{\Delta}_{\sqrt{\beta_1},t}\right) \hat{f}(u_t) \leq -\frac{1}{2} \sum_{i=1}^{d} \zeta_i(t) + C_1 \overline{\Delta}_t \cdot f(u_t)$$

$$+ C_2 \|\eta_{v_{t-1}} \circ m_{t-1}\|^2 + \sum_{i=1}^{d} \Delta_{t,i} |\nabla_i f(u_t) m_{t-1,i}|$$

$$+ (L_f + 1) \sum_{i=1}^{d} \eta_{v_t,i}^2 g_{t,i}^2 + M_t.$$

Next, we construct an auxiliary variable

$$\Pi_{\Delta,t} := \prod_{k=1}^{t} \left(1 + \left(\frac{D_1}{1 - \sqrt{\beta_1}} + 1\right) \overline{\Delta}_{\sqrt{\beta_1},k}\right)^{-1} \quad (t \geq 1), \ \Pi_{\Delta,0} := 1.$$

Multiplying both sides of the above inequality by $\Pi_{\Delta,t}$, we obtain

$$\Pi_{\Delta,t} \hat{f}(u_{t+1}) - \Pi_{\Delta,t-1} \hat{f}(u_t) \leq -\frac{1}{2} \Pi_{\Delta,t} \sum_{i=1}^{d} \zeta_i(t) + C_2 \|\eta_{v_{t-1}} \circ m_{t-1}\|^2 + \sum_{i=1}^{d} \Delta_{t,i} |\nabla_i f(u_t) m_{t-1,i}|$$

$$+ (L_f + 1) \sum_{i=1}^{d} \eta_{v_t,i}^2 g_{t,i}^2 + \Pi_{\Delta,t} M_t.$$

With the inequality, we complete the proof. $\qquad \square$

### D.3.3. PROOF OF LEMMA D.3

*Proof.* For any $\phi \in \mathbb{R}$, we consider $\frac{\sqrt{S_T}}{(T+1)^\phi}$. We have

$$\frac{\sqrt{S_T}}{(T+1)^\phi} = \frac{S_T}{(T+1)^\phi \sqrt{S_T}} = \frac{S_0 + \sum_{t=1}^{T} \|g_t\|^2}{(T+1)^\phi \sqrt{S_T}} = \frac{S_0}{(T+1)^\phi \sqrt{S_T}} + \sum_{t=1}^{T} \frac{\|g_t\|^2}{(T+1)^\phi \sqrt{S_T}}$$

$$\leq \frac{S_0}{(T+1)^\phi \sqrt{S_T}} + \sum_{t=1}^{T} \frac{\|g_t\|^2}{(T+1)^\phi \sqrt{S_T}} \leq \sqrt{S_0} + \sum_{t=1}^{T} \frac{\|g_t\|^2}{(t+1)^\phi \sqrt{S_{t-1}}}$$

$$= \sqrt{dv} + \sum_{t=1}^{T} \frac{\|g_t\|^2}{(t+1)^\phi \sqrt{S_{t-1}}}.$$

By multiplying both sides of the above inequality by $\Pi_{\Delta,T}$ and noting the monotonicity of $\{\Pi_{\Delta,t}\}_{t \geq 1}$ as well as the fact that $\Pi_{\Delta,T} \leq 1$ for all $T \geq 1$, the lemma follows. $\qquad \square$

### D.3.4. PROOF OF LEMMA D.4

*Proof.* We take $\phi = 4$ in Lemma D.3 and bound the expectation of the partial sum $\sum_{t=1}^{T} \Lambda_{4,t}$. We have

$$
\mathbb{E}\left[\sum_{t=1}^{T} \Pi_{\Delta,t}\Lambda_{4,t}\right] = \sum_{t=1}^{T} \mathbb{E}[\Pi_{\Delta,t}\Lambda_{4,t}] = \sum_{t=1}^{T} \mathbb{E}\left[\frac{\Pi_{\Delta,t}\|g_t\|^2}{(t+1)^4\sqrt{S_{t-1}}}\right] = \sum_{t=1}^{T} \mathbb{E}\left[\frac{\Pi_{\Delta,t}\mathbb{E}[\|g_t\|^2|\mathscr{F}_{t-1}]}{(t+1)^4\sqrt{S_{t-1}}}\right]
$$

$$
\overset{\text{Property } 1}{\leq} \sum_{t=1}^{T} \mathbb{E}\left[\frac{(A+2L_fB)\Pi_{\Delta,t}(f(w_t)-f^*)+C}{(t+1)^4\sqrt{S_{t-1}}}\right]
$$

$$
\leq C_3 \sum_{t=1}^{T} \frac{1}{(t+1)^4}\,\mathbb{E}\left[\Pi_{\Delta,t}(f(w_t)-f^*)\right] + C_4 \sum_{t=1}^{T} \frac{1}{(t+1)^4}, \tag{27}
$$

where

$$
C_3 := \frac{A+2L_fB}{\sqrt{S_0}},\quad C_4 := \frac{C}{\sqrt{S_0}}.
$$

Based on the results in Lemma D.2, we can compute:

$$
\mathbb{E}\left[\Pi_{\Delta,t}(f(w_t)-f^*)\right] = \mathcal{O}\left(\sum_{k=1}^{t} \mathbb{E}\|\eta_{v_k} \circ g_k\|^2\right) + \mathcal{O}(1) = \mathcal{O}(t).
$$

Substitute the above result into Equation (27), and combine $\forall\, p \geq 2$

$$
\sum_{t=1}^{T} \frac{1}{(t+1)^p} \leq \sum_{t=1}^{T} \frac{1}{(t+1)^2} \leq \sum_{t=1}^{+\infty} \frac{1}{t^2} = \frac{\pi^2}{6}.
$$

We get

$$
\mathbb{E}\left[\sum_{t=1}^{T} \Pi_{\Delta,t}\Lambda_{4,t}\right] = \mathcal{O}(1).
$$

It can be observed that the right-hand side of the above inequality is independent of $T$. Thus, according to the *Lebesgue's Monotone Convergence* theorem, we have

$$
\sum_{t=1}^{T} \Pi_{\Delta,t}\Lambda_{4,t} \to \sum_{t=1}^{+\infty} \Pi_{\Delta,t}\Lambda_{4,t} \quad \text{a.s.,}
$$

and

$$
\mathbb{E}\left[\sum_{t=1}^{+\infty} \Pi_{\Delta,t}\Lambda_{4,t}\right] = \lim_{T\to\infty} \mathbb{E}\left[\sum_{t=1}^{T} \Pi_{\Delta,t}\Lambda_{4,t}\right] = \lim_{T\to\infty} \sum_{t=1}^{T} \mathbb{E}[\Pi_{\Delta,t}\Lambda_{4,t}] = \mathcal{O}(1).
$$

By setting

$$
\zeta := \sqrt{dv} + \sum_{t=1}^{+\infty} \Pi_{\Delta,t}\Lambda_{4,t},
$$

and combining Lemma D.3, we have

$$
\sqrt{S_T} \leq \Pi_{\Delta,T}^{-1}(T+1)^4\zeta < \Pi_{\Delta,\infty}^{-1}(T+1)^4\zeta. \tag{28}
$$

Meanwhile,

$$
\mathbb{E}[\zeta] = \sqrt{dv} + \mathbb{E}\left[\sum_{t=1}^{+\infty} \Lambda_{4,t}\right] = \mathcal{O}(1). \tag{29}
$$

Then through Equation (28), we have

$$
\frac{1}{2}\ln\left(\frac{S_T}{v}\right) \leq 4\ln(T+1) + \ln\left(\Pi_{\Delta,\infty}^{-1}\zeta\right)
$$

$$\leq 4\ln(T+1) + \ln\left(\max\left\{e, \Pi_{\Delta,\infty}^{-1}\zeta\right\}\right)$$

$$\leq 4\ln(T+1)\left(1 + \frac{\ln\left(\max\left\{e, \Pi_{\Delta,\infty}^{-1}\zeta\right\}\right)}{4\ln(T+1)}\right)$$

$$\overset{\ln(T+1)\geq 1/2}{\leq} 4\ln(T+1)\left(1 + \frac{1}{2}\ln\left(\max\left\{e, \Pi_{\Delta,\infty}^{-1}\zeta\right\}\right)\right).$$

The lemma follows □

### D.3.5. PROOF OF LEMMA D.5

*Proof.* According to the second conclusion of Lemma D.2, we have

$$\sum_{t=1}^{T}\mathbb{E}\left[\Pi_{\Delta,t}\sum_{i=1}^{d}\zeta_i(t)\right] = \mathcal{O}\left(\sum_{t=1}^{T}\mathbb{E}\|\eta_{v_t}\circ g_t\|^2\right) + \mathcal{O}(1).$$

To prove the conclusion of this lemma, we only need to bound $\sum_{t=1}^{T}\mathbb{E}\|\eta_{v_t}\circ g_t\|^2$. Specifically,

$$\mathbb{E}\|\eta_{v_t}\circ g_t\|^2 = \mathbb{E}\left[\sum_{i=1}^{d}\eta_{v_t,i}^2 g_{t,i}^2\right] = \mathbb{E}\left[\sum_{i=1}^{d}\frac{\eta_t^2 g_{t,i}^2}{(\sqrt{v_{t,i}}+\mu)^2}\right] \leq \mathbb{E}\left[\sum_{i=1}^{d}\frac{1}{t^{2\delta}}\frac{g_{t,i}^2}{tv_{t,i}}\right]$$

$$\overset{\text{Property 3}}{\leq} \frac{(t+1)^{2\delta}}{t^{2\delta}}\mathbb{E}\left[\sum_{i=1}^{d}\frac{1}{\alpha_1(t+1)^{2\delta}}\frac{g_{t,i}^2}{S_{t,i}}\right] \tag{30}$$

$$\overset{(a)}{\leq} \frac{2^{2\delta}}{\alpha_1}\zeta^{\frac{\delta}{2}}\Pi_{\Delta,\infty}^{-\frac{\delta}{2}}\sum_{i=1}^{d}\frac{g_{t,i}^2}{S_{t,i}^{1+\frac{\delta}{4}}}. \tag{31}$$

In step $(a)$ of the above derivation, we apply Lemma D.4 to $(t+1)^{2\delta}$. Specifically, according to Lemma D.4, we have

$$\sqrt{S_t} \leq \Pi_{\Delta,\infty}^{-1}(t+1)^4\zeta.$$

Next, with the estimate for $\mathbb{E}\|\eta_{v_t}\circ g_t\|^2$, we can estimate $\sum_{t=1}^{T}\mathbb{E}\|\eta_{v_t}\circ g_t\|^2$. To achieve this, note that

$$\sum_{t=1}^{T}\mathbb{E}\|\eta_{v_t}\circ g_t\|^2 = \mathbb{E}\left[\frac{2^{2\delta}}{\alpha_1}\zeta^{\frac{\delta}{2}}\Pi_{\Delta,\infty}^{-\frac{\delta}{2}}\sum_{i=1}^{d}\sum_{t=1}^{T}\frac{g_{t,i}^2}{S_{t,i}^{1+\frac{\delta}{4}}}\right] \leq \mathbb{E}\left[\frac{2^{2\delta}}{\alpha_1}\zeta^{\frac{\delta}{2}}\Pi_{\Delta,\infty}^{-\frac{\delta}{2}}\sum_{i=1}^{d}\int_{S_{0,i}}^{S_{T,i}}\frac{1}{x^{1+\frac{\delta}{4}}}\mathrm{d}x\right]$$

$$\leq \begin{cases} \frac{2^{2\delta}}{\alpha_1}\mathbb{E}\left[\zeta^{\frac{\delta}{4}}\Pi_{\Delta,\infty}^{-\frac{\delta}{2}}\right], & \text{if } \delta \in (0, 1/2) \\ \frac{2^{2\delta}}{\alpha_1}\mathbb{E}\left[\ln\left(\frac{S_T}{dv}\right)\right], & \text{if } \delta = 0 \end{cases}$$

$$\overset{(b)}{\leq} \begin{cases} \mathcal{O}(1), & \text{if } \delta \in (0, 1/2) \\ \frac{d2^{2\delta}}{\alpha_1}\mathbb{E}\left[\ln\left(\frac{S_T}{dv}\right)\right], & \text{if } \delta = 0 \end{cases}. \tag{32}$$

In step $(b)$, we used the following inequality, i.e. the *Hölder's* inequality, to obtain the $\mathcal{O}(1)$ result

$$\mathbb{E}\left[\zeta^{\frac{\delta}{4}}\Pi_{\Delta,\infty}^{-\frac{\delta}{2}}\right] \leq \mathbb{E}^{\delta/4}[\zeta]\cdot\mathbb{E}^{\frac{4-\delta}{4}}\left[\Pi_{\Delta,\infty}^{-\frac{2\delta}{4-\delta}}\right] \overset{\text{Lemma D.1 and D.4}}{\leq} C_\zeta^{\delta/4}\cdot C_{v,d,\frac{2\delta}{4-\delta}}^{\frac{4-\delta}{4}} = \mathcal{O}(1).$$

This completes the proof. □

### D.3.6. PROOF OF LEMMA D.7

*Proof.* We only present the case where $\delta = 0$. The case where $\delta > 0$ can be treated using exactly the same approach. Recall the approximate descent inequality (Lemma 4.1)

$$\Pi_{\Delta,t}\hat{f}(u_{t+1}) - \Pi_{\Delta,t-1}\hat{f}(u_t) \leq -\frac{1}{2}\Pi_{\Delta,t}\sum_{i=1}^{d}\zeta_i(t) + C_2\|\eta_{v_{t-1}}\circ m_{t-1}\|^2 + \sum_{i=1}^{d}\Delta_{t,i}|\nabla_i f(u_t)m_{t-1,i}|$$

$$+ (L_f + 1) \sum_{i=1}^{d} \eta_{v_t,i}^2 g_{t,i}^2 + \Pi_{\Delta,t} M_t.$$

We divide both sides of the above inequality by $\ln^2(t+1)$. Noticing that $\ln^2(t+1) < \ln^2(t+2)$, we obtain

$$\frac{\Pi_{\Delta,t}\hat{f}(u_{t+1})}{\ln^2(t+2)} - \frac{\Pi_{\Delta,t-1}\hat{f}(u_t)}{\ln^2(t+1)} \leq C_2 \frac{\|\eta_{v_{t-1}} \circ m_{t-1}\|^2}{\ln^2(t+1)} + \sum_{i=1}^{d} \frac{\Delta_{t,i}|\nabla_i f(u_t) m_{t-1,i}|}{\ln^2(t+1)} \tag{33}$$

$$+ (L_f + 1) \sum_{i=1}^{d} \frac{\eta_{v_t,i}^2 g_{t,i}^2}{\ln^2(t+1)} + \Pi_{\Delta,t} \frac{M_t}{\ln^2(t+1)}. \tag{34}$$

Assign

$$\Omega_t := C_2 \frac{\|\eta_{v_{t-1}} \circ m_{t-1}\|^2}{\ln^{t+1}} + \sum_{i=1}^{d} \frac{\Delta_{t,i}|\nabla_i f(u_t) m_{t-1,i}|}{\ln^2(t+1)} + (L_f + 1) \sum_{i=1}^{d} \frac{\eta_{v_t,i}^2 g_{t,i}^2}{\ln^2(t+1)}.$$

For the term $\sum_{t=1}^{+\infty} \mathbb{E}[\Omega_t]$, we can estimate it as

$$\sum_{t=1}^{+\infty} \mathbb{E}[\Omega_t] := C_2 \sum_{t=1}^{+\infty} \frac{\mathbb{E}\|\eta_{v_{t-1}} \circ m_{t-1}\|^2}{\ln^{t+1}} + \sum_{t=1}^{+\infty} \sum_{i=1}^{d} \frac{\mathbb{E}[\Delta_{t,i}|\nabla_i f(u_t) m_{t-1,i}|]}{\ln^2(t+1)}$$

$$+ (L_f + 1) \sum_{i=1}^{d} \sum_{t=1}^{+\infty} \frac{\mathbb{E}[\eta_{v_t,i}^2 g_{t,i}^2]}{\ln^2(t+1)}$$

$$\overset{(a)}{\leq} \mathcal{O}(1) + \sum_{i=1}^{d} \mathcal{O}\left(\sum_{t=1}^{+\infty} \mathbb{E}\left[\frac{\mathbb{E}[\eta_{v_t,i}^2 g_{t,i}^2]}{\ln^2(t+1)}\right]\right)$$

$$\overset{(b)}{\leq} \mathcal{O}(1) + \sum_{i=1}^{d} \mathcal{O}\left(\sum_{t=1,S_{t,i}>2v}^{+\infty} \mathbb{E}\left[\zeta'^2 \frac{\mathbb{E}[\eta_{v_t,i}^2 g_{t,i}^2]}{\ln^2(S_{t,i}/v)}\right]\right).$$

In step $(a)$, we use Property 4 and Lemma D.2. In step $(b)$, we use the last result from Lemma D.4, which states

$$\ln\left(\frac{S_{t,i}}{v}\right) \leq \ln\left(\frac{S_t}{v}\right) \leq \zeta' \ln(T+1).$$

Then, using the series-integral inequality, we bound $\sum_{t=1}^{+\infty} \mathbb{E}[\Omega_t]$ and obtain

$$\sum_{t=1}^{+\infty} \mathbb{E}[\Omega_t] \leq \mathcal{O}(1) + \mathcal{O}\left(\sum_{i=1}^{d} \mathbb{E}\left[\sum_{t=1,S_{t,i}>2v}^{+\infty} \frac{(\zeta'^2)\frac{g_{t,i}^2}{v}}{\ln^2(\frac{S_{t,i}}{v})\frac{S_{t,i}}{v}}\right]\right)$$

$$< \mathcal{O}(1) + \mathcal{O}\left(\sum_{i=1}^{d} \mathbb{E}\left[\int_2^{+\infty} \frac{\zeta'^2}{x\ln^2 x} dx\right]\right)$$

$$= \mathcal{O}(1) + \mathcal{O}(\mathbb{E}[\zeta'^2]).$$

Next, we use the explicit expression for $\zeta'$ given in Lemma D.4 to bound $\mathbb{E}[\zeta'^2]$. We have:

$$\mathbb{E}[\zeta'^2] = \mathbb{E}\left[16\left(1 + \frac{1}{2}\ln\left(\max\left\{e, \Pi_{\Delta,\infty}^{-1}\zeta\right\}\right)\right)\right] < +\infty.$$

As a result, we have

$$\sum_{t=1}^{+\infty} \mathbb{E}[\Omega_t] < +\infty. \tag{35}$$

According to the *Lebesgue's Monotone Convergence* theorem, we know that the above result implies

$$\sum_{t=1}^{+\infty} \mathbb{E}[\Omega_t | \mathscr{F}_{t-1}] < +\infty \text{ a.s.} \tag{36}$$

Next, we take the conditional expectation with respect to $\mathscr{F}_{t-1}$ on both sides of Equation (33), which obtains

$$\mathbb{E}\left[\frac{\Pi_{\Delta,t+1}\hat{f}(u_{t+1})}{\ln^2(t+2)}\bigg| \mathscr{F}_{t-1}\right] \leq \frac{\Pi_{\Delta,t}\hat{f}(u_t)}{\ln^2(t+1)} + \mathbb{E}[\Omega_t | \mathscr{F}_{t-1}] + 0. \tag{37}$$

Based on the result from Equation (36) and the *Supermartingale Convergence* theorem, we deduce that $\frac{\Pi_{\Delta,t}\hat{f}(u_t)}{\ln^2(t+1)}$ converges almost surely. Then, according to Property 5, we bound $f(w_t) - f^*$ using $\hat{f}(u_t)$. This concludes our first result. Next, we take the expectation on both sides of Equation (33), which yields

$$\mathbb{E}\left[\frac{\Pi_{\Delta,t+1}\hat{f}(u_{t+1})}{\ln^2(t+2)}\right] \leq \mathbb{E}\left[\frac{\Pi_{\Delta,t}\hat{f}(u_t)}{\ln^2(t+1)}\right] + \mathbb{E}[\Omega_t] + 0. \tag{38}$$

Based on the convergence result of the expectation summation in Equation (35) and a summation formula for a recursive sequence, we conclude our second result. Thus, the analysis for the case $\delta = 0$ has been completed. The case $\delta > 0$ could be analyzed using the same method, which concludes the lemma. $\square$

### D.3.7. PROOF OF LEMMA D.8

*Proof.* Since the case of $\delta > 0$ is relatively straightforward, we first analyze the scenario where $\delta > 0$. According to the second conclusion for $\delta > 0$ in Lemma D.7, we obtain

$$\mathbb{E}[S_T^{3/4}] = \mathbb{E}[\Pi_{\Delta,T}^{-3/4}\Pi_{\Delta,T}^{3/4}S_T^{3/4}] \overset{\textit{Hölder's inequality}}{\leq} \mathbb{E}^{1/4}[\Pi_{\Delta,T}^{-3}]\,\mathbb{E}^{3/4}[\Pi_{\Delta,T}S_T].$$

Then according to Lemma D.1, we have $\mathbb{E}[\Pi_{\Delta,T}^{-3}] \leq C_{v,d,3}$. For the other term, $\mathbb{E}[\Pi_{\Delta,T}S_T]$, we can handle the term as follows.

$$\mathbb{E}[\Pi_{\Delta,T}S_T] \leq S_0 + \mathbb{E}\left[\Pi_{\Delta,T}\sum_{t=1}^{T}\|g_t\|^2\right] \leq dv + \mathbb{E}\left[\sum_{t=1}^{T}\Pi_{\Delta,T}\|g_t\|^2\right]$$

$$= dv + \sum_{t=1}^{T}\mathbb{E}\left[\Pi_{\Delta,T}\,\mathbb{E}[\|g_t\|^2 | \mathscr{F}_{t-1}]\right]$$

$$\overset{\text{Property 1}}{\leq} dv + \sum_{t=1}^{T}\mathbb{E}\left[\Pi_{\Delta,T}((A + 2L_f B)(f(w_t) - f^*) + C)\right]$$

$$\overset{\text{Lemma D.7}}{\leq} dv + ((A + 2L_f B)M_\delta + C)T.$$

This implies

$$\mathbb{E}[\sqrt{S_T}] \leq C_{v,d,3}^{1/4}(dv + ((A + 2L_f B)M_\delta + C)T^{3/4} = \mathcal{O}(T^{3/4}).$$

For the case where $\delta = 0$, we use the same approach as in the case of $\delta > 0$ and apply the corresponding conclusion for $\delta = 0$ from Lemma D.7. Thus, we obtain

$$\mathbb{E}[S_T^{3/4}] = \mathcal{O}(T^{3/4}\ln^{3/2}T). \qquad \square$$

### D.3.8. PROOF OF LEMMA D.9

*Proof.* We discuss two cases based on the value of $\lambda$. In the first case, when $\lambda = 1$, we have

$$v_{t+1} = \left(1 - \frac{1}{t+1}\right)v_t + \frac{1}{t+1}g_t^{\circ 2} \ (\forall\, t \geq 1),$$

that is

$$(t+1)v_{t+1} = tv_t + g_t^{\circ 2}.$$

Summing over all the coordinates, we obtain

$$(t+1)\Sigma_{v_{t+1}} = t\Sigma_{v_t} + \|g_t\|^2. \tag{39}$$

Multiplying both sides of the above equation by $\Pi_{\Delta,t}$, and noting that $\Pi_{\Delta,t} \geq \Pi_{\Delta,t+1}$, we have

$$(t+1)\Pi_{\Delta,t+1}\Sigma_{v_{t+1}} = t\Pi_{\Delta,t}\Sigma_{v_t} + \Pi_{\Delta,t}\|g_t\|^2.$$

Taking the expectation on both sides, we obtain

$$\begin{aligned}
(t+1)\,\mathbb{E}[\Pi_{\Delta,t+1}\Sigma_{v_{t+1}}] &\leq t\,\mathbb{E}[\Pi_{\Delta,t}\Sigma_{v_t}] + \mathbb{E}[\Pi_{\Delta,t}\|g_t\|^2] \\
&= t\,\mathbb{E}[\Pi_{\Delta,t}\Sigma_{v_t}] + \mathbb{E}[\Pi_{\Delta,t}\,\mathbb{E}[\|g_t\|^2|\mathscr{F}_{t-1}]] \\
&\overset{\text{Property 1}}{\leq} t\,\mathbb{E}[\Pi_{\Delta,t}\Sigma_{v_t}] + (A+2L_fB)\,\mathbb{E}[\Pi_{\Delta,t}(f(w_t)-f^*)] + C \\
&\leq t\,\mathbb{E}[\Pi_{\Delta,t}\Sigma_{v_t}] + (A+2L_fB)\Big(\sup_{t\geq 1}\mathbb{E}[\Pi_{\Delta,t}(f(w_t)-f^*)]\Big) + C \\
&\overset{\text{Lemma D.7}}{\leq} \begin{cases} t\,\mathbb{E}[\Pi_{\Delta,t}\Sigma_{v_t}] + (A+2L_fB)M_\delta + C, & \text{if } \delta \in (0,1/2) \\ t\,\mathbb{E}[\Pi_{\Delta,t}\Sigma_{v_t}] + (A+2L_fB)M_0\ln^2 t + C, & \text{if } \delta = 0 \end{cases}.
\end{aligned}$$

By iterating the above inequality, we finally have

$$(t+1)\,\mathbb{E}[\Pi_{\Delta,t+1}\Sigma_{v_{t+1}}] \leq \begin{cases} \mathbb{E}[\Pi_{\Delta,2}\Sigma_{v_1}] + \big((A+2L_fB)M_\delta + C\big)t, & \text{if } \delta \in (0,1/2) \\ \mathbb{E}[\Pi_{\Delta,2}\Sigma_{v_1}] + \big((A+2L_fB)M_0\ln^2 t + C\big)t, & \text{if } \delta = 0 \end{cases}.$$

This implies that for any $t \geq 1$,

$$\mathbb{E}[\Pi_{\Delta,t+1}\Sigma_{v_{t+1}}] \leq \begin{cases} \mathbb{E}[\Pi_{\Delta,2}\Sigma_{v_1}] + (A+2L_fB)M_\delta + C, & \text{if } \delta \in (0,1/2) \\ \mathbb{E}[\Pi_{\Delta,2}\Sigma_{v_1}] + (A+2L_fB)M_0\ln^2 t + C, & \text{if } \delta = 0 \end{cases},$$

that is

$$\sup_{t\geq 1}\mathbb{E}[\Pi_{\Delta,t}\Sigma_{v_t}] < \begin{cases} \mathbb{E}[\Pi_{\Delta,2}\Sigma_{v_1}] + (A+2L_fB)M_\delta + C, & \text{if } \delta \in (0,1/2) \\ \mathbb{E}[\Pi_{\Delta,2}\Sigma_{v_1}] + (A+2L_fB)M_0\ln^2 t + C, & \text{if } \delta = 0 \end{cases}.$$

Next, we discuss the scenario when $\lambda > 1$. In this case, we have the following inequality.

$$\Sigma_{v_{t+1}} \leq \Sigma_{v_t} + \frac{1}{(t+1)^\lambda}\|g_t\|^2.$$

We multiply both sides of the above inequality by $\Pi_{\Delta,t}$ and, noting its monotonicity, we have

$$\Pi_{\Delta,t+1}\Sigma_{v_{t+1}} \leq \Pi_{\Delta,t}\Sigma_{v_t} + \frac{\Pi_{\Delta,t}}{(t+1)^\lambda}\|g_t\|^2. \tag{40}$$

Taking the conditional expectation with respect to $\mathscr{F}_{t-1}$ on both sides of the inequality, we have

$$\mathbb{E}[\Pi_{\Delta,t+1}\Sigma_{v_{t+1}}|\mathscr{F}_{t-1}] \leq \Pi_{\Delta,t}\Sigma_{v_t} + \frac{\Pi_{\Delta,t}}{(t+1)^\lambda}\,\mathbb{E}[\Pi_{\Delta,t}\|g_t\|^2|\mathscr{F}_{t-1}].$$

According to Property 1 and Lemma D.7, we obtain

$$\sum_{t=1}^{+\infty}\frac{\Pi_{\Delta,t}}{(t+1)^\lambda}\,\mathbb{E}[\Pi_{\Delta,t}\|g_t\|^2|\mathscr{F}_{t-1}]$$

$$
\leq \begin{cases} \left((A + 2L_f B) \sup_{t \geq 1} \left(\Pi_{\Delta,t}(f(w_t) - f^*)\right) + C\right) \cdot \sum_{t=1}^{+\infty} \frac{1}{(t+1)^\lambda}, & \text{if } \delta \in (0, 1/2) \\ \left((A + 2L_f B) \sup_{t \geq 1} \left(\frac{\Pi_{\Delta,t}(f(w_t) - f^*)}{\ln^2 t}\right) + C\right) \cdot \sum_{t=1}^{+\infty} \frac{\ln^2 t}{(t+1)^\lambda}, & \text{if } \delta = 0 \end{cases}
$$

$$
< +\infty \quad \text{a.s.}
$$

By the *Supermartingale Convergence theorem*, we obtain that $\Pi_{\Delta,t} \Sigma_{v_t}$ converges almost surely, which implies that $\sup_{t \geq 1} \Pi_{\Delta,t} \Sigma_{v_t} < +\infty$ a.s. According to Lemma D.1, where $\sup_{t \geq 1} \Pi_{\Delta,t}^{-1} < +\infty$ a.s., we can immediately deduce that $\sup_{t \geq 1} \Sigma_{v_t} < +\infty$ a.s. Next, we prove that the expected supremum is finite. Taking the expectation on both sides of Equation (40), we obtain

$$
\mathbb{E}\left[\Pi_{\Delta,t+1} \Sigma_{v_{t+1}}\right] \leq \mathbb{E}\left[\Pi_{\Delta,t} \Sigma_{v_t}\right] + \frac{1}{(t+1)^\lambda} \mathbb{E}[\Pi_{\Delta,t} \|g_t\|^2].
$$

By summing the above recursive inequalities and using the results from Property 1 and Lemma D.7, we can easily prove that

$$
\sup_{t \geq 1} \mathbb{E}[\Pi_{\Delta,t} \Sigma_{v_t}] < \begin{cases} (A + 2L_f B) M_\delta + C) \sum_{t=1}^{+\infty} \frac{1}{(t+1)^\lambda}, & \text{if } \delta \in (0, 1/2) \\ (A + 2L_f B) M_0 + C) \sum_{t=1}^{+\infty} \frac{\ln^2 t}{(t+1)^\lambda}, & \text{if } \delta = 0 \end{cases} < +\infty.
$$

With this inequality we complete the proof. $\qquad \square$

### D.3.9. PROOF OF LEMMA D.10

*Proof.* According to the result from Lemma D.5, it is straightforward to see that when $\delta > 0$,

$$
\sum_{t=2}^{T} \mathbb{E}\left[\Pi_{\Delta,t} \frac{\eta_{t-1} \|\nabla f(w_t)\|^2}{\sqrt{\Sigma_{v_{t-1}}} + \mu}\right] \leq \sum_{t=2}^{T} \mathbb{E}\left[\Pi_{\Delta,t} \frac{\eta_{t-1}}{\sqrt{\Sigma_{v_{t-1}}} + \mu} \sum_{i=1}^{d} (\nabla_i f(w_t))^2\right]
$$

$$
\leq \sum_{t=1}^{T} \mathbb{E}\left[\Pi_{\Delta,t} \sum_{i=1}^{d} \zeta_i(t)\right]
$$

$$
< C_{4,\delta} < +\infty.
$$

Next, we apply the *Lebesgue's Monotone Convergence theorem*

$$
\sum_{t=2}^{+\infty} \Pi_{\Delta,t} \frac{\eta_{t-1} \|\nabla f(w_t)\|^2}{\sqrt{\Sigma_{v_{t-1}}} + \mu} < +\infty \quad \text{a.s.}
$$

Then, by combining the almost surely boundedness of $\sup_{t \geq 1} \Pi_{\Delta,t}^{-1}$ and $\sup_{t \geq 1} \Sigma_{v_t}$ from Lemma D.1 and Lemma D.9, it follows

$$
\sum_{t=1}^{+\infty} \eta_t \|\nabla f(w_t)\|^2 \leq \|\nabla f(w_1)\|^2 + \sum_{t=2}^{+\infty} \eta_{t-1} \|\nabla f(w_t)\|^2
$$

$$
< \|\nabla f(w_1)\|^2 + \left(\sup_{t \geq 1} \Pi_{\Delta,t+1}^{-3/2}\right) \cdot \left(\sqrt{\sup_{t \geq 1} \Pi_{\Delta,t} \Sigma_{v_t}} + \mu\right) \cdot \sum_{t=2}^{+\infty} \Pi_{\Delta,t} \frac{\eta_{t-1} \|\nabla f(w_t)\|^2}{\sqrt{\Sigma_{v_{t-1}}} + \mu}
$$

$$
< +\infty \quad \text{a.s.}
$$

According to the L-smoothness assumption (Assumption 2.2), it is immediate that

$$
|\|\nabla f(w_t)\| - \|\nabla f(u_t)\|| \leq L_f \|w_t - u_t\| = \frac{L_f \beta_1}{1 - \beta_1} \|\eta_{v_{t-1}} \circ m_{t-1}\|,
$$

that is,

$$
\|\nabla f(u_t)\|^2 \leq \left(\|\nabla f(w_t)\| + \frac{L_f \beta_1}{1 - \beta_1} \|\eta_{v_{t-1}} \circ m_{t-1}\|\right)^2
$$

$$
\leq 2\|\nabla f(w_t)\|^2 + \frac{2L_f^2 \beta_1^2}{(1 - \beta_1)^2} \|\eta_{v_{t-1}} \circ m_{t-1}\|^2.
$$

This implies

$$\sum_{t=1}^{+\infty} \eta_t \|\nabla f(u_t)\|^2 \leq 2 \sum_{t=1}^{+\infty} \eta_t \|\nabla f(w_t)\|^2 + \frac{2L_f^2 \beta_1^2}{(1-\beta_1)^2} \sum_{t=1}^{+\infty} \|\eta_{v_{t-1}} \circ m_{t-1}\|^2.$$

According to Property 5, for any $\delta > 0$,

$$\left(\frac{L_f \beta_1}{1-\beta_1}\right)^2 \sum_{t=1}^{T} \mathbb{E} \|\eta_{v_{t-1}} \circ m_{t-1}\|^2 \leq \mathcal{O}\left(\sum_{t=1}^{T} \mathbb{E} \|\eta_{v_t} \circ g_t\|^2\right) + \mathcal{O}(1)$$

$$\overset{\text{Eq. (32)}}{\leq} \mathcal{O}(1).$$

Applying the *Lebesgue's Monotone Convergence theorem*, we obtain

$$\left(\frac{L_f \beta_1}{1-\beta_1}\right)^2 \sum_{t=1}^{T} \|\eta_{v_{t-1}} \circ m_{t-1}\|^2 < +\infty \quad \text{a.s.,} \tag{41}$$

that is

$$\sum_{t=1}^{+\infty} \eta_t \|\nabla f(u_t)\|^2 \leq 2 \sum_{t=1}^{+\infty} \eta_t \|\nabla f(w_t)\|^2 + \frac{2L_f^2 \beta_1^2}{(1-\beta_1)^2} \sum_{t=1}^{+\infty} \|\eta_{v_{t-1}} \circ m_{t-1}\|^2 < +\infty \quad \text{a.s..} \qquad \square$$

### D.3.10. PROOF OF THEOREM 3.1

*Proof.* According to Lemma D.5, we have:

$$\sum_{t=1}^{T} \mathbb{E}\left[\Pi_{\Delta,t} \sum_{i=1}^{d} \zeta_i(t)\right] \leq \begin{cases} C_{4,\delta}, & \text{if } \delta \in (0, 1/2) \\ C_5 + C_6 \mathbb{E}\left[\ln(S_T)\right], & \text{if } \delta = 0 \end{cases}.$$

According to the monotonicity of $\eta_{v_t,i}$ in Property 2 and the monotonicity of $\Pi_{\Delta,t}$ itself, we obtain the following inequality.

$$\sum_{t=1}^{T} \mathbb{E}\left[\Pi_{\Delta,T} \frac{\|\nabla f(w_t)\|^2}{T^{\frac{1}{2}+\delta}(\sqrt{v_T}+\mu)}\right] \leq \sum_{t=1}^{T} \mathbb{E}\left[\Pi_{\Delta,T} \sum_{i=1}^{d} \zeta_i(t)\right] \leq \begin{cases} C_{4,\delta}, & \text{if } \delta \in (0, 1/2) \\ C_5 + C_6 \mathbb{E}\left[\ln(S_T)\right], & \text{if } \delta = 0 \end{cases}.$$

For the leftmost part of the above inequality, we apply the *Cauchy-Schwarz inequality* and obtain

$$\mathbb{E}\left[\Pi_{\Delta,T}^{-1} T^{\frac{1}{2}+\delta}(\sqrt{v_T}+\mu)\right]\left(\sum_{t=1}^{T} \mathbb{E}\left[\Pi_{\Delta,T} \frac{\|\nabla f(w_t)\|^2}{T^{\frac{1}{2}+\delta}(\sqrt{v_T}+\mu)}\right]\right) \geq \mathbb{E}\left[\sqrt{\sum_{t=1}^{T} \|\nabla f(w_t)\|^2}\right],$$

which means

$$\mathbb{E}\left[\sqrt{\sum_{t=1}^{T} \|\nabla f(w_t)\|^2}\right] \leq \begin{cases} C_{4,\delta} \mathbb{E}\left[\Pi_{\Delta,T}^{-1} T^{\frac{1}{2}+\delta}(\sqrt{v_T}+\mu)\right], & \text{if } \delta \in (0, 1/2) \\ C_5 \mathbb{E}\left[\Pi_{\Delta,T}^{-1} T^{\frac{1}{2}+\delta}(\sqrt{v_T}+\mu)\right] + C_6 \mathbb{E}\left[\Pi_{\Delta,T}^{-1} T^{\frac{1}{2}+\delta}(\sqrt{v_T}+\mu)\right] \mathbb{E}\left[\ln(S_T)\right], & \text{if } \delta = 0 \end{cases}.$$

Combining the results from Lemma D.8, Lemma D.9 and Lemma D.1, we obtain

$$\mathbb{E}\left[\Pi_{\Delta,T}^{-1} T^{\frac{1}{2}+\delta}(\sqrt{v_T}+\mu)\right] \leq 2T^{\frac{1}{2}+\delta}\sqrt{\mathbb{E}[\Pi_{\Delta,T}^{-3}]}\sqrt{\mathbb{E}[\Pi_{\Delta,T}(v_T+\mu^2)]}$$

$$\leq \begin{cases} C_{v,d,3}^{1/2}\mathcal{O}(T^{\frac{1}{2}+\delta}), & \text{if } \gamma > 1 \\ C_{v,d,3}^{1/2}\mathcal{O}(T^{\frac{1}{2}+\delta}), & \text{if } \gamma = 1, \ \delta \in (0, 1] \\ C_{v,d,3}^{1/2}\mathcal{O}(\sqrt{T}\ln T), & \text{if } \gamma = 1, \ \delta = 0. \end{cases}$$

and

$$\mathbb{E}[\ln(S_T)] = \frac{4}{3}\,\mathbb{E}[\ln(S_T^{3/4})] \leq \frac{4}{3}\ln(\mathbb{E}[S_T^{3/4}]) = \begin{cases} \mathcal{O}(\ln T), & \text{if } \delta \in (0, 1/2) \\ \mathcal{O}(\ln T) + \mathcal{O}(\ln\ln T), & \text{if } \delta = 0 \end{cases}.$$

Combining the two estimates above, we finally obtain

$$\mathbb{E}\left[\sqrt{\sum_{t=1}^{T}\|\nabla f(w_t)\|^2}\right] \leq \begin{cases} \mathcal{O}(T^{\frac{1}{2}+\delta}), & \text{if } \delta \in (0, 1/2) \\ \mathcal{O}(\sqrt{T}\ln T), & \text{if } \gamma > 1,\ \delta = 0 \\ \mathcal{O}(\sqrt{T}\ln^2 T), & \text{if } \gamma = 1,\ \delta = 0. \end{cases} \tag{42}$$

This implies that for any $s \in (0, 1)$, the inequality

$$\frac{1}{T}\sum_{t=1}^{T}\|\nabla f(w_t)\|^2 \leq \begin{cases} \mathcal{O}\left(\frac{1}{s^2}\frac{1}{T^{\frac{1}{2}-\delta}}\right), & \text{if } \delta \in (0, 1/2) \\ \mathcal{O}\left(\frac{1}{s^2}\frac{\ln T}{\sqrt{T}}\right), & \text{if } \gamma > 1,\ \delta = 0 \\ \mathcal{O}\left(\frac{1}{s^2}\frac{\ln^2 T}{\sqrt{T}}\right), & \text{if } \gamma = 1,\ \delta = 0 \end{cases} \qquad \square$$

holds with probability at least $1 - s$.

### D.3.11. PROOF OF LEMMA D.6

*Proof.* From Eq. (42), we have $\forall\, t_0 \geq 0$,

$$\lim_{T\to+\infty}\mathbb{E}\left[\inf_{t_0 < t \leq T}\|\nabla f(w_t)\|\right] \leq \lim_{T\to+\infty}\frac{1}{T - t_0 + 1}\mathbb{E}\left[\sqrt{\sum_{t=t_0}^{T}\|\nabla f(w_t)\|^2}\right] = 0.$$

Since, for a fixed $t_0$, the sequence $\{\inf_{t_0 < t \leq T}\|f(w_t)\|\}_{T > t_0}$ is monotonically decreasing and non-negative, by the *Lebesgue's Monotone Convergence theorem*, we readily obtain:

$$\mathbb{E}\left[\inf_{t > t_0}\|\nabla f(w_t)\|\right] = \mathbb{E}\left[\lim_{T\to+\infty}\inf_{t_0 < t \leq T}\|\nabla f(w_t)\|\right] = \lim_{T\to+\infty}\mathbb{E}\left[\inf_{t_0 < t \leq T}\|\nabla f(w_t)\|\right] = 0$$

The second equality follows from the *Lebesgue's Monotone Convergence theorem*, which allows the interchange of the limit and expectation. Since $\inf_{t > t_0}\|\nabla f(w_t)\| \geq 0$, we can directly deduce that $\inf_{t > t_0}\|\nabla f(w_t)\| = 0$ a.s., from the fact that $\mathbb{E}[\inf_{t > t_0}\|\nabla f(w_t)\|] = 0$. Furthermore, given the arbitrariness of $t_0$, we can obtain that there exists a subsequence $\{w_{c_t}\}_{t \geq 1}$ of $\{w_t\}_{t \geq 1}$ such that

$$\lim_{t\to+\infty}\|\nabla f(w_{c_t})\| = 0 \quad \text{a.s.}$$

This completes the proof. $\qquad \square$

### D.3.12. PROOF OF THEOREM 3.2

*Proof.* According to Lemma D.10, we have

$$\left|\|\nabla f(w_t)\| - \|\nabla f(u_t)\|\right| \leq L_f\|w_t - u_t\| = \frac{L_f\beta_1}{1 - \beta_1}\|\eta_{v_{t-1}} \circ m_{t-1}\| \to 0 \text{ a.s.}$$

This implies that we only need to prove $\lim_{t\to+\infty}\|\nabla f(u_t)\| = 0$ a.s. To achieve this objective, we proceed as follows.

For any $l > 0$, we construct the following stopping time[2] sequence $\{\tau_{l,n}\}_{n \geq 1}$ :

$$\tau_{l,1} := \min\{t \geq 1 : \|\nabla f(u_t)\| > l\}, \quad \tau_{l,2} := \min\{t > \tau_{l,1} : \|\nabla f(u_t)\| \leq l\},$$

---

[2]In this paper, we adopt the following definition of stopping time: Let $\tau$ be a random variable defined on the filtered probability space $(\Omega, \mathscr{F}, (\mathscr{F}_n)_{n\in\mathbb{N}}, \mathbb{P})$ with values in $\mathbb{N} \cup \{+\infty\}$. Then $\tau$ is called a stopping time (with respect to the filtration $(\mathscr{F}_n)_{n\in\mathbb{N}}$) if the condition $\{\tau = n\} \in \mathscr{F}_n$ holds for all $n$.

...,
$$\tau_{l,2k-1} := \min\{t > \tau_{l,2k-2} : \|\nabla f(u_t)\| > l\}, \quad \tau_{l,2k} := \min\{t > \tau_{l,2k-1} : \|\nabla f(u_t)\| \le l\}.$$

According to the subsequence convergence result in Lemma D.6, we know that when $\tau_{2k-1} < +\infty$ ($\forall\, k \ge 1$), it must hold that $\tau_{2k} < +\infty$ a.s. We now discuss two cases.

1. When there exists some $k_0 \ge 1$ such that $\tau_{2k_0-1} = +\infty$, this implies that eventually $\{\|\nabla f(u_t)\|\}_{t \ge 1}$ will remain below $l$, i.e.,

$$\limsup_{t \to +\infty} \|\nabla f(u_t)\| < l. \tag{43}$$

2. Next, we focus on the second case, where for all $\tau_{2k-1}$, we have $\tau_{2k-1} < +\infty$. In this situation, we examine the behavior of $\sup_{\tau_{2k-1} \le t < \tau_{2k}} \|\nabla f(u_t)\|$. We have

$$\sup_{\tau_{2k-1} \le t < \tau_{2k}} \|\nabla f(u_t)\| \le l + \sup_{\tau_{2k-1} \le t < \tau_{2k}} \|\nabla f(u_t)\| - \|\nabla f(u_{\tau_{2k-1}-1})\|$$

$$\le l + \left( \sum_{t=\tau_{2k-1}-1}^{\tau_{2k}-1} \big| \|\nabla f(u_t)\| - \|\nabla f(u_{t-1})\| \big| \right)$$

$$\overset{\text{l-smooth}}{\le} l + \left( L_f \sum_{t=\tau_{2k-1}-1}^{\tau_{2k}-1} \|u_t - u_{t-1}\| \right)$$

$$\overset{\text{Eq. (2)}}{\le} l + L_f \underbrace{\left( \sum_{t=\tau_{2k-1}-1}^{\tau_{2k}-1} \|\eta_{v_t} \circ g_t\| \right)}_{\Upsilon_{k,1}}$$

$$+ \frac{\beta_1 L_f^2}{1-\beta_1} \underbrace{\left( \sum_{t=\tau_{2k-1}-1}^{\tau_{2k}-1} \|\Delta_t \circ m_{t-1}\| \right)}_{\Upsilon_{k,2}}.$$

Our next goal is to prove separately that $\limsup_{k \to +\infty} \Upsilon_{k,1} = 0$ a.s. and $\limsup_{k \to +\infty} \Upsilon_{k,2} = 0$ a.s. For $\Upsilon_{k,1}$, we have

$$\Upsilon_{k,1} = \left( \sum_{t=\tau_{2k-1}-1}^{\tau_{2k}-1} \|\eta_{v_t} \circ g_t\| \right) = \left( \sum_{t=\tau_{2k-1}-1}^{\tau_{2k}-1} \sum_{i=1}^{d} \eta_{v_t,i} |g_{t,i}| \right)$$

$$= \left( \sum_{t=\tau_{2k-1}-1}^{\tau_{2k}-1} \sum_{i=1}^{d} \frac{\eta_t |g_{t,i}|}{\sqrt{v_t} + \mu} \right) \le \left( \sum_{t=\tau_{2k-1}-1}^{\tau_{2k}-1} \sum_{i=1}^{d} \frac{\eta_t |g_{t,i}|}{\mu} \right)$$

$$= \frac{1}{\mu} \left( \sum_{t=\tau_{2k-1}-1}^{\tau_{2k}-1} \sum_{i=1}^{d} \eta_t\, \mathbb{E}[|g_{t,i}| \,|\, \mathscr{F}_{t-1}] \right)$$

$$+ \frac{1}{\mu} \left( \sum_{t=\tau_{2k-1}-1}^{\tau_{2k}-1} \sum_{i=1}^{d} \eta_t (|g_{t,i}| - \mathbb{E}[|g_{t,i}| \,|\, \mathscr{F}_{t-1}]) \right)$$

$$= \frac{1}{\mu} \underbrace{\left( \sum_{t=\tau_{2k-1}-1}^{\tau_{2k}-1} \eta_t \mathbb{E}[|g_t| \,|\, \mathscr{F}_{t-1}] \right)}_{\Upsilon_{k,1,1}} + \frac{1}{\mu} \underbrace{\left( \sum_{t=\tau_{2k-1}-1}^{\tau_{2k}-1} \sum_{i=1}^{d} \eta_t (|g_{t,i}| - \mathbb{E}[|g_{t,i}| \,|\, \mathscr{F}_{t-1}]) \right)}_{\Upsilon_{k,1,2}}.$$

For $\Upsilon_{k,1,1}$, we have

$$\Upsilon_{k,1,1} \overset{\text{Property 1}}{\le} \left( \sum_{t=\tau_{2k-1}-1}^{\tau_{2k}-1} \eta_t \big( (A + 2L_f B)(f(w_t) - f^*) + C \big) \right)$$

$$
\leq \left((A + 2L_f B) \sup_{t \geq 1}(f(w_t) - f^*) + C\right) \cdot \left(\sum_{t=\tau_{2k-1}-1}^{\tau_{2k}-1} \eta_t\right)
$$

$$
= \left((A + 2L_f B) \sup_{t \geq 1}(f(w_t) - f^*) + C\right) \cdot \left(\eta_{\tau_{2k-1}} + \left(\sum_{t=\tau_{2k-1}-1}^{\tau_{2k}} \eta_t\right)\right)
$$

$$
\overset{(a)}{\leq} \frac{1}{l^2}\left((A + 2L_f B) \sup_{t \geq 1}(f(w_t) - f^*) + C\right) \cdot \left(\eta_{\tau_{2k-1}} + \left(\sum_{t=\tau_{2k-1}}^{\tau_{2k}-1} \eta_t \|\nabla f(u_t)\|^2\right)\right).
$$

In step $(a)$, this is due to the fact that, over the interval $[\tau_{2k-1}, \tau_{2k})$, we always have $\|\nabla f(u_t)\|^2 > l^2$. Based on Lemma D.10, we know that

$$
\sum_{t=1}^{+\infty} \eta_t \|\nabla f(u_t)\|^2 < +\infty \quad \text{a.s.}
$$

By applying the *Cauchy's* convergence principle, we can prove that

$$
\lim_{k \to +\infty} \sum_{t=\tau_{2k-1}}^{\tau_{2k}-1} \eta_t \|\nabla f(u_t)\|^2 = 0 \quad \text{a.s.}
$$

On the other hand, it is evident that $\lim_{k \to +\infty} \eta_{\tau_{2k-1}} = 0$. Meanwhile, based on Lemma D.7 and Lemma D.1, we have

$$
\sup_{t \geq 1}(f(w_t) - f^*) \leq \left(\sup_{t \geq 1} \Pi_{\Delta,t}^{-1}\right) \cdot \left(\sup_{t \geq 1}\left(\Pi_{\Delta,t+1}(f(w_t) - f^*)\right)\right) < +\infty \quad \text{a.s.}
$$

Therefore,

$$
\limsup_{k \to +\infty} \Upsilon_{k,1,1} = \lim_{k \to +\infty} \Upsilon_{k,1,1} = 0.
$$

For $\Upsilon_{k,1,2}$, we consider the following martingale difference sequence

$$
\overline{X}_T := \sum_{t=1}^{T} \sum_{i=1}^{d} \eta_t (|g_{t,i}| - \mathbb{E}[|g_{t,i}| \mid \mathscr{F}_{t-1}]).
$$

We can compute

$$
\sum_{t=1}^{+\infty} \mathbb{E}\left[\left(\sum_{i=1}^{d} \eta_t(|g_{t,i}| - \mathbb{E}[|g_{t,i}| \mid \mathscr{F}_{t-1}])\right)^2 \middle| \mathscr{F}_{t-1}\right] \leq d \sum_{t=1}^{+\infty} \eta_t^2 \sum_{i=1}^{d} \mathbb{E}[(|g_{t,i}| - \mathbb{E}[|g_{t,i}| \mid \mathscr{F}_{t-1}])^2]
$$

$$
\leq d \sum_{t=1}^{+\infty} \eta_t^2 \sum_{i=1}^{d} \mathbb{E}[\|g_t\|^2 \mid \mathscr{F}_{t-1}]
$$

$$
\overset{\text{Property 1}}{\leq} d \sum_{t=1}^{+\infty} \eta_t^2 \sum_{i=1}^{d} \left((A + 2L_f B) \sup_{t \geq 1}(f(w_t) - f^*) + C\right)
$$

$$
\leq d \sum_{i=1}^{d} \left((A + 2L_f B)\left(\sup_{t \geq 1} \Pi_{\Delta,t}\right)\left(\sup_{t \geq 1}(f(w_t) - f^*) + C\right)\right) \cdot \sum_{t=1}^{+\infty} \eta_t^2
$$

$$
\overset{\text{Lemma D.1 and D.7}}{<} +\infty \quad \text{a.s.}
$$

By the *Martingale Convergence* theorem, we obtain

$$
\lim_{T \to +\infty} \overline{X}_T = \sum_{t=1}^{+\infty} \sum_{i=1}^{d} \eta_t (|g_{t,i}| - \mathbb{E}[|g_{t,i}| \mid \mathscr{F}_{t-1}]) < +\infty \quad \text{a.s.}
$$

Using the *Cauchy's Convergence* principle, we prove that

$$\limsup_{k\to+\infty} \Upsilon_{k,1,2} = \lim_{k\to+\infty} \sum_{t=\tau_{2k-1}-1}^{\tau_{2k}-1} \sum_{i=1}^{d} \eta_t(|g_{t,i}| - \mathbb{E}[|g_{t,i}||\mathscr{F}_{t-1}]) = 0 \quad \text{a.s.}$$

Combining the above two limit proofs for $\Upsilon_{t,1,1}$ and $\Upsilon_{t,1,2}$, we conclude that

$$\limsup_{k\to+\infty} \Upsilon_{t,1} = 0 \quad \text{a.s.}$$

Similarly, it can be shown that $\lim_{k\to+\infty} \Upsilon_{k,2} = 0$ a.s. Combining the limit results for $\Upsilon_{k,1}$ and $\Upsilon_{k,2}$, we conclude that

$$\limsup_{k\to+\infty} \sup_{\tau_{2k-1}\le t<\tau_{2k}} \|\nabla f(u_t)\| \le l + 0 = l.$$

Moreover, combining $\sup_{\tau_{2k}\le t<\tau_{2k+1}} \|\nabla f(u_t)\| < l$, we can deduce that

$$\limsup_{t\to+\infty} \|\nabla f(u_t)\| \le l \quad \text{a.s.}$$

Then, due to the arbitrariness of $l$, we conclude that

$$\limsup_{t\to+\infty} \|\nabla f(u_t)\| = 0 \quad \text{a.s.}$$

This implies that

$$\lim_{t\to+\infty} \|\nabla f(u_t)\| = 0 \quad \text{a.s.} \qquad \square$$

### D.3.13. PROOF OF THEOREM 3.3

*Proof.* Since we have already proved the almost sure convergence in Theorem 3.2, it is natural to attempt to prove $L_1$ convergence via the *Lebesgue's Dominated Convergence* theorem. To achieve this, we need to find a function $h$ that is $\mathscr{F}_\infty$-measurable and satisfies $\mathbb{E}|h| < +\infty$, and such that for all $t \ge 1$, we have $\|\nabla f(w_t)\| \le |h|$. Since for all $t$, we naturally have $\|\nabla f(w_t)\| \le \sup_{k\ge 1} \|\nabla f(w_k)\|$, we only need to prove that $\mathbb{E}[\sup_{k\ge 1} \|\nabla f(w_k)\|] < +\infty$. We proceed to achieve this goal in the rest of the proof.

Recall the *Approximate Descent Inequality* (Lemma 4.1). We have

$$\Pi_{\Delta,t}\hat{f}(u_{t+1}) - \Pi_{\Delta,t-1}\hat{f}(u_t) \le -\frac{1}{2}\Pi_{\Delta,t}\sum_{i=1}^{d}\zeta_i(t) + C_2\|\eta_{v_{t-1}}\circ m_{t-1}\|^2 + \sum_{i=1}^{d}\Delta_{t,i}|\nabla_i f(u_t)m_{t-1,i}|$$

$$+ (L_f+1)\sum_{i=1}^{d}\eta_{v_t,i}^2 g_{t,i}^2 + \Pi_{\Delta,t}M_t. \tag{44}$$

For any $\lambda > 0$, define the stopping time $\tau_\lambda$ as the first time the sequence $\{\Pi_{\Delta,t}\hat{f}(u_t)\}_{t\ge 1}$ exceeds $\lambda$, i.e.,

$$\tau_\lambda := \min\{t \ge 2 : \Pi_{\Delta,t}\hat{f}(u_t) > \lambda\}.$$

It can be rigorously verified that $\tau_\lambda$ is a stopping time with respect to the filtration $\{\mathscr{F}_t\}_{t\ge 1}$, and satisfies a special property $[\tau_\lambda = n] \in \mathscr{F}_{n-1}$ for all $n \ge 1$. This implies that the preceding time $\tau_\lambda - 1$ is also a stopping time. Next, for any deterministic time $T \ge 3$, we define $\tau_{\lambda,T} := \tau_\lambda \wedge T$. We then sum the indices of Equation (44) from 1 to $\tau_{\lambda,T} - 1$. Specifically, we have

$$\Pi_{\Delta,\tau_{\lambda,T}-1}\hat{f}(u_{\tau_{\lambda,T}}) \le \Pi_{\Delta,0}\hat{f}(u_1) + C_2\sum_{t=1}^{\tau_{\lambda,T}-1}\|\eta_{v_{t-1}}\circ m_{t-1}\|^2 + \sum_{t=1}^{\tau_{\lambda,T}-1}\sum_{i=1}^{d}\Delta_{t,i}|\nabla_i f(u_t)m_{t-1,i}|$$

$$+ (L_f+1)\sum_{t=1}^{\tau_{\lambda,T}-1}\sum_{i=1}^{d}\eta_{v_t,i}^2 g_{t,i}^2 + \sum_{t=1}^{\tau_{\lambda,T}-1}\Pi_{\Delta,t}M_t.$$

Taking the expectation on both sides, we obtain

$$\mathbb{E}\left[\Pi_{\Delta,\tau_{\lambda,T}-1}\hat{f}(u_{\tau_{\lambda,T}})\right] \leq \mathbb{E}\left[\Pi_{\Delta,0}\hat{f}(u_1)\right] + C_2\,\mathbb{E}\left[\sum_{t=1}^{\tau_{\lambda,T}-1}\|\eta_{v_{t-1}}\circ m_{t-1}\|^2\right]$$

$$+ \mathbb{E}\left[\sum_{t=1}^{\tau_{\lambda,T}-1}\sum_{i=1}^{d}\Delta_{t,i}|\nabla_i f(u_t)m_{t-1,i}|\right] + (L_f+1)\,\mathbb{E}\left[\sum_{t=1}^{\tau_{\lambda,T}-1}\sum_{i=1}^{d}\eta_{v_t,i}^2 g_{t,i}^2\right]$$

$$+ \mathbb{E}\left[\sum_{t=1}^{\tau_{\lambda,T}-1}\Pi_{\Delta,t}M_t\right].$$

Since $\{\Pi_{\Delta,t}M_t, \mathscr{F}_t\}_{t\geq 1}$ is a martingale difference sequence and $\tau_{\lambda,T}\leq T < +\infty$, by *Doob's Stopped* theorem, we know that

$$\mathbb{E}\left[\sum_{t=1}^{\tau_{\lambda,T}-1}\Pi_{\Delta,t}M_t\right] = 0.$$

This implies

$$\mathbb{E}\left[\Pi_{\Delta,\tau_{\lambda,T}-1}\hat{f}(u_{\tau_{\lambda,T}})\right] \leq \mathbb{E}\left[\Pi_{\Delta,0}\hat{f}(u_1)\right] + C_2\,\mathbb{E}\left[\sum_{t=1}^{\tau_{\lambda,T}-1}\|\eta_{v_{t-1}}\circ m_{t-1}\|^2\right]$$

$$+ \mathbb{E}\left[\sum_{t=1}^{\tau_{\lambda,T}-1}\sum_{i=1}^{d}\Delta_{t,i}|\nabla_i f(u_t)m_{t-1,i}|\right] + (L_f+1)\,\mathbb{E}\left[\sum_{t=1}^{\tau_{\lambda,T}-1}\sum_{i=1}^{d}\eta_{v_t,i}^2 g_{t,i}^2\right].$$

Using Property 5 and Lemma D.2, we obtain

$$C_2\,\mathbb{E}\left[\sum_{t=1}^{\tau_{\lambda,T}-1}\|\eta_{v_{t-1}}\circ m_{t-1}\|^2\right] + \mathbb{E}\left[\sum_{t=1}^{\tau_{\lambda,T}-1}\sum_{i=1}^{d}\Delta_{t,i}|\nabla_i f(u_t)m_{t-1,i}|\right]$$

$$+ (L_f+1)\,\mathbb{E}\left[\sum_{t=1}^{\tau_{\lambda,T}-1}\sum_{i=1}^{d}\eta_{v_t,i}^2 g_{t,i}^2\right]$$

$$\leq C_2\,\mathbb{E}\left[\sum_{t=1}^{T}\|\eta_{v_{t-1}}\circ m_{t-1}\|^2\right] + \mathbb{E}\left[\sum_{t=1}^{T}\sum_{i=1}^{d}\Delta_{t,i}|\nabla_i f(u_t)m_{t-1,i}|\right]$$

$$+ (L_f+1)\,\mathbb{E}\left[\sum_{t=1}^{T}\sum_{i=1}^{d}\eta_{v_t,i}^2 g_{t,i}^2\right]$$

$$= \mathcal{O}\left(\sum_{t=1}^{T}\mathbb{E}\left[\|\eta_{v_t}\circ g_t\|^2\right]\right) + \mathcal{O}(1)$$

$$\overset{\text{Eq. (32)}}{=} \mathcal{O}(1).$$

This means

$$\mathbb{E}\left[\Pi_{\Delta,\tau_{\lambda,T}-1}\hat{f}(u_{\tau_{\lambda,T}})\right] \leq \overline{M} < +\infty,$$

where

$$\overline{M} := C_2\,\mathbb{E}\left[\sum_{t=1}^{+\infty}\|\eta_{v_{t-1}}\circ m_{t-1}\|^2\right] + \mathbb{E}\left[\sum_{t=1}^{+\infty}\sum_{i=1}^{d}\Delta_{t,i}|\nabla_i f(u_t)m_{t-1,i}|\right]$$

$$+ (L_f+1)\,\mathbb{E}\left[\sum_{t=1}^{+\infty}\sum_{i=1}^{d}\eta_{v_t,i}^2 g_{t,i}^2\right].$$

Meanwhile, we observe the following event decomposition

$$\left[\sup_{2\leq t<T}\Pi_{\Delta,t-1}\hat{f}(u_t)>\lambda\right]=\bigcup_{k=2}^{T-1}[\tau_\lambda=k]=\bigcup_{k=2}^{T-1}[\tau_{\lambda,T}=k].$$

Moreover, since for any $j\neq k$, we have $[\tau_{\lambda,T}=j]\cap[\tau_{\lambda,T}=k]=\emptyset$, it follows that

$$\mathbb{P}\left[\sup_{2\leq t<T}\Pi_{\Delta,t-1}\hat{f}(u_t)>\lambda\right]=\sum_{k=2}^{T-1}\mathbb{P}[\tau_{\lambda,T}=k]\overset{\text{Markov's inequality}}{\leq}\frac{1}{\lambda}\sum_{k=2}^{T-1}\mathbb{E}\left[\Pi_{\Delta,k}\hat{f}(u_k)\mathbb{I}_{[\tau_{\lambda,T}=k]}\right]$$

$$<\frac{1}{\lambda}\mathbb{E}\left[\Pi_{\Delta,\tau_{\lambda,T}-1}\hat{f}(u_{\tau_{\lambda,T}})\right]\leq\frac{\overline{M}}{\lambda}. \tag{45}$$

Next, for any $K\geq1$, we compute $\mathbb{E}\left[\left(\sup_{2\leq t<T}\Pi_{\Delta,t-1}\hat{f}(u_t)\right)^{3/4}\wedge K\right]$. We have

$$\mathbb{E}\left[\left(\sup_{2\leq t<T}\Pi_{\Delta,t-1}\hat{f}(u_t)\right)^{3/4}\wedge K\right]=-\int_0^{+\infty}x\ \mathrm{d}\left(\mathbb{P}\left[\left(\sup_{2\leq t<T}\Pi_{\Delta,t-1}\hat{f}(u_t)\right)^{3/4}\wedge K>x\right]\right)$$

$$=-\int_0^{+\infty}\left(\int_0^x1\mathrm{d}\lambda\right)\mathrm{d}\left(\mathbb{P}\left[\left(\sup_{2\leq t<T}\Pi_{\Delta,t-1}\hat{f}(u_t)\right)^{3/4}\wedge K>x\right]\right)$$

$$\overset{\textit{Fubini's theorem}}{=}-\int_0^{+\infty}\left(\int_\lambda^{+\infty}1\mathrm{d}\left(\mathbb{P}\left[\left(\sup_{2\leq t<T}\Pi_{\Delta,t-1}\hat{f}(u_t)\right)^{3/4}\wedge K>x\right]\right)\right)\mathrm{d}\lambda$$

$$=\int_0^{+\infty}\mathbb{P}\left[\left(\sup_{2\leq t<T}\Pi_{\Delta,t-1}\hat{f}(u_t)\right)^{3/4}\wedge K>\lambda\right]\mathrm{d}\lambda$$

$$\leq1+\int_1^{+\infty}\mathbb{P}\left[\left(\sup_{2\leq t<T}\Pi_{\Delta,t-1}\hat{f}(u_t)\right)^{3/4}\wedge K>\lambda\right]\mathrm{d}\lambda$$

$$=1+\int_1^{+\infty}\mathbb{P}\left[\left(\sup_{2\leq t<T}\Pi_{\Delta,t-1}\hat{f}(u_t)\right)\wedge K^{4/3}>\lambda^{4/3}\right]\mathrm{d}\lambda$$

$$<1+\int_1^{+\infty}\mathbb{P}\left[\left(\sup_{2\leq t<T}\Pi_{\Delta,t-1}\hat{f}(u_t)\right)>\lambda^{4/3}\right]\mathrm{d}\lambda$$

$$\overset{\text{Eq. (45)}}{<}1+\int_1^{+\infty}\frac{\overline{M}}{\lambda^{4/3}}\mathrm{d}\lambda$$

$$=1+3\overline{M}.$$

Next, we take $K\to+\infty$ and apply the *Lebesgue's Monotone Convergence* theorem, which yield

$$\mathbb{E}\left[\left(\sup_{2\leq t<T}\Pi_{\Delta,t-1}\hat{f}(u_t)\right)^{3/4}\right]\leq1+3\overline{M}.$$

By taking $T\to+\infty$ and applying the *Lebesgue's Monotone Convergence* theorem once again, we obtain

$$\mathbb{E}\left[\left(\sup_{t\geq2}\Pi_{\Delta,t-1}\hat{f}(u_t)\right)^{3/4}\right]\leq1+3\overline{M}.$$

Note that for any finite $t$, we have $\Pi_{\Delta,t+1}\geq\Pi_{\Delta,\infty}$ (where $\Pi_{\Delta,\infty}$ is defined in Lemma D.1). Thus, we have

$$\mathbb{E}\left[\Pi_{\Delta,\infty}^{3/4}\left(\sup_{t\geq2}\hat{f}(u_t)\right)^{3/4}\right]\leq\mathbb{E}\left[\left(\sup_{t\geq2}\Pi_{\Delta,t-1}\hat{f}(u_t)\right)^{3/4}\right]\leq1+3\overline{M}.$$

Next, by applying *Hölder's* inequality, we obtain

$$\mathbb{E}\left[\left(\sup_{t\geq2}\hat{f}(u_t)\right)^{1/2}\right]\leq\mathbb{E}^{1/3}\left[\Pi_{\Delta,\infty}^{-3/2}\right]\mathbb{E}^{2/3}\left[\Pi_{\Delta,\infty}^{3/4}\left(\sup_{t\geq2}\hat{f}(u_t)\right)^{3/4}\right]\overset{\text{Lemma D.1}}{\leq}C_{v,d,3/2}^{1/3}(1+3\overline{M})^{2/3}.$$

Then, according to Property 5, we can bound $f(w_t) - f^*$ using $\hat{f}(u_t)$, i.e.,

$$f(w_t) - f^* \leq (L_f + 1)(f(u_t) - f^*) + \frac{(L_f + 1)\beta_1^2}{2(1 - \beta_1)^2} \|\eta_{v_{t-1}} \circ m_{t-1}\|^2 + L_f f^*$$

$$\leq (L_f + 1)\hat{f}(u_t) + \frac{(L_f + 1)\beta_1^2}{2\alpha_1(1 - \beta_1)^2} + L_f f^*.$$

That means

$$\mathbb{E}\left[\left(\sup_{t \geq 2} \left(f(w_t) - f^*\right)\right)^{1/2}\right] \leq \sqrt{L_f + 1} C_{v,d,3/2}^{1/3}(1 + 3\overline{M})^{2/3} + \sqrt{\frac{(L_f + 1)\beta_1^2}{2\alpha_1(1 - \beta_1)^2} + L_f|f^*|}.$$

Finally, according to Lemma C.2, we obtain

$$\mathbb{E}\left[\sup_{t \geq 2} \|\nabla f(w_t)\|\right] \leq \sqrt{2L_f}\,\mathbb{E}\left[\left(\sup_{t \geq 2} \left(f(w_t) - f^*\right)\right)^{1/2}\right]$$

$$< \sqrt{2L_f}\left(\sqrt{L_f + 1} C_{v,d,3/2}^{1/3}(1 + 3\overline{M})^{2/3} + \sqrt{\frac{(L_f + 1)\beta_1^2}{2\alpha_1(1 - \beta_1)^2} + L_f|f^*|}\right).$$

Adding the first term concludes

$$\mathbb{E}\left[\sup_{t \geq 1} \|\nabla f(w_t)\|\right] < \|\nabla f(w_1)\| + \sqrt{2L_f}\left(\sqrt{L_f + 1} C_{v,d,3/2}^{1/3}(1 + 3\overline{M})^{2/3} + \sqrt{\frac{(L_f + 1)\beta_1^2}{2\alpha_1(1 - \beta_1)^2} + L_f|f^*|}\right)$$

$$< +\infty.$$

By combining the almost sure convergence result from Theorem 3.2 with the *Lebesgue's Dominated Convergence* theorem, we obtain the $L_1$ convergence result, namely

$$\lim_{t \to +\infty} \mathbb{E}[\|\nabla f(w_t)\|] = 0. \qquad \square$$

# E. The Proof of Lemma D.2

## E.1. Auxiliary Lemmas for Proving Lemma D.2

**Lemma E.1.** *For any epoch step $t \geq 1$, the following inequality holds*

$$\|m_t\|^2 \leq (1 - \beta_1) \sum_{k=1}^{t} \beta_1^{t-k} \|g_k\|^2, \quad \text{and} \quad \|\eta_{v_t} \circ m_t\|^2 \leq (1 - \beta_1) \sum_{k=1}^{t} \beta_1^{t-k} \|\eta_{v_t} \circ g_k\|^2.$$

*Proof.* Due to Property 4, we have

$$\|m_t\|^2 \leq \beta_1 \|m_{t-1}\|^2 + (1 - \beta_1)\|g_t\|^2.$$

We multiply $1/\beta_1^t$ on the both sides of above inequality and obtain

$$\beta_1^{-t}\|m_t\|^2 \leq \beta_1^{-(t-1)}\|m_{t-1}\|^2 + \beta_1^{-t}(1 - \beta_1)\|g_t\|^2,$$

Iterating the above inequality, we acquire

$$\beta_1^{-t}\|m_t\|^2 \leq \|m_0\|^2 + (1 - \beta_1) \sum_{k=1}^{t} \beta_1^{-k} \|g_k\|^2,$$

that is

$$\|m_t\|^2 \leq \beta_1^t\|m_0\|^2 + (1 - \beta_1) \sum_{k=1}^{t} \beta_1^{t-k} \|g_k\|^2$$

$$= (1 - \beta_1) \sum_{k=1}^{t} \beta_1^{t-k} \|g_k\|^2.$$

Similarly, applying the same approach and noting the monotonicity of $\eta_{v_t,i}$, we obtain

$$\|\eta_{v_t} \circ m_t\|^2 \leq (1 - \beta_1) \sum_{k=1}^{t} \beta_1^{t-k} \|\eta_{v_t} \circ g_k\|^2. \qquad \square$$

**Lemma E.2.** *For any $t \geq 1$ and any positive, monotonically decreasing, adapted process $\{Z(t), \mathscr{F}_{t-1}\}$ with $Z(t) \leq 1$, the following inequality holds*

$$\sum_{i=1}^{d} \sum_{t=1}^{n} (\sqrt{\beta_1})^{n-t} \Delta_{t,i} Z(t) |\nabla_i f(w_t) m_{t,i}| \leq \frac{(1 - \beta_1)(1 - \sqrt{\beta_1})}{8} \sum_{i=1}^{d} \sum_{k=1}^{n} \eta_{v_{k-1},i} Z(k) (\sqrt{\beta_1})^{n-k} (\nabla_i f(w_k))^2$$

$$+ D_1 \sum_{k=1}^{n} Z(k) (\sqrt{\beta_1})^{n-k} \overline{\Delta}_{\beta_1,k} (f(w_k) - f^*)$$

$$+ D_2 \sum_{k=1}^{n} (\sqrt{\beta_1})^{n-k} Z(k) \|\eta_{v_{k-1}} \circ m_{k-1}\|^2$$

$$+ D_3 \sum_{k=1}^{n} Z(k) (\sqrt{\beta_1})^{n-k} \overline{\Delta}_{\sqrt{\beta_1},k}$$

$$+ \sum_{k=1}^{n} N_{n,k},$$

*where*

$$\overline{\Delta}_{\sqrt{\beta_1},k} := \sum_{i=1}^{d} \mathbb{E} \left[ \sum_{t=k}^{+\infty} (\sqrt{\beta_1})^{t-k} \Delta_{t,i} \Big| \mathscr{F}_{k-1} \right],$$

$$N_{n,k} := \sum_{i=1}^{d} \left( \Delta_{\sqrt{\beta_1},k,i} (\sqrt{\beta_1})^{n-k} Z(k) |\nabla_i f(w_k) g_{k,i}| - \mathbb{E} \left[ \Delta_{\sqrt{\beta_1},k,i} (\sqrt{\beta_1})^{n-k} Z(k) |\nabla_i f(w_k) g_{k,i}| \mid \mathscr{F}_{k-1} \right] \right)$$

$$D_1 := \frac{2}{1 - \sqrt{\beta_1}} (A + 2L_f B)(L_f + 1), \ D_2 := \frac{L_f}{1 - \sqrt{\beta_1}}, \ D_3 := \frac{2}{1 - \sqrt{\beta_1}} ((A + 2L_f B)|f^*| + C). \qquad (46)$$

*Proof.* Note that

$$|\nabla_i f(w_t) m_{t,i}| = |\nabla_i f(w_t)(\beta_1 m_{t-1,i} + (1 - \beta_1) g_{t,i})| \leq \beta_1 |\nabla_i f(w_t) m_{t-1,i}| + (1 - \beta_1)|\nabla_i f(w_t) g_{t,i}|$$

$$\leq \beta_1 |\nabla_i f(w_{t-1}) m_{t-1,i}| + \beta_1 |(\nabla_i f(w_t) - \nabla_i f(w_{t-1})) m_{t-1,i}| + (1 - \beta_1)|\nabla_i f(w_t) g_{t,i}|$$

$$\overset{\text{L-smooth}}{\leq} \beta_1 |\nabla_i f(w_{t-1}) m_{t-1,i}| + (1 - \beta_1)|\nabla_i f(w_t) g_{t,i}| + L_f \|\eta_{v_{t-1}} \circ m_{t-1}\| |m_{t-1,i}|.$$

By iterating the above recursive inequality, we obtain

$$|\nabla_i f(w_t) m_{t,i}| \leq (1 - \beta_1) \sum_{k=1}^{t} \beta_1^{t-k} |\nabla_i f(w_k) g_{k,i}| + L_f \sum_{k=1}^{t} \beta_1^{t-k} \|\eta_{v_{k-1}} \circ m_{k-1}\| |m_{k-1,i}|.$$

Then we get that

$$\Delta_{t,i} Z(t) |\nabla_i f(w_t) m_{t,i}| \leq (1 - \beta_1) \Delta_{t,i} \sum_{k=1}^{t} \beta_1^{t-k} Z(k) |\nabla_i f(w_k) g_{k,i}|$$

$$+ L_f \Delta_{t,i} \sum_{k=1}^{t} \beta_1^{t-k} Z(k) \|\eta_{v_{k-1}} \circ m_{k-1}\| |m_{k-1,i}|.$$

Next, we proceed with the calculation.

$$\sum_{t=1}^{n}(\sqrt{\beta_1})^{n-t}\Delta_{t,i}Z(t)|\nabla_i f(w_t)m_{t,i}| \leq (1-\beta_1)\sum_{t=1}^{n}(\sqrt{\beta_1})^{n-t}\Delta_{t,i}\sum_{k=1}^{t}\beta_1^{t-k}Z(k)|\nabla_i f(w_k)g_{k,i}|$$

$$+ L_f\sum_{t=1}^{n}(\sqrt{\beta_1})^{n-t}\Delta_{t,i}\sum_{k=1}^{t}\beta_1^{t-k}Z(k)\|\eta_{v_{k-1}}\circ m_{k-1}\||m_{k-1,i}|$$

$$= (1-\beta_1)\sum_{t=1}^{n}\sum_{k=1}^{t}(\sqrt{\beta_1})^{n-t}\Delta_{t,i}\beta_1^{t-k}Z(k)|\nabla_i f(w_k)g_{k,i}|$$

$$+ L_f\sum_{t=1}^{n}\sum_{k=1}^{t}(\sqrt{\beta_1})^{n-t}\Delta_{t,i}\beta_1^{t-k}Z(k)\|\eta_{v_{k-1}}\circ m_{k-1}\||m_{k-1,i}|$$

$$= (1-\beta_1)\sum_{k=1}^{n}\sum_{t=k}^{n}(\sqrt{\beta_1})^{n-t}\Delta_{t,i}\beta_1^{t-k}Z(k)|\nabla_i f(w_k)g_{k,i}|$$

$$+ L_f\sum_{k=1}^{n}\sum_{t=k}^{n}(\sqrt{\beta_1})^{n-t}\Delta_{t,i}\beta_1^{t-k}Z(k)\|\eta_{v_{k-1}}\circ m_{k-1}\||m_{k-1,i}|$$

$$= (1-\beta_1)\sum_{k=1}^{n}\left(\sum_{t=k}^{n}(\sqrt{\beta_1})^{t-k}\Delta_{t,i}\right)(\sqrt{\beta_1})^{n-k}Z(k)|\nabla_i f(w_k)g_{k,i}|$$

$$+ L_f\sum_{k=1}^{n}\left(\sum_{t=k}^{n}(\sqrt{\beta_1})^{t-k}\Delta_{t,i}\right)(\sqrt{\beta_1})^{n-k}Z(k)\|\eta_{v_{k-1}}\circ m_{k-1}\||m_{k-1,i}|$$

$$< (1-\beta_1)\sum_{k=1}^{n}\left(\underbrace{\sum_{t=k}^{+\infty}(\sqrt{\beta_1})^{t-k}\Delta_{t,i}}_{\Delta_{\sqrt{\beta_1},k,i}}\right)(\sqrt{\beta_1})^{n-k}Z(k)|\nabla_i f(w_k)g_{k,i}|$$

$$+ \frac{L_f}{1-(\sqrt{\beta_1})}\sum_{k=1}^{n}(\sqrt{\beta_1})^{n-k}Z(k)\|\eta_{v_{k-1}}\circ m_{k-1}\||\eta_{v_{k-1},i}m_{k-1,i}|$$

$$= (1-\beta_1)\underbrace{\sum_{k=1}^{n}\Delta_{\sqrt{\beta_1},k,i}(\sqrt{\beta_1})^{n-k}Z(k)|\nabla_i f(w_k)g_{k,i}|}_{\Psi_{n,i}}$$

$$+ \frac{L_f}{1-(\sqrt{\beta_1})}\sum_{k=1}^{n}(\sqrt{\beta_1})^{n-k}Z(k)\|\eta_{v_{k-1}}\circ m_{k-1}\||\eta_{v_{k-1},i}m_{k-1,i}|. \quad (47)$$

Next, we estimate $\Psi_{n,i}$ and obtain

$$\Psi_{n,i} = \sum_{k=1}^{n}\mathbb{E}\left[\Delta_{\sqrt{\beta_1},k,i}(\sqrt{\beta_1})^{n-k}Z(k)|\nabla_i f(w_k)g_{k,i}|\,|\mathscr{F}_{k-1}\right]$$

$$+ \sum_{k=1}^{n}\underbrace{\left(\Delta_{\sqrt{\beta_1},k,i}(\sqrt{\beta_1})^{n-k}Z(k)|\nabla_i f(w_k)g_{k,i}| - \mathbb{E}\left[\Delta_{\sqrt{\beta_1},k,i}(\sqrt{\beta_1})^{n-k}Z(k)|\nabla_i f(w_k)g_{k,i}|\,|\mathscr{F}_{k-1}\right]\right)}_{N_{n,k,i}}$$

$$\overset{AM\text{-}GM}{\leq} \frac{1-\sqrt{\beta_1}}{8}\sum_{k=1}^{n}\eta_{v_{k-1},i}(\sqrt{\beta_1})^{n-k}Z(k)(\nabla_i f(w_k))^2$$

$$+ \frac{2}{1-\sqrt{\beta_1}}\sum_{k=1}^{n}\mathbb{E}^2\left[\sqrt{\Delta_{\sqrt{\beta_1},k,i}}(\sqrt{\beta_1})^{\frac{n-k}{2}}\sqrt{Z(k)}g_{k,i}^2|\mathscr{F}_{k-1}\right] + \sum_{k=1}^{n}N_{n,k,i}$$

$$\leq \frac{1-\sqrt{\beta_1}}{8}\sum_{k=1}^{n}\eta_{v_{k-1},i}Z(k)(\sqrt{\beta_1})^{n-k}(\nabla_i f(w_k))^2$$

$$+\frac{2}{1-\sqrt{\beta_1}}\sum_{k=1}^{n}Z(k)(\sqrt{\beta_1})^{n-k}\mathbb{E}[\Delta_{\sqrt{\beta_1},k,i}|\mathscr{F}_{k-1}]\,\mathbb{E}\left[g_{k,i}^2|\mathscr{F}_{k-1}\right]+\sum_{k=1}^{n}N_{n,k,i}. \tag{48}$$

Summing Equation (47) over the coordinate components $i$, we obtain

$$\sum_{i=1}^{d}\sum_{t=1}^{n}(\sqrt{\beta_1})^{n-t}\Delta_{t,i}Z(t)|\nabla_i f(w_t)m_{t,i}| \leq (1-\beta_1)\sum_{i=1}^{d}\Psi_{n,i}+L_f\sqrt{d}\sum_{k=1}^{n}(\sqrt{\beta_1})^{n-k}Z(k)\|\eta_{v_{k-1}}\circ m_{k-1}\|^2$$

$$\overset{\text{Eq. (48)}}{\leq}\frac{(1-\beta_1)(1-\sqrt{\beta_1})}{8}\sum_{i=1}^{d}\sum_{k=1}^{n}\eta_{v_{k-1},i}Z(k)(\sqrt{\beta_1})^{n-k}(\nabla_i f(w_k))^2$$

$$+\frac{2}{1-\sqrt{\beta_1}}\sum_{i=1}^{d}\sum_{k=1}^{n}Z(k)(\sqrt{\beta_1})^{n-k}\mathbb{E}[\Delta_{\sqrt{\beta_1},k,i}|\mathscr{F}_{k-1}]\,\mathbb{E}\left[g_{k,i}^2|\mathscr{F}_{k-1}\right]$$

$$+\frac{L_f}{1-\sqrt{\beta_1}}\sum_{k=1}^{n}(\sqrt{\beta_1})^{n-k}Z(k)\|\eta_{v_{k-1}}\circ m_{k-1}\|^2+\underbrace{\sum_{i=1}^{d}\sum_{k=1}^{n}N_{n,k,i}}_{\sum_{k=1}^{n}N_{n,k}}$$

$$\leq\frac{(1-\beta_1)(1-\sqrt{\beta_1})}{8}\sum_{i=1}^{d}\sum_{k=1}^{n}\eta_{v_{k-1},i}Z(k)(\sqrt{\beta_1})^{n-k}(\nabla_i f(w_k))^2$$

$$+\frac{2}{1-\sqrt{\beta_1}}\sum_{k=1}^{n}Z(k)(\sqrt{\beta_1})^{n-k}\left(\underbrace{\sum_{i=1}^{d}\mathbb{E}[\Delta_{\sqrt{\beta_1},k,i}|\mathscr{F}_{k-1}]}_{\overline{\Delta}_{\sqrt{\beta_1},k}}\right)\mathbb{E}\left[\|g_k\|^2|\mathscr{F}_{k-1}\right]$$

$$+\frac{L_f}{1-\sqrt{\beta_1}}\sum_{k=1}^{n}(\sqrt{\beta_1})^{n-k}Z(k)\|\eta_{v_{k-1}}\circ m_{k-1}\|^2$$

$$+\sum_{k=1}^{n}N_{n,k}. \tag{49}$$

Noting that

$$\mathbb{E}\left[\|g_k\|^2|\mathscr{F}_{k-1}\right]\overset{\text{Property 1}}{\leq}(A+2L_f B)(f(w_k)-f^*)+C.$$

Using the above inequality to estimate the first term on the right-hand side of Equation (49) yields

$$\sum_{i=1}^{d}\sum_{t=1}^{n}(\sqrt{\beta_1})^{n-t}\Delta_{t,i}Z(t)|\nabla_i f(w_t)m_{t,i}| \leq \frac{1-\beta_1}{8}\sum_{i=1}^{d}\sum_{k=1}^{n}\eta_{v_{k-1},i}Z(k)(\sqrt{\beta_1})^{n-k}(\nabla_i f(w_k))^2$$

$$+D_1\sum_{k=1}^{n}Z(k)(\sqrt{\beta_1})^{n-k}\overline{\Delta}_{\beta_1,k}(f(w_k)-f^*)$$

$$+D_2\sum_{k=1}^{n}(\sqrt{\beta_1})^{n-k}Z(k)\|\eta_{v_{k-1}}\circ m_{k-1}\|^2$$

$$+D_3\sum_{k=1}^{n}Z(k)(\sqrt{\beta_1})^{n-k}\overline{\Delta}_{\sqrt{\beta_1},k}$$

$$+\sum_{k=1}^{n}N_{n,k},$$

where

$$D_1 := \frac{2}{1-\sqrt{\beta_1}}(A+2L_fB)(L_f+1), \ D_2 := \frac{L_f}{1-\sqrt{\beta_1}}, \ D_3 := \frac{2}{1-\sqrt{\beta_1}}\left((A+2L_fB)|f^*|+C\right). \qquad \square$$

**Lemma E.3.** *For any $t \geq 1$, $\varphi > 0$ and any positive, monotonically decreasing, adapted process $\{Z(t), \mathscr{F}_{t-1}\}$ with $Z(t) \leq 1$, $Z(t-1) - Z(t) \leq \varphi\overline{\Delta}_{\sqrt{\beta_1},t}Z(t)$ ($\forall\, t \geq 1$), the following inequality holds.*

$$-\sum_{t=1}^n \sqrt{\beta_1}^{n-t}\sum_{i=1}^d Z(t)\eta_{v_t,i}\nabla_i f(w_t)m_{t,i} \leq -\frac{3(1-\beta_1)}{8}\sum_{i=1}^d\sum_{k=1}^n \eta_{v_{k-1},i}Z(k)(\sqrt{\beta_1})^{n-k}(\nabla_i f(w_k))^2$$

$$+\left(\frac{D_1}{1-\sqrt{\beta_1}}+1\right)\sum_{k=1}^n Z(k)(\sqrt{\beta_1})^{n-k}\overline{\Delta}_{\beta_1,k}(f(w_k)-f^*)$$

$$+\frac{D_2+F_1}{1-\sqrt{\beta_1}}\sum_{k=1}^n(\sqrt{\beta_1})^{n-k}\|\eta_{v_{k-1}}\circ m_{k-1}\|^2$$

$$+\frac{D_3}{1-\sqrt{\beta_1}}\sum_{k=1}^n Z(k)(\sqrt{\beta_1})^{n-k}\overline{\Delta}_{\sqrt{\beta_1},k}$$

$$+\frac{1}{1-\sqrt{\beta_1}}\sum_{k=1}^n N_{n,k}$$

$$+\sum_{t=1}^n(\sqrt{\beta_1})^{n-t}\sum_{k=1}^t \beta_1^{t-k}M'_{k,1,i}, \tag{50}$$

*where*

$$M'_{k,1,i} := (1-\beta_1)Z(k)\eta_{v_{k-1},i}\nabla_i f(w_k)(\nabla_i f(w_k)-g_{k,i}),$$

*and $N_{n,k}$ is defined in Lemma E.2.*

*Proof.* According to the update rule of the Adam algorithm (Equation (2.1)), we derive the following recursive formula

$$-Z(t)\eta_{v_t,i}\nabla_i f(w_t)m_{t,i} = -\beta_1 Z(t)\eta_{v_t,i}\nabla_i f(w_t)m_{t-1,i} + Z(t)\eta_{v_t,i}\nabla_i f(w_t)\left(\beta_1 m_{t-1,i}-m_{t,i}\right)$$

$$= -\beta_1 Z(t)\eta_{v_t,i}\nabla_i f(w_t)m_{t-1,i} - (1-\beta_1)Z(t)\eta_{v_t,i}\nabla_i f(w_t)g_{t,i}$$

$$= -\beta_1(\eta_{v_t,i}Z(t)-\eta_{v_{t-1},i}Z(t-1))f(w_t)m_{t-1,i}$$

$$\quad -\beta_1 Z(t-1)\eta_{v_{t-1},i}\nabla_i f(w_t)m_{t-1,i} - (1-\beta_1)Z(t)\eta_{v_t,i}\nabla_i f(w_t)g_{t,i}$$

$$\overset{(a)}{\leq} -\beta_1(\eta_{v_t,i}-\eta_{v_{t-1},i})Z(t)\nabla_i f(w_t)m_{t-1,i}$$

$$\quad +(Z(t-1)-Z(t))\eta_{v_{t-1},i}|\nabla_i f(w_t)m_{t-1,i}|$$

$$\quad -\beta_1 Z(t-1)\eta_{v_{t-1},i}\nabla_i f(w_{t-1})m_{t-1,i} - (1-\beta_1)Z(t)\eta_{v_t,i}\nabla_i f(w_t)g_{t,i}$$

$$\quad +\beta_1 Z(t-1)\eta_{v_{t-1},i}|\nabla_i f(w_t)-\nabla_i f(w_{t-1})|m_{t-1,i}$$

$$\overset{(b)}{\leq} \underbrace{\beta_1\Delta_{t,i}Z(t)\nabla_i f(w_t)m_{t-1,i}+(1-\beta_i)\Delta_{t,i}Z(t)\nabla_i f(w_t)g_{t,i}}_{=\Delta_{t,i}Z(t)\nabla_i f(w_t)m_{t,i}}$$

$$\quad +\varphi Z(t)\overline{\Delta}_{\sqrt{\beta_1},t}\eta_{v_{t-1},i}|\nabla_i f(w_t)m_{t-1,i}|$$

$$\quad -\beta_1 Z(t-1)\eta_{v_{t-1},i}\nabla_i f(w_{t-1})m_{t-1,i} - (1-\beta_1)Z(t)\eta_{v_{t-1},i}\nabla_i f(w_t)g_{t,i}$$

$$\quad +L_f\|\eta_{v_{t-1}}\circ m_{t-1}\||\eta_{v_{t-1},i}m_{t-1,i}|$$

$$\overset{(c)}{\leq} \Delta_{t,i}Z(t)|\nabla_i f(w_t)m_{t,i}|$$

$$\quad +\frac{1}{2L_f}Z(t)\overline{\Delta}_{\sqrt{\beta_1},t}(\nabla_i f(w_t))^2+\frac{\varphi^2 L_f}{2\sqrt{v}}\eta_{v_{t-1},i}^2 m_{t-1,i}^2$$

$$\quad -\beta_1 Z(t-1)\eta_{v_{t-1},i}\nabla_i f(w_{t-1})m_{t-1,i} - (1-\beta_1)Z(t)\eta_{v_{t-1},i}(\nabla_i f(w_t))^2$$

$$+ L_f \|\eta_{v_{t-1}} \circ m_{t-1}\| |\eta_{v_{t-1},i} m_{t-1,i}|$$
$$+ \underbrace{(1 - \beta_1) Z(t) \eta_{v_{t-1},i} \nabla_i f(w_t)(\nabla_i f(w_t) - g_{t,i})}_{M'_{t,1,i}}.$$

In step $(a)$, we apply the following substitution

$$\eta_{v_t,i} Z(t) - \eta_{v_{t-1},i} Z(t-1) = (\eta_{v_t,i} - \eta_{v_{t-1},i}) Z(t) - \eta_{v_{t-1},i}(Z(t) - Z(t-1)).$$

In step $(b)$, we first apply a transformation to the fourth term from the previous step, denoted by

$$-(1 - \beta_1) Z(t) \eta_{v_t,i} \nabla_i f(w_t) g_{t,i} = -(1 - \beta_1) Z(t) \eta_{v_{t-1},i} \nabla_i f(w_t) g_{t,i} + (1 - \beta_1) \Delta_{t,i} Z(t) \nabla_i f(w_t) g_{t,i}.$$

Next, we combine the second term of this transformation with the first term from the prior step of Step $(b)$ in order to obtain

$$\beta_1 \Delta_{t,i} Z(t) \nabla_i f(w_t) m_{t,i}.$$

We then use the inequality $Z(t-1) - Z(t) \leq \varphi \overline{\Delta}_{\sqrt{\beta_1},t} Z(t)$.

Finally, in step $(c)$, we begin by applying an absolute value bound to the first term from the previous step:

$$\beta_1 \Delta_{t,i} Z(t) \nabla_i f(w_t) m_{t,i} \leq \beta_1 \Delta_{t,i} Z(t) |\nabla_i f(w_t) m_{t,i}|.$$

Next, for the second term in the previous step, we use the following application of the *AM-GM* inequality

$$\varphi Z(t) \overline{\Delta}_{\sqrt{\beta_1},t} \eta_{v_{t-1},i} |\nabla_i f(w_t) m_{t-1,i}| \leq \frac{1}{2L_f} Z(t) \overline{\Delta}_{\sqrt{\beta_1},t} (\nabla_i f(w_t))^2 + \frac{\varphi^2 L_f}{2\sqrt{v}} \eta^2_{v_{t-1},i} m^2_{t-1,i}.$$

Summing both sides of the above inequality over the coordinate components $i$ and applying the arithmetic mean inequality, we obtain

$$-\sum_{i=1}^{d} Z(t) \eta_{v_t,i} \nabla_i f(w_t) m_{t,i} \leq \beta_1 \sum_{i=1}^{d} -Z(t-1) \eta_{v_{t-1},i} \nabla_i f(w_{t-1}) m_{t-1,i} + \frac{1}{2L_f} Z(t) \overline{\Delta}_{\sqrt{\beta_1},t} \|\nabla f(w_t)\|^2$$
$$+ \sum_{i=1}^{d} \Delta_{t,i} Z(t) |\nabla_i f(w_t) m_{t,i}| + F_1 \|\eta_{v_{t-1}} \circ m_{t-1}\|^2 + \sum_{i=1}^{d} M'_{t,1,i}$$
$$- (1 - \beta_1) \sum_{i=1}^{d} Z(t) \eta_{v_{t-1},i} (\nabla_i f(w_t))^2$$
$$\stackrel{\text{Lemma C.2}}{\leq} \beta_1 \sum_{i=1}^{d} -Z(t-1) \eta_{v_{t-1},i} \nabla_i f(w_{t-1}) m_{t-1,i} + Z(t) \overline{\Delta}_{\sqrt{\beta_1},t} (f(w_t) - f^*)$$
$$+ \sum_{i=1}^{d} \Delta_{t,i} Z(t) |\nabla_i f(w_t) m_{t,i}| + F_1 \|\eta_{v_{t-1}} \circ m_{t-1}\|^2 + \sum_{i=1}^{d} M'_{t,1,i}$$
$$- (1 - \beta_1) \sum_{i=1}^{d} Z(t) \eta_{v_{t-1},i} (\nabla_i f(w_t))^2,$$

where

$$F_1 := \sqrt{d} L_f + \frac{k^2 L_f}{2\sqrt{v}}.$$

By iterating the above inequality, we obtain

$$-\sum_{i=1}^{d} Z(t) \eta_{v_t,i} \nabla_i f(w_t) m_{t,i} \leq \sum_{k=1}^{t} \beta_1^{t-k} \sum_{i=1}^{d} \Delta_{k,i} Z(k) |\nabla_i f(w_k) m_{k,i}| + \sum_{i=1}^{d} Z(t) \overline{\Delta}_{\sqrt{\beta_1},t} (f(w_t) - f^*)$$

$$+ F_1 \sum_{k=1}^{t} \beta_1^{t-k} \|\eta_{v_{k-1}} \circ m_{k-1}\|^2 + \sum_{k=1}^{t} \beta_1^{t-k} M'_{k,1,i}$$

$$- (1 - \beta_1) \sum_{k=1}^{t} \beta_1^{t-k} Z(k) \eta_{v_{k-1},i} (\nabla_i f(w_k))^2.$$

We further obtain

$$-\sum_{t=1}^{n} (\sqrt{\beta_1})^{n-t} \sum_{i=1}^{d} Z(t) \eta_{v_t,i} \nabla_i f(w_t) m_{t,i} \leq \sum_{t=1}^{n} \sqrt{\beta_1}^{n-t} \sum_{k=1}^{t} \beta_1^{t-k} \sum_{i=1}^{d} \Delta_{k,i} Z(k) |\nabla_i f(w_k) m_{k,i}|$$

$$+ F_1 \sum_{t=1}^{n} \sqrt{\beta_1}^{n-t} \sum_{k=1}^{t} \beta_1^{t-k} \|\eta_{v_{k-1}} \circ m_{k-1}\|^2$$

$$+ \sum_{t=1}^{n} (\sqrt{\beta_1})^{n-t} \sum_{k=1}^{t} \beta_1^{t-k} M'_{k,1,i}$$

$$+ \sum_{t=1}^{n} (\sqrt{\beta_1})^{n-t} \sum_{i=1}^{d} Z(t) \overline{\Delta}_{\sqrt{\beta_1},t} (f(w_t) - f^*)$$

$$- (1 - \beta_1) \sum_{t=1}^{n} \sqrt{\beta_1}^{n-t} \sum_{k=1}^{t} \beta_1^{t-k} Z(k) \eta_{v_{k-1},i} (\nabla_i f(w_k))^2$$

$$\overset{\text{Lemma C.1}}{\leq} \underbrace{\frac{1}{1 - \sqrt{\beta_1}} \sum_{k=1}^{n} (\sqrt{\beta_1})^{n-k} \Delta_{k,i} Z(k) |\nabla_i f(w_k) m_{k,i}|}_{\Phi_{n,1}}$$

$$+ \frac{F_1}{1 - \sqrt{\beta_1}} \sum_{k=1}^{n} (\sqrt{\beta_1})^{n-k} \|\eta_{v_{k-1}} \circ m_{k-1}\|^2$$

$$+ \sum_{t=1}^{n} (\sqrt{\beta_1})^{n-t} \sum_{k=1}^{t} \beta_1^{t-k} M'_{k,1,i}$$

$$+ \sum_{t=1}^{n} (\sqrt{\beta_1})^{n-t} \sum_{i=1}^{d} Z(t) \overline{\Delta}_{\sqrt{\beta_1},t} (f(w_t) - f^*)$$

$$- (1 - \beta_1) \sum_{k=1}^{n} (\sqrt{\beta_1})^{n-k} Z(k) \eta_{v_{k-1},i} (\nabla_i f(w_k))^2. \tag{51}$$

Now, we focus on estimating the term $\Phi_{n,1}$. By applying Lemma E.2, we obtain

$$\frac{1}{1 - \sqrt{\beta_1}} \sum_{i=1}^{d} \sum_{t=1}^{n} (\sqrt{\beta_1})^{n-t} \Delta_{t,i} Z(t) |\nabla_i f(w_t) m_{t,i}| \leq \frac{1 - \beta_1}{8} \sum_{i=1}^{d} \sum_{k=1}^{n} \eta_{v_{k-1},i} Z(k) (\sqrt{\beta_1})^{n-k} (\nabla_i f(w_k))^2$$

$$+ \frac{D_1}{1 - \sqrt{\beta_1}} \sum_{k=1}^{n} Z(k) (\sqrt{\beta_1})^{n-k} \overline{\Delta}_{\beta_1,k} (f(w_k) - f^*)$$

$$+ \frac{D_2}{1 - \sqrt{\beta_1}} \sum_{k=1}^{n} (\sqrt{\beta_1})^{n-k} Z(k) \|\eta_{v_{k-1}} \circ m_{k-1}\|^2$$

$$+ \frac{D_3}{1 - \sqrt{\beta_1}} \sum_{k=1}^{n} Z(k) (\sqrt{\beta_1})^{n-k} \overline{\Delta}_{\sqrt{\beta_1},k}$$

$$+ \frac{1}{1 - \sqrt{\beta_1}} \sum_{k=1}^{n} N_{n,k},$$

where $D_1$, $D_2$, $D_3$, $\overline{\Delta}_{\sqrt{\beta_1},k}$ are defined in Equation (46). Substituting the above estimate for $\Phi_{n,1}$ back into Equation (51), we obtain

$$
\begin{aligned}
-\sum_{t=1}^{n}\sqrt{\beta_1}^{n-t}\sum_{i=1}^{d}Z(t)\eta_{v_t,i}\nabla_i f(w_t)m_{t,i} \leq\ & -\frac{3(1-\beta_1)}{8}\sum_{i=1}^{d}\sum_{k=1}^{n}\eta_{v_{k-1},i}Z(k)(\sqrt{\beta_1})^{n-k}(\nabla_i f(w_k))^2 \\
& +\left(\frac{D_1}{1-\sqrt{\beta_1}}+1\right)\sum_{k=1}^{n}Z(k)(\sqrt{\beta_1})^{n-k}\overline{\Delta}_{\beta_1,k}(f(w_k)-f^*) \\
& +\frac{D_2+F_1}{1-\sqrt{\beta_1}}\sum_{k=1}^{n}(\sqrt{\beta_1})^{n-k}\|\eta_{v_{k-1}}\circ m_{k-1}\|^2 \\
& +\frac{D_3}{1-\sqrt{\beta_1}}\sum_{k=1}^{n}Z(k)(\sqrt{\beta_1})^{n-k}\overline{\Delta}_{\sqrt{\beta_1},k} \\
& +\frac{1}{1-\sqrt{\beta_1}}\sum_{k=1}^{n}N_{n,k} \\
& +\sum_{t=1}^{n}(\sqrt{\beta_1})^{n-t}\sum_{k=1}^{t}\beta_1^{t-k}M'_{k,1,i}. \qquad \square
\end{aligned}
$$

### E.2. Proof of Lemma D.2

*Proof.* First, we compute $f(w_{t+1})-f(w_t)$. Based on the $L$-smoothness condition, we make the following estimate

$$
\begin{aligned}
f(w_{t+1})-f(w_t) &\leq \nabla f(w_t)^\top(w_{t+1}-w_t)+\frac{L_f}{2}\|w_{t+1}-w_t\|^2 \\
&= -\sum_{i=1}^{d}\eta_{v_t,i}\nabla_i f(w_t)m_{t,i}+\frac{L_f}{2}\|\eta_{v_t}\circ m_t\|^2. \qquad (52)
\end{aligned}
$$

Next, we construct $\Pi_{\Delta,t}$, which is defined as follows

$$
\Pi_{\Delta,t}:=\prod_{k=1}^{t}\left(1+\left(\frac{D_1}{1-\sqrt{\beta_1}}+1\right)\overline{\Delta}_{\sqrt{\beta_1},k}\right)^{-1}\ (t\geq 1),\ \Pi_{\Delta,0}:=1,
$$

where $D_1,\overline{\Delta}_{\sqrt{\beta_1},k}$ are defined in Equation (46). Note that the $\Pi_{\Delta,t}$ here is a specific $Z(t)$ used in Lemma E.2 and Lemma E.3 with $\varphi=\frac{D_1}{1-\sqrt{\beta_1}}+1$. We can subsequently apply the results from Lemma E.2 and Lemma E.3. We multiply both sides of Equation (52) by this specific $\Pi_{\Delta,t}$ and, noting its monotonically decreasing property, we obtain

$$
\Pi_{\Delta,t+1}(f(w_{t+1})-f^*)-\Pi_{\Delta,t}(f(w_t)-f^*)\leq-\sum_{i=1}^{d}\Pi_{\Delta,t}\eta_{v_t,i}\nabla_i f(w_t)m_{t,i}+\frac{L_f}{2}\|\eta_{v_t}\circ m_t\|^2.
$$

Next, we compute

$$
\sum_{t=1}^{n}(\sqrt{\beta_1})^{n-t}\left(\Pi_{\Delta,t+1}(f(w_{t+1})-f^*)-\Pi_{\Delta,t}(f(w_t)-f^*)\right).
$$

We have

$$
\begin{aligned}
&\sum_{t=1}^{n}(\sqrt{\beta_1})^{n-t}\left(\Pi_{\Delta,t+1}(f(w_{t+1})-f^*)-\Pi_{\Delta,t}(f(w_t)-f^*)\right) \\
\leq & -\sum_{t=1}^{n}(\sqrt{\beta_1})^{n-t}\sum_{i=1}^{d}\Pi_{\Delta,t}\eta_{v_t,i}\nabla_i f(w_t)m_{t,i}+\frac{L_f}{2}\sum_{t=1}^{n}(\sqrt{\beta_1})^{n-t}\sum_{i=1}^{d}\|\eta_{v_t}\circ m_t\|^2 \\
\overset{\text{Lemma E.3}}{\leq} & -\frac{3(1-\beta_1)}{8}\sum_{i=1}^{d}\sum_{t=1}^{n}\eta_{v_{t-1},i}\Pi_{\Delta,t}(\sqrt{\beta_1})^{n-t}(\nabla_i f(w_t))^2
\end{aligned}
$$

$$+ \left( \frac{D_1}{1 - \sqrt{\beta_1}} + 1 \right) \sum_{t=1}^{n} \Pi_{\Delta,t} (\sqrt{\beta_1})^{n-t} \overline{\Delta}_{\beta_1,t} (f(w_t) - f^*)$$

$$+ \left( \frac{D_2 + F_1}{\sqrt{\beta_1}(1 - \sqrt{\beta_1})} + \frac{L_f}{2} \right) \sum_{t=1}^{n} (\sqrt{\beta_1})^{n-t} \| \eta_{v_t} \circ m_t \|^2$$

$$+ \frac{D_3}{1 - \sqrt{\beta_1}} \sum_{t=1}^{n} \Pi_{\Delta,t} (\sqrt{\beta_1})^{n-t} \overline{\Delta}_{\sqrt{\beta_1},k} + \frac{1}{1 - \sqrt{\beta_1}} \sum_{t=1}^{n} N_{n,t}$$

$$+ \sum_{t=1}^{n} (\sqrt{\beta_1})^{n-t} \sum_{k=1}^{t} \beta_1^{t-k} M'_{k,1,i}. \tag{53}$$

We then observe that the left side of the above inequality can be rewritten as

$$\sum_{t=1}^{n} (\sqrt{\beta_1})^{n-t} \left( \Pi_{\Delta,t+1} (f(w_{t+1}) - f^*) - \Pi_{\Delta,t} (f(w_t) - f^*) \right)$$

$$= \sum_{t=1}^{n} (\sqrt{\beta_1})^{n-t} \Pi_{\Delta,t+1} (f(w_{t+1}) - f^*) - \sum_{t=1}^{n} (\sqrt{\beta_1})^{n-t} \Pi_{\Delta,t} (f(w_t) - f^*)$$

$$= \sum_{t=1}^{n} (\sqrt{\beta_1})^{(n+1)-(t+1)} \Pi_{\Delta,t+1} (f(w_{t+1}) - f^*) - \sum_{t=1}^{n} (\sqrt{\beta_1})^{n-t} \Pi_{\Delta,t} (f(w_t) - f^*)$$

$$= \sum_{t=2}^{n+1} (\sqrt{\beta_1})^{(n+1)-t} \Pi_{\Delta,t} (f(w_t) - f^*) - \sum_{t=1}^{n} (\sqrt{\beta_1})^{n-t} \Pi_{\Delta,t} (f(w_t) - f^*)$$

$$= -(\sqrt{\beta_1})^n (f(w_1) - f^*)$$

$$+ \underbrace{\sum_{t=1}^{n+1} (\sqrt{\beta_1})^{(n+1)-t} \Pi_{\Delta,t} (f(w_t) - f^*)}_{F_{n+1}} - \underbrace{\sum_{t=1}^{n} (\sqrt{\beta_1})^{n-t} \Pi_{\Delta,t} (f(w_t) - f^*)}_{F'_n}.$$

Substituting the above transformation back to Equation (53), we obtain

$$F_{n+1} - \left( F'_n + \left( \frac{D_1}{1 - \sqrt{\beta_1}} + 1 \right) \sum_{t=1}^{n} \Pi_{\Delta,t} (\sqrt{\beta_1})^{n-t} \overline{\Delta}_{\beta_1,t} (f(w_t) - f^*) \right)$$

$$\leq (\sqrt{\beta_1})^n (f(w_1) - f^*) - \frac{3(1 - \beta_1)}{8} \sum_{i=1}^{d} \sum_{t=1}^{n} \eta_{v_{t-1},i} \Pi_{\Delta,t} (\sqrt{\beta_1})^{n-t} (\nabla_i f(w_t))^2$$

$$+ \left( \frac{D_2 + F_1}{\sqrt{\beta_1}(1 - \sqrt{\beta_1})} + \frac{L_f}{2} \right) \sum_{t=1}^{n} (\sqrt{\beta_1})^{n-t} \| \eta_{v_t} \circ m_t \|^2$$

$$+ \frac{D_3}{1 - \sqrt{\beta_1}} \sum_{t=1}^{n} \Pi_{\Delta,t} (\sqrt{\beta_1})^{n-t} \overline{\Delta}_{\sqrt{\beta_1},k} + \frac{1}{1 - \sqrt{\beta_1}} \sum_{t=1}^{n} N_{n,t}$$

$$+ \sum_{t=1}^{n} (\sqrt{\beta_1})^{n-t} \sum_{k=1}^{t} \beta_1^{t-k} M'_{k,1,i}.$$

Observe that

$$F'_n + \left( \frac{D_1}{1 - \sqrt{\beta_1}} + 1 \right) \sum_{t=1}^{n} \Pi_{\Delta,t} (\sqrt{\beta_1})^{n-t} \overline{\Delta}_{\beta_1,t} (f(w_t) - f^*)$$

$$= \sum_{t=1}^{n} (\sqrt{\beta_1})^{n-t} \left( 1 + \left( \frac{D_1}{1 - \sqrt{\beta_1}} + 1 \right) \overline{\Delta}_{\beta_1,t} \right) \Pi_{\Delta,t} (f(w_t) - f^*)$$

$$= \sum_{t=1}^{n} (\sqrt{\beta_1})^{n-t} \Pi_{\Delta, t-1}(f(w_t) - f^*) = F_n.$$

We get

$$F_{n+1} - F_n$$

$$\leq (\sqrt{\beta_1})^n (f(w_1) - f^*) - \frac{3(1-\beta_1)}{8} \sum_{i=1}^{d} \sum_{t=1}^{n} \eta_{v_{t-1},i} \Pi_{\Delta,t} (\sqrt{\beta_1})^{n-t} (\nabla_i f(w_t))^2$$

$$+ \left( \frac{D_2 + F_1}{\sqrt{\beta_1}(1 - \sqrt{\beta_1})} + \frac{L_f}{2} \right) \sum_{t=1}^{n} (\sqrt{\beta_1})^{n-t} \|\eta_{v_t} \circ m_t\|^2$$

$$+ \frac{D_3}{1 - \sqrt{\beta_1}} \sum_{t=1}^{n} \Pi_{\Delta,t} (\sqrt{\beta_1})^{n-t} \overline{\Delta}_{\sqrt{\beta_1},k} + \frac{1}{1 - \sqrt{\beta_1}} \sum_{t=1}^{n} N_{n,t}$$

$$+ \sum_{t=1}^{n} (\sqrt{\beta_1})^{n-t} \sum_{k=1}^{t} \beta_1^{t-k} M'_{k,1,i}.$$

Next, we take the expectation on both sides of the above inequality and note that $\mathbb{E}[N_{n,t}] = \mathbb{E}[M'_{k,1,i}] = 0$. We then obtain

$$\mathbb{E}[F_{n+1}] - \mathbb{E}[F_n]$$

$$\leq (\sqrt{\beta_1})^n (f(w_1) - f^*) - \frac{3(1-\beta_1)}{8} \sum_{i=1}^{d} \sum_{t=1}^{n} (\sqrt{\beta_1})^{n-t} \mathbb{E}\left[ \eta_{v_{t-1},i} \Pi_{\Delta,t} (\nabla_i f(w_t))^2 \right]$$

$$+ \left( \frac{D_2 + F_1}{\sqrt{\beta_1}(1 - \sqrt{\beta_1})} + \frac{L_f}{2} \right) \sum_{t=1}^{n} (\sqrt{\beta_1})^{n-t} \mathbb{E}\left[ \|\eta_{v_t} \circ m_t\|^2 \right]$$

$$+ \frac{D_3}{1 - \sqrt{\beta_1}} \sum_{t=1}^{n} (\sqrt{\beta_1})^{n-t} \mathbb{E}\left[ \overline{\Delta}_{\sqrt{\beta_1},k} \right] + 0 + 0.$$

Summing both sides of the above inequality over the index $n$ from $1$ to $T$, we obtain

$$\mathbb{E}[F_{T+1}] - \mathbb{E}[F_1]$$

$$\leq \frac{1}{1 - \sqrt{\beta_1}}(f(w_1) - f^*) - \frac{3(1-\beta_1)}{8} \sum_{n=1}^{T} \sum_{i=1}^{d} \sum_{t=1}^{n} (\sqrt{\beta_1})^{n-t} \mathbb{E}\left[ \eta_{v_{t-1},i} \Pi_{\Delta,t} (\nabla_i f(w_t))^2 \right]$$

$$+ \left( \frac{D_2 + F_1}{\sqrt{\beta_1}(1 - \sqrt{\beta_1})} + \frac{L_f}{2} \right) \sum_{n=1}^{T} \sum_{t=1}^{n} (\sqrt{\beta_1})^{n-t} \mathbb{E}\left[ \|\eta_{v_t} \circ m_t\|^2 \right]$$

$$+ \frac{D_3}{1 - \sqrt{\beta_1}} \sum_{n=1}^{T} \sum_{t=1}^{n} (\sqrt{\beta_1})^{n-t} \mathbb{E}\left[ \overline{\Delta}_{\sqrt{\beta_1},k} \right]$$

$$\overset{\text{Lemma C.1}}{\leq} \frac{1}{1 - \sqrt{\beta_1}}(f(w_1) - f^*) - \frac{3(1-\beta_1)}{8} \sum_{n=1}^{T} \sum_{i=1}^{d} \mathbb{E}\left[ \eta_{v_{n-1},i} \Pi_{\Delta,n} (\nabla_i f(w_n))^2 \right]$$

$$+ \frac{1}{1 - \sqrt{\beta_1}} \left( \frac{D_2 + F_1}{\sqrt{\beta_1}(1 - \sqrt{\beta_1})} + \frac{L_f}{2} \right) \sum_{n=1}^{T} \mathbb{E}\left[ \|\eta_{v_n} \circ m_n\|^2 \right] + \frac{D_3}{\sqrt{v}(1 - \sqrt{\beta_1})^2}.$$

To maintain consistency with the notation in the subsequent proofs, we replace the index $n$ with $t$ in the summation $\sum_{n=1}^{T}$ on the right side of the above inequality as follows.

$$\mathbb{E}[F_{T+1}] - \mathbb{E}[F_1]$$

$$\overset{\text{Lemma C.1}}{\leq} \frac{1}{1 - \sqrt{\beta_1}}(f(w_1) - f^*) - \frac{3(1-\beta_1)}{8} \sum_{t=1}^{T} \sum_{i=1}^{d} \mathbb{E}\left[ \eta_{v_{t-1},i} \Pi_{\Delta,t} (\nabla_i f(w_t))^2 \right]$$

$$+ \frac{1}{1 - \sqrt{\beta_1}} \left( \frac{D_2 + F_1}{\sqrt{\beta_1}(1 - \sqrt{\beta_1})} + \frac{L_f}{2} \right) \sum_{t=1}^{T} \mathbb{E}\left[ \|\eta_{v_t} \circ m_t\|^2 \right] + \frac{D_3}{\sqrt{v}(1 - \sqrt{\beta_1})^2}. \tag{54}$$

Using Lemma E.1, we can transform the third term on the right side of the above inequality as follows

$$\sum_{t=1}^{T} \mathbb{E}\left[ \|\eta_{v_t} \circ m_t\|^2 \right] \le (1 - \beta_1) \sum_{t=1}^{T} \sum_{k=1}^{t} \beta_1^{t-k} \mathbb{E}\|\eta_{v_t} \circ g_k\|^2 \overset{\text{Lemma C.1}}{\le} \sum_{t=1}^{T} \mathbb{E}\|\eta_{v_t} \circ g_t\|^2.$$

Substituting this back into Equation (54) and rearranging the terms, we obtain the following two inequalities.

$$\mathbb{E}[\Pi_{\Delta,t}(f(w_t) - f^*)] \le \mathbb{E}[F_1] + \frac{1}{1 - \sqrt{\beta_1}}(f(w_1) - f^*) + \frac{D_3}{\sqrt{v}(1 - \sqrt{\beta_1})^2}$$

$$+ \frac{1}{1 - \sqrt{\beta_1}} \left( \frac{D_2 + F_1}{\sqrt{\beta_1}(1 - \sqrt{\beta_1})} + \frac{L_f}{2} \right) \sum_{t=1}^{T} \mathbb{E}\|\eta_{v_t} \circ g_t\|^2, \tag{55}$$

and

$$\sum_{t=1}^{T} \sum_{i=1}^{d} \mathbb{E}\left[ \eta_{v_{t-1},i} \Pi_{\Delta,t}(\nabla_i f(w_t))^2 \right] \le \frac{8}{3(1 - \beta_1)} \mathbb{E}[F_1] + \frac{8}{3(1 - \beta_1)} \frac{1}{1 - \sqrt{\beta_1}}(f(w_1) - f^*)$$

$$+ \frac{D_3}{\sqrt{v}(1 - \sqrt{\beta_1})^2} + \frac{8}{(1 - \sqrt{\beta_1})3(1 - \beta_1)} \left( \frac{D_2 + F_1}{\sqrt{\beta_1}(1 - \sqrt{\beta_1})} + \frac{L_f}{2} \right) \sum_{t=1}^{T} \mathbb{E}\|\eta_{v_t} \circ g_t\|^2. \tag{56}$$

Next, we proceed to estimate

$$\sum_{t=1}^{T} \sum_{i=1}^{d} \mathbb{E}\left[ \Pi_{\Delta,t} \Delta_{t,i} |\nabla_i f(u_t) m_{t-1,i}| \right].$$

We have

$$\sum_{t=1}^{T} \sum_{i=1}^{d} \mathbb{E}\left[ \Pi_{\Delta,t} \Delta_{t,i} |\nabla_i f(u_t) m_{t-1,i}| \right]$$

$$\overset{\text{Cauchy-Schwarz inequality}}{\le} \sum_{i=1}^{d} \sum_{t=1}^{T} \sqrt{\mathbb{E}\left[ \Delta_{t,i}^2 \Pi_{\Delta,t}(\nabla_i f(u_t))^2 \right]} \cdot \sqrt{\mathbb{E}\left[ \Pi_{\Delta,t} m_{t-1,i} \right]}$$

$$\le \frac{1}{\sqrt{v}} \sum_{i=1}^{d} \sum_{t=1}^{T} \sqrt{\mathbb{E}\left[ \eta_{v_{t-1},i} \Pi_{\Delta,t}(\nabla_i f(u_t))^2 \right]} \cdot \sqrt{\mathbb{E}\left[ \Pi_{\Delta,t} m_{t-1,i}^2 \right]}$$

$$\overset{\text{Cauchy-Schwarz inequality}}{\le} \frac{1}{\sqrt{v}} \sum_{i=1}^{d} \sqrt{\sum_{t=1}^{T} \mathbb{E}\left[ \eta_{v_{t-1},i} \Pi_{\Delta,t}(\nabla_i f(u_t))^2 \right]} \cdot \sqrt{\sum_{t=1}^{T} \mathbb{E}\left[ \Pi_{\Delta,t} m_{t-1,i}^2 \right]}$$

$$\overset{\text{Cauchy-Schwarz inequality}}{\le} \frac{1}{\sqrt{v}} \sqrt{\sum_{i=1}^{d} \sum_{t=1}^{T} \mathbb{E}\left[ \eta_{v_{t-1},i} \Pi_{\Delta,t}(\nabla_i f(u_t))^2 \right]} \cdot \sqrt{\sum_{t=1}^{T} \mathbb{E}\left[ \Pi_{\Delta,t} \|m_{t-1}\|^2 \right]}$$

$$\overset{\text{Lemma E.1}}{\le} \frac{\sqrt{1 - \beta_1}}{\sqrt{v}} \sqrt{\sum_{i=1}^{d} \sum_{t=1}^{T} \mathbb{E}\left[ \eta_{v_{t-1},i} \Pi_{\Delta,t}(\nabla_i f(u_t))^2 \right]} \cdot \sqrt{\sum_{t=1}^{T} \sum_{k=1}^{t} \beta_1^{t-k} \mathbb{E}[\Pi_{\Delta,t} \|g_k\|^2]}$$

$$\overset{(a)}{\le} \mathcal{O}\left( \sum_{t=1}^{T} \mathbb{E}[\|\eta_{v_t} \circ g_t\|^2] \right) + \mathcal{O}(1).$$

In the final step $(a)$, we first apply Property 1 to bound $\mathbb{E}[\Pi_{\Delta,t}\|g_t\|^2]$ for all $t \le T$, by $\mathbb{E}[\Pi_{\Delta,t}(f(w_t) - f^*)]$ for $t \le T$. Then, using Equation (55), we further bound $\mathbb{E}[\Pi_{\Delta,t}(f(w_t) - f^*)]$ as $\mathcal{O}\left( \sum_{t=1}^{T} \|\eta_{v_t} \circ g_t\|^2 \right)$. Finally, we apply Equation (56) to the previous summation. This completes the proof. □

