# OpenReview forum: "A Comprehensive Framework for Analyzing the Convergence of Adam: Bridging the Gap with SGD"
_ICML.cc/2025/Conference — ICML 2025 poster_

### Official Review · Reviewer_LTUb · 2025-03-11

**Overall Recommendation:** 4

**Summary:**

The authors propose a new theoretical framework with fairly weak assumptions, within which they are able to establish convergence rates for Adam.

**Claims And Evidence:**

N/A

**Essential References Not Discussed:**

N/A

**Experimental Designs Or Analyses:**

N/A

**Methods And Evaluation Criteria:**

No simulation sutdy nor application on real datasets.

**Other Comments Or Suggestions:**

No comments or suggestions.

**Other Strengths And Weaknesses:**

Strengths:
The authors successfully obtain results for Adam under remarkably weak assumptions (smoothness and ABC inequality).
Given the short time available, it was not possible to fully and rigorously review all the proofs. However, based on what I was able to check, the proofs appear to be correct and well-detailed.
I also appreciate the effort made to enhance the readability of the proofs, particularly through the use of the dependency graph.

Weaknesses:
While it is now widely accepted that simulations are not strictly necessary to demonstrate that Adam works, it would have been valuable to present an application example that could not have been theoretically addressed by previous works but can now be handled before moving on to simulations.

**Questions For Authors:**

No question.

**Relation To Broader Scientific Literature:**

The main contribution rely on a new set of  weak assumptions to obtain theoretical results for ADAM algorithm. More precisely, they obtain analogous rate of convergence as in the literature, but with weaker conditions.

**Theoretical Claims:**

Given the short time available, it was not possible to fully and rigorously review all the proofs. However, based on what I was able to check, the proofs appear to be correct.

---

> ### Author Rebuttal · Authors · 2025-03-31
>
> Dear Reviewer LTUb,
>
> We sincerely appreciate your thorough evaluation of our manuscript and your positive feedback. Your recognition of our theoretical framework and the establishment of convergence rates for Adam under weak assumptions is highly encouraging.
>
> We acknowledge your suggestion to include an application example demonstrating the practical implications of our theoretical findings. While our primary focus has been on the theoretical aspects, we understand the value of illustrating how our results can address scenarios previously unmanageable by earlier works. In response, we plan to incorporate a relevant application example in our revised manuscript to highlight the practical applicability of our theoretical contributions.
>
> Thank you once again for your insightful comments and for your recommendation to accept our work. Your feedback has been instrumental in enhancing the quality and impact of our manuscript.
>
> Sincerely,
>
> Authors of Paper 1314

---

### Official Review · Reviewer_nypN · 2025-03-11

**Overall Recommendation:** 3

**Summary:**

The paper studies the convergence properties of Adam under smooth nonconvex settings. The paper presents convergence results in the sense of almost sure, $L_1$ and non-asymptotic, under relaxed noise assumption, i.e. the ABC inequality. The non-asymptotic convergence result is in the order of $O(1/\sqrt{T})$, which is generally consistent with that of SGD.

**Claims And Evidence:**

Most of the claims are generally clear.
1. I do have a question for Theorem 3.1. According to Theorem 3.1, the notation $O()$ doesn't omit the dpendence on dimensionality $d$. I am wondering if this is possible with your assumptions? Or you just simply miss this?
2. Also regarding $O()$ in Theorem 3.1, could you please also specify the dependence on $1-\beta_1$?

**Essential References Not Discussed:**

No.

**Experimental Designs Or Analyses:**

No experiments in the paper.

**Methods And Evaluation Criteria:**

No experiments in the paper.

**Other Comments Or Suggestions:**

1. For Theorem 3.1, it seems kind of weird to state the convergence result in high-probability form, since it has a dependence on the probability as $O(1/s^2)$, while standard high-probability convergence results are usually in the order of $O(log(1/s))$. I think just equation (42) is good enough for the statement.

2. I suggest the authors to distribute some space at least in the appendix to aggregate the definitions for the values. It's really hard to follow the proof or even find out the detailed results of the theorems with a lot of defined values like $C,C_1,...$ with very seperate definitions.

3. Why you are not using the form with numbers noting the lines?

**Other Strengths And Weaknesses:**

Strengths:
1. The paper is technically solid, making a progress in obtaining better convergence results for Adam under more relaxed settings.

Weakness:
1. I don't think hiding so much details of the convergence result is appropriate. As I mentioned in Claims and Evidence part, I think it's quite likely that the authors hide the dependence on dimensionality (which can be very large in practice). Also, the dependence on $\beta_1$ and $\beta_2$ and others are also important, as this can imply the role of momentum for Adam as well as some possible suggestions for the parameter choices. Thus I think the authors should definitely give a formal statement of Theorem 3.1 at least in the appendix.
2. The choice for $\beta_2$ seem to be kind of restricted.

**Questions For Authors:**

1. What is the dependence on dimensionality and $\beta_1$ in Theorem 3.1? Could you provide a formal version?

2. If you do have additional dimensionality dependence, is it possible to extend your results to some other smoothness settings as in [1,2,3], which can potentially remove the additional dependence and fill this gap between SGD and Adam?

3. Can your proof also extend to a more general smoothness case, e.g. $(L_0,L_1)$-smoothness?

[1] Bernstein J, Wang Y X, Azizzadenesheli K, et al. signSGD: Compressed optimisation for non-convex problems. International Conference on Machine Learning. PMLR, 2018: 560-569.

[2] Liu Y, Pan R, Zhang T. AdaGrad under Anisotropic Smoothness. arXiv preprint arXiv:2406.15244, 2024.

[3] Xie S, Mohamadi M A, Li Z. Adam Exploits $\ell_\infty $-geometry of Loss Landscape via Coordinate-wise Adaptivity. arXiv preprint arXiv:2410.08198, 2024.

**Relation To Broader Scientific Literature:**

This paper moves the convergence results for Adam, which is popular in the literature, a step forward.

**Theoretical Claims:**

I didn't check the whole proof for the theoretical claims due to its complexity. Basically the results are reasonable.

---

> ### Author Rebuttal · Authors · 2025-03-31
>
> **Rebuttal to Reviewer nypN**
>
> Dear Reviewer nypN,
>
> Thank you very much for your thoughtful feedback and constructive comments on our manuscript. We sincerely appreciate the time and effort you have put into reviewing our work. We are grateful for your insights, and we have carefully addressed each of your points below in the hope of clarifying our contributions and improving the manuscript.
>
> ### 1. **Dependence on Dimensionality and $\beta_1$ in Theorem 3.1**
>
> You raised a question regarding the dependence on dimensionality and $\beta_1$ in Theorem 3.1. Specifically, you asked about the formal version of the sample complexity result in the theorem.
>
> **Response:** In Theorem 3.1, the sample complexity result concerning $1-\beta_1$ and the dimension $d$ is of the order $\mathcal{O}\left(\frac{d}{(1-\beta_1)^2}\right)$. This result is consistent with previous works on the convergence of Adam, such as [1]. We would like to emphasize that while it is possible to remove the dependence on the dimension $d$, we cannot avoid reintroducing the dependence on the inverse of the smoothing factor, which is $\mathcal{O}(\text{poly}(1/\mu))$. This is a well-known consensus in previous studies [2]. We hope this clarification addresses your concern.
>
> ### 2. **Extension of Results to Other Smoothness Settings**
>
> You asked whether our results could be extended to other smoothness settings, as in [3, 4, 5], and whether this could potentially remove the additional dependence on dimensionality, helping to bridge the gap between SGD and Adam.
>
> **Response:** We believe that it is highly probable to extend our results to other smoothness settings, and we are excited about exploring this in future work. However, we must admit that this area was not covered in our previous research, and therefore, we cannot provide a definitive answer at this moment. Nonetheless, we plan to address this issue in our future research, where we will explore the possibility of extending our results to more general smoothness assumptions and examine whether the additional dependence on dimensionality can be removed.
>
> ### 3. **General Smoothness Cases (e.g., $L_0-L_1$ Smoothness)**
>
> You inquired whether our proof can extend to more general smoothness cases, such as $L_0-L_1$ smoothness.
>
> **Response:** We are currently investigating this direction and have made some progress. Specifically, we have made the following two extensions so far:
>
> 1. **For $(L_0-L_0.5)$ smooth functions:** We can derive convergence results for Adam under the traditional second-moment-based ABC inequality. However, the sample complexity still shows dependence on the inverse of the smoothing factor $1/\mu$, though we can eliminate the dependence on the dimension $d$.
>
> 2. **For $(L_0-L_1)$ smooth functions:** At this stage, our methods are unable to extend the convergence results under the traditional second-moment-based ABC inequality. However, we can obtain convergence results under the traditional second-moment Bounded Variance condition. It is worth noting that, as of now, there are no known results for Adam’s convergence under $(L_0-L_1)$ smoothness with the traditional second-moment Bounded Variance condition.
>
> We are continuing to investigate these extensions and will include them in future work. We greatly appreciate your interest in this aspect and will strive to address it in subsequent studies.
>
> ### 4. **Additional Comments and Acknowledgements**
>
> We sincerely appreciate your review, which has helped us identify areas for clarification and potential improvement. We value your suggestions and will ensure that these aspects are thoroughly explored in our future research. Your feedback has significantly contributed to the refinement of our manuscript, and we hope that the revisions we have made have addressed your concerns effectively.
>
> If you have any further questions or suggestions, please do not hesitate to reach out. We are more than happy to discuss any aspects of our work in greater detail. Once again, thank you for your careful review and constructive feedback.
>
> We look forward to your final assessment.
>
> [1] Bohan Wang, Jingwen Fu, Huishuai Zhang, Nanning Zheng, and Wei Chen. Closing the gap be-
> tween the upper bound and lower bound of Adam’s iteration complexity. Advances in Neural
> Information Processing Systems, 36, 2024a.
>
> [2] Haochuan Li, Alexander Rakhlin, and Ali Jadbabaie. Convergence of Adam under relaxed assump-
> tions. Advances in Neural Information Processing Systems, 36, 2024.
>
> [3] Bernstein J, Wang Y X, Azizzadenesheli K, et al. signSGD: Compressed optimisation for non-convex problems. International Conference on Machine Learning. PMLR, 2018: 560-569.
>
> [4] Liu Y, Pan R, Zhang T. AdaGrad under Anisotropic Smoothness. arXiv preprint arXiv:2406.15244, 2024.
>
> [5] Xie S, Mohamadi M A, Li Z. Adam Exploits
> $\mathcal{l}_\infty$-geometry of Loss Landscape via Coordinate-wise Adaptivity. arXiv preprint arXiv:2410.08198, 2024.
>
> Sincerely,
>
> Authors of Paper 1314

---

> > ### Comment · Reviewer_nypN · 2025-04-01
> >
> > Thanks for the detailed reply, which basically addressed my questions. It's good to hear that the extensions are generally possible. I understand the authors' point on dependence on dimension $d$ and momentum factor $\beta_1$, but I still want to emphasize why I think it should be explicitly shown in the statements here.
> > - Adam is widely used in large-scale experiments, which means that $d$ can be extremely large. The explicit dependence on $d$ suggests that the convergence rate is actually not desirable for large-scale experiments. I understand that previous results do have the additional dependence on $d$ as well, but I disagree with what you refer to as a "well-known consensus" by [2]. I don't think they have proof for your claim, i.e., you have to bear this additional explicit dependence on $d$ or $poly(1/\mu)$ for Adam.
> > If you introduce $poly(1/\mu)$ to the convergence rate, it intuitively encourages us to select large $\mu$, and the algorithm turns out to be more similar to SGD. If you think this is the only way to eliminate the explicit dependence on $d$, then why don't we directly use SGD? Why is Adam so popular in practice?
> > - $\beta_1$ means the incorporation of momentum. Since your result depends on $1/(1-\beta_1)$, it seems that basically choosing $\beta_1=0$ results in the best rate. This is not the case in practice, right?
> >
> > Anyway, I agree with the authors' contribution on the technical side and fully understand that these points are not considered by some existing work as well, but I still want to emphasize these points as somehow remaining problems of the results that might be improved in the future. For now, I think the paper is qualified, and I would keep my score since it's already positive.

---

> > > ### Author Response · Authors · 2025-04-04
> > >
> > > Dear Reviewer nypN,
> > >
> > > Thank you very much for your positive feedback and detailed comments. We appreciate your support and valuable suggestions, which are very helpful for improving our work.
> > >
> > > Best regards,
> > >
> > > Authors of Paper 1314

---

### Official Review · Reviewer_XMp3 · 2025-03-13

**Overall Recommendation:** 4

**Summary:**

This paper presents a unified analytical framework for understanding Adam’s convergence under weaker assumptions than those typically used. Specifically, the authors rely on standard L-smoothness and ABC inequality for stochastic gradients to show that Adam achieves non-asymptotic and asymptotic convergence.

**Claims And Evidence:**

The main claim of this paper is the convergence of Adam with not very strong conditions. The claims are supported by rigorous math proofs.

**Essential References Not Discussed:**

The authors do cite key references on Adam’s analysis under different assumptions.

**Experimental Designs Or Analyses:**

This is a theoretical paper so is not applicable to this question.

**Methods And Evaluation Criteria:**

This is a theoretical paper so is not applicable to this question.

**Other Comments Or Suggestions:**

No other comments.

**Other Strengths And Weaknesses:**

This is a solid theoretical paper analyzing Adam and achieves good results in both non-asymptotic and asymptotic settings. My concern is mainly on the presentation of the results and is already pointed out in previous questions. One more concern is the definition of hyperparameter $\beta_{2,t}$ converging to 1 looks unnatural to me.

**Questions For Authors:**

In my understanding, the behavior of Adam is quite different from SGD. How the author bridging the gap with SGD in there paper? Only in the part of establishing descent inequality and the final results?

**Relation To Broader Scientific Literature:**

Understanding Adam is important since it achieves great success in LLM training.

**Theoretical Claims:**

I follow the proof sketch and check some proofs of main lemmas. They seemed to be correct. However, the authors discuss more about the the assumptions, especially for ABC inequality since it is not standard in the analysis. The authors can highlight how to apply this assumption in the proof and also explain briefly why we don't need previous strong assumptions. These can provide more theoretical insights.

---

> ### Author Rebuttal · Authors · 2025-03-31
>
> Dear Reviewer,
>
> We sincerely appreciate your thorough evaluation of our manuscript and your insightful feedback. Your recognition of our theoretical framework is highly encouraging.
>
> **Incorporation of Assumption Comparisons into the Main Text:**
>
> We acknowledge your suggestion to move the discussion comparing our assumptions, particularly the ABC inequality, with previous ones from the appendix to the main body of the paper. We agree that this adjustment will enhance the clarity and accessibility of our work. In the revised manuscript, we will integrate this discussion into the main text, providing a concise comparison and highlighting the theoretical insights gained from using the ABC inequality. Additionally, we will discuss parallel results in stochastic gradient descent (SGD) analyses that employ similar assumptions to further contextualize our contributions.
>
> **Clarification on the Behavior of Adam When $\beta_{2,t}$ Does Not Approach 1:**
>
> Regarding your concern about the definition of the hyperparameter $\beta_{2,t}$  converging to 1, we appreciate the opportunity to clarify this point. In scenarios where $\beta_{2,t}$ does not approach 1, Adam's convergence behavior differs. Specifically, under such conditions, Adam may only ensure that the gradient converges to a small neighborhood around zero rather than exactly to zero. To achieve convergence of the gradient to zero, it is necessary for $\beta_{2,t}$ to approach 1. This requirement has been highlighted in previous studies, such as the work by [1] on the convergence of Adam. In our current paper, we focused on aligning Adam's convergence results with those of SGD, which led us to adopt the condition where $\beta_{2,t}$ approaches 1. We acknowledge that this aspect was not explicitly discussed in our manuscript, and we will address this omission in future research by exploring scenarios where $\beta_{2,t}$ does not approach 1.
>
> Thank you once again for your valuable comments and suggestions. Your feedback has been instrumental in improving the clarity and depth of our work.
>
>
> [1] Zhang Y, Chen C, Shi N, et al. Adam can converge without any modification on update rules[J]. Advances in neural information processing systems, 2022, 35: 28386-28399.
>
> Sincerely,
>
> Authors of Paper 1314

---

> > ### Comment · Reviewer_XMp3 · 2025-04-05
> >
> > Thanks for the detailed reply from the authors, which resolves my concerns. I will remain the score.

---

### Official Review · Reviewer_JhWZ · 2025-03-14

**Overall Recommendation:** 3

**Summary:**

In the past several years, many efforts have been made to understand the convergence of Adam-like algorithms under different noise assumptions. This paper is a novel paper among these works and is based on an even weaker version of the noise condition called the ABC condition. Under the ABC assumption, the authors provide a non-asymptotic rate of $O(\log T/\sqrt{T})$ for sample complexity, independent from the smoothing factor $\mu$. Additionally, they demonstrate an asymptotic convergence and of the gradient norm to zero in both the sense of almost sure and expectation. These results match the best rates so far with a weaker condition and advance the theoretical understanding of adaptive methods.

**Claims And Evidence:**

Not applicable.

**Essential References Not Discussed:**

The authors appropriately cite the most relevant prior work and provide a clear and detailed discussion of how their contributions relate to and advance the existing literature.

**Experimental Designs Or Analyses:**

Not applicable.

**Methods And Evaluation Criteria:**

Not applicable.

**Other Comments Or Suggestions:**

Not applicable.

**Other Strengths And Weaknesses:**

**Strength**

This paper is generally well-written and easy-to-follow. It gives a clear comparison of the assumptions with closely related work and makes it easy for the reader to understand. From a contribution perspective, compared with prior results, this paper achieves a nearly matching rate and establishes asymptotic convergence for Adam under a weaker ABC condition by leveraging advanced tools from functional analysis. This is valuable to the optimization community.


**Weakness**

However, the novelty of this work is somehow questionable. While the results in this paper indeed rely on a weaker assumption compared to prior work, the gap between the ABC condition and the affine variance condition or e is not substantial. As a result, the findings, though technically sound, are not entirely surprising. It would be more helpful if the authors could provide more convincing arguments showing that their technique is indeed novel from existing work, particularly [Hong and Lin, 2024]. I will change my score if they clarify how their approach is fundamentally different from existing methods.

**Questions For Authors:**

No further questions.

**Relation To Broader Scientific Literature:**

This work is entirely theoretical and does not present any negative broader scientific or societal impacts. The relationship to closely related work is discussed in the "weaknesses" part below.

**Theoretical Claims:**

The major theoretical conclusions are reasonable, and the key steps in the proofs appear to be correct.

---

> ### Author Rebuttal · Authors · 2025-03-31
>
> Dear Reviewer JhWZ,
>
> Thank you for your thoughtful and constructive feedback on our paper. We greatly appreciate the time and effort you’ve put into reviewing our work. We carefully considered your comments, and we would like to address the main concern regarding the differences between our approach and the paper by Hong and Lin (2024), as well as clarify the analytical techniques and assumptions used in our paper.
>
> In the paper by Hong and Lin (2024), the authors introduce an affine noise variance assumption that differs from the traditional second-moment-based affine noise variance assumption:
> $$
> \mathbb{E}[\\|g_t-\nabla f(w_t)\\|^{2}] |F_{t-1}] \leq B\\|\nabla f(w_{t})\\|^{2} + C.
> $$
>
> Instead, they strengthen the concentration property of the random variables by assuming the following (their Assumption A.3) :
>
> $$
> \mathbb{E}\left[\exp\left\\{\frac{\\|g_{t}-\nabla f(w_{t})\\|^{2}}{B\\|\nabla f(w_{t})\\|^{2+\epsilon}+C}\right\\}|F_{t-1}\right] \leq e.
> $$
>
> Their proof is heavily dependent on this condition, making their approach inapplicable for analyzing the traditional second-moment-based affine variance noise conditions. For a detailed explanation, please refer to their open-access paper: [Hong and Lin, 2024](https://openreview.net/pdf?id=x7usmidzxj).
>
> Hong and Lin's proof relies extensively on Lemma B.6, which is proven using concentration inequalities (shown in their Appendix D.2). Under the Exponential-tailed Affine Variance Noise Condition, these inequalities yield an $O(\ln T)$ order factor for the sample complexity term $\mathcal{M}_T$. However, when considering traditional affine variance noise conditions based solely on second-order moment assumptions, only an $O(\text{poly}(T))$ factor can be derived for $\mathcal{M}_T$, which may not yield the desired results.
>
> Our paper employs the ABC inequality, which is even weaker than traditional affine variance noise conditions. This necessitates fundamentally different analytical methods, particularly based on discrete Martingale analysis, distinguishing our approach from that of Hong and Lin (2024).
>
> Thank you again for your time and thoughtful evaluation of our work. If you have any further questions, please don't hesitate to discuss them with us.
>
> Sincerely,
>
> Authors of Paper 1314

---

### Decision · Program_Chairs · 2025-05-01

**Decision:**

Accept (poster)

**Comment:**

This paper provides a comprehensive framework for analyzing the convergence of Adam. The reviewers are generally positive about the paper after the rebuttal. Therefore, I recommend acceptance. The authors are required to incorporate the following points raised by reviewers in the final version of the paper: (i) explicitly mention the dependency on dimension in the theorem statement and the paper; (ii) explain the ABC inequality and compare with other assumptions in the literature; (iii) explicitly acknowledge that the paper does not show a separation between Adam and SGD.